# Deep Networks Learn to Parse Uniform-Depth Context-Free Languages from Local Statistics

**Jack T. Parley** [1]   **Francesco Cagnetta** [2]   **Matthieu Wyart** [3] [1]

## Abstract

Understanding how the structure of language can be learned from sentences alone is a central question in both cognitive science and machine learning. Studies of the internal representations of Large Language Models (LLMs) support their ability to parse text when predicting the next word, while representing semantic notions independently of surface form. Yet, which data statistics make these feats possible, and how much data is required, remain largely unknown. Probabilistic context-free grammars (PCFGs) provide a tractable testbed for studying these questions. However, prior work has focused either on the post-hoc characterization of the parsing-like algorithms used by trained networks; or on the learnability of PCFGs with fixed syntax, where parsing is unnecessary. Here, we *(i)* introduce a tunable class of PCFGs in which both the degree of ambiguity and the correlation structure across scales can be controlled; *(ii)* provide a learning mechanism—an inference algorithm inspired by the structure of deep convolutional networks—that links learnability and sample complexity to specific language statistics; and *(iii)* validate our predictions empirically across deep convolutional and transformer-based architectures. Overall, we propose a unifying framework where correlations at different scales lift local ambiguities, enabling the emergence of hierarchical representations of the data.

[1]Institute of Physics, École Polytechnique Fédérale de Lausanne (EPFL), Lausanne, Switzerland [2]Theoretical and Scientific Data Science, SISSA, Trieste, Italy [3]Department of Physics and Astronomy, Johns Hopkins University, Baltimore, Maryland. Correspondence to: <jack.parley@epfl.ch>, <francesco.cagnetta@sissa.it>, <mwyart1@jh.edu>.

*Proceedings of the $43^{rd}$ International Conference on Machine Learning*, Seoul, South Korea. PMLR 306, 2026. Copyright 2026 by the author(s).

## 1. Introduction

How languages are learned is a central question of linguistics. The celebrated "poverty of the stimulus" argument (Chomsky, 1980) questioned whether learning from examples alone was even possible, and led to the proposition that grammar is innate. The distributional approach, instead, posited that the statistics of the input data are sufficient to deduce the language structure (Ellis, 2002; Saffran & Kirkham, 2018; Saffran et al., 1996)—a possibility later established by the advent of Large Language Models (LLMs) (Devlin et al., 2019; Radford et al., 2018). Understanding the reasons for their success, however, remains a central challenge. Studies of internal representations of LLMs indicate that (i) they parse text (Peters et al., 2018; Tenney et al., 2019; Manning et al., 2020; Diego-Simón et al., 2024) and (ii) they build representations of semantic notions, such as the "Golden Gate Bridge" (Templeton et al., 2024; Gurnee et al., 2026), which are extremely robust toward variations in syntax. The mechanism whereby these two feats are learned is, however, essentially unknown, including which correlations in text are exploited, and how much training data is required to do so.

Inspired by the symbolic generative approach (Chomsky, 1957), probabilistic context–free grammars (PCFGs) provide a powerful tool to model languages and to study the questions above quantitatively. In these models, a tree of latent (or "nonterminal") variables underlies the text structure. These latent variables can be used to model syntax, but also abstract semantic notions. The classic bottom-up method to parse a PCFG is the inside algorithm (Baker, 1979), a sum-product algorithm which recursively computes the probability of latent symbols deriving spans of a sentence. Recent works have shown that transformers approximately implement the inside algorithm (Allen-Zhu & Li, 2025; Zhao et al., 2023) and can perform context-free recognition (Jerad et al., 2026), but do not address how such algorithms are learned during training nor how much data is needed.

A strategy to study learnability (Clark, 2017) is to consider synthetic PCFGs of controlled structure (Malach & Shalev-Shwartz, 2018; Mossel, 2016). In (Cagnetta et al., 2024), the Random Hierarchy Model (RHM) was introduced, a fixed-

tree PCFG where production rules (describing how latent variables generate strings of other variables) are chosen randomly. The mechanism for how deep networks learn such data and the associated sample complexity were elucidated for supervised learning (Cagnetta et al., 2024), next token prediction (Cagnetta & Wyart, 2024) and diffusion models (Favero et al., 2025): strings of tokens that correlate with the tasks are grouped together, in a recursive manner, allowing the network to reconstruct the tree of latent variables hierarchically. This hierarchical reconstruction of latent variables closely mirrors the Bayes-optimal algorithm for fixed-tree problems where parsing is not required, i.e. belief propagation (Sclocchi et al., 2025; Mei, 2025). Already in the fixed-tree case, depth is essential to efficiently learn such algorithms (Cagnetta et al., 2024). Furthermore, if production rules are not random but fine-tuned to remove the task-relevant correlations, learning becomes virtually impossible in high dimension (Cagnetta et al., 2024); showing that the learnability of worst and typical cases differ. A central limitation of these works is that they consider fixed trees—syntax is frozen, and parsing is not necessary. Removing the fixed-topology assumption profoundly changes the learning problem: (A) learners need to decipher which span of the text corresponds to a given latent variable, and (B) ambiguity appears, where the same substring of the sentence could have been generated by different latent variables.

## 1.1. Our contribution

(i) We introduce a family of synthetic grammars with random production rules and varying trees, obtained by letting production rules generate strings of different lengths. Although the number of parse trees grows exponentially with the full sentence length, we show that global ambiguity (having multiple parse trees with different root labels per sentence) is controlled by a crossover driven by the number of production rules. This property allows us to have detailed control of the level of ambiguity.

(ii) We provide a mechanism for learning the root classification task (and discuss the implications of our findings for next-token prediction in the conclusion), in the form of an algorithm that can learn the production rules so as to parse sentences into a hierarchical tree of latent variables. This analysis, which builds on iterative clustering based on correlations (Malach & Shalev-Shwartz, 2018; Cagnetta et al., 2024) to identify latent variables, overcomes problems (A) and (B). It predicts that the sample complexity scales as a power law of the number of production rules and vocabulary size, corresponding to the minimal sample size where correlations between the root and substrings of adjacent tokens can be measured precisely.

(iii) We test these predictions empirically both for convolutional and transformer-based deep neural networks, and

confirm our predictions quantitatively.

## 1.2. Previous works

**Languages with random rules:** In (DeGiuli, 2019), it was shown that a transition toward fully structureless, noisy languages can take place. In (Hsu et al., 2012) the impossibility of learning was established under similar conditions.

**Learning PCFGs** is a classic example of learning a latent-variable probabilistic model. The standard approach relies on maximum likelihood and expectation-maximization (EM) (Dempster et al., 1977), involving iterative re-estimation of rule probabilities (Baker, 1979; Pereira & Schabes, 1992; Klein & Manning, 2004). However, the application of EM to infer a PCFG offers no guarantees of convergence to global minima (Balle et al., 2014). Of particular relevance for us is the "hard" setting, where training data does not include parse trees. Progress in this setting is limited: (Bailly et al., 2013) cast PCFG inference as a low-rank factorization, which, however, leads to an intractable optimization problem, while (Clark & Yoshinaka, 2013) showed that substitutable context-free languages are identifiable in the large samples limit from positive examples, but neither provides finite sample complexity guarantees. (Cohen & Smith, 2012) give a bound on sample complexity if some optimizer is found, but they show that this optimization is NP hard in general. This motivates the introduction of our controlled synthetic PCFGs, which allow us to extract quantitative predictions of sample complexity even in the "hard" setting.

## 2. Varying-tree Random Hierarchy Model

### 2.1. Generative Model

In this section, we define the family of synthetic grammars that we use as a tractable model of language data.

**Definition 2.1** (PCFG). A Probabilistic Context-Free Grammar (PCFG) is a tuple $\mathcal{G} \coloneqq (\Sigma, \mathcal{N}, S, \mathcal{R}, \mathcal{P})$, where

  i) $\Sigma$ is a finite set of terminal symbols;

 ii) $\mathcal{N}$ is a finite set of nonterminal symbols;

iii) $S \in \mathcal{N}$ is the start symbol;

 iv) $\mathcal{R}$ is a finite set of production rules $A \to \alpha$, where $A \in \mathcal{N}$ is a single nonterminal and $\alpha \in (\mathcal{N} \cup \Sigma)^*$ is a string of terminals and/or nonterminals;

  v) $\mathcal{P}$ is a probability distribution over rules $(A \to \alpha)$, for each $A \in \mathcal{N}$, $\sum_{\alpha|(A\to\alpha)\in\mathcal{R}} \mathcal{P}(A \to \alpha) = 1$.

A PCFG describes a probability over strings of terminals, $P_{\mathcal{G}}(\boldsymbol{x})$. Starting from $S$, the generation of such strings—or

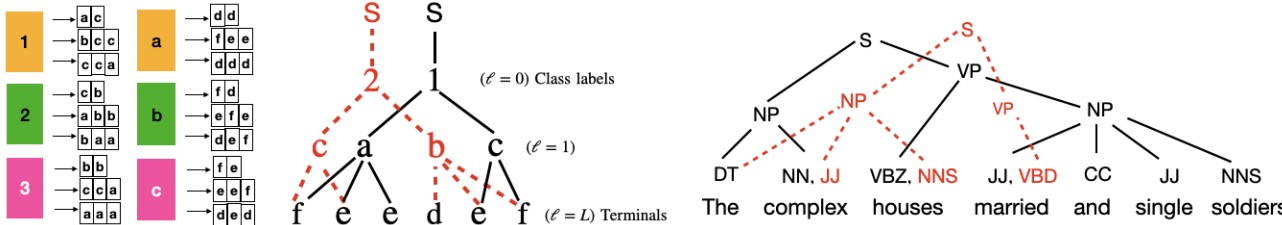

*Figure 1. Left:* example grammar instance of a varying-tree RHM, with $L=2$, $v=3$ symbols per layer and $m_2=1$, $m_3=2$ binary/ternary rules per symbol. For each higher-level symbol, rules are sampled uniformly at random and without replacement from the set of $v^2$ $(v^3)$ binary (ternary) tuples of lower-level symbols. *Middle:* starting from the start symbol $S$, a sentence belonging to the grammar is sampled by iteratively re-writing nonterminals according to the rules, in this case yielding the sentence "feedef". Parsing this sentence, however, reveals an additional grammatical parse tree with different class label (red). This is due to the local ambiguity caused by overlapping rules. *Right:* example of a "garden-path" sentence, used in cognitive science (Frazier & Rayner, 1982) to study how humans parse, with both lexical and syntactic ambiguity. Reading *the complex houses* as a noun phrase (NP) leads to an additional parse tree in red, analogously to the middle panel (note however it does not form a complete span).

*derivation*—proceeds by first selecting a rule for each nonterminal symbol according to $\mathcal{P}$, then replacing that symbol with the right-hand side of the rule. The process repeats until there are only terminals left. The string probability is the product of the probabilities of the rules used in its derivation. The associated context-free *language* is the set of all terminal strings $\boldsymbol{x}$ such that $P_{\mathcal{G}}(\boldsymbol{x}) > 0$.

**Definition 2.2** (Layered PCFG). A layered PCFG is a PCFG where the sets of nonterminals and rules factors into $L$ layers, $\mathcal{N} = \bigcup_{\ell=0}^{L-1} \mathcal{N}^{(\ell)}$ with $\mathcal{N}^{(L)} \equiv \Sigma$, and $\mathcal{R} = \bigcup_{\ell=0}^{L} \mathcal{R}^{(\ell)}$. The rules connect one level to the next: $\mathcal{R}^{(0)}$ contains $(S \to \alpha)$ rules with $\alpha \in (\mathcal{N}_0)^*$, each $\mathcal{R}^{(\ell)}$ with $\ell \geq 1$ contains $(A \to \alpha)$ rules with $A \in \mathcal{N}_{\ell-1}$ and $\alpha \in (\mathcal{N}_\ell)^*$.

**Definition 2.3** (Varying-Tree RHM). A Varying-tree Random Hierarchy Model (RHM) is a layered PCFG where:

i) Level-0 rules connect $S$ to a single nonterminal in $\mathcal{N}_0$;

ii) For each level $\ell = 1, \ldots, L$ and nonterminal $z \in \mathcal{N}_{\ell-1}$, there are $m_2$ binary rules $(z \to ab)$ and $m_3$ ternary rules $(z \to abc)$ with $a, b, c \in \mathcal{N}_\ell$;

iii) For each parent nonterminal $z \in \mathcal{N}_{\ell-1}$, the pairs $ab$ (resp. triples $abc$) appearing in the rules are sampled uniformly without replacement from the set of all possible pairs (resp. triples) over $\mathcal{N}_\ell$, of size $|\mathcal{N}_\ell|^2$ (resp. $|\mathcal{N}_\ell|^3$).

An example grammar instance constructed in this manner is shown in Fig. 1.

For simplicity, we assume that $|\mathcal{N}_0| = \ldots = \mathcal{N}_L = v$. Furthermore, we assume that level-0 rules have probability $1/v$, binary rules have probability $p_2/m_2$, and ternary rules have probability $p_3/m_3$. These assumptions correspond to having uniform probability within each class of rules, and branching probability $p_s$ for $s \in \{2, 3\}$ at each node of the tree. We stress that relaxing these assumptions does not affect our theoretical approach. We parametrize the number of

rules as $m_2 = f_2 v$ and $m_3 = f_3 v^2$, where $f_2$ and $f_3$ denote the fraction of admissible pairs and triples.

The Varying-Tree RHM generates sentences with random length $D$, varying in the integer interval $[2^L, \ldots, 3^L]$. The sentence length probability $P(d, L)$, studied in App. A, is centered around the *typical* length $\langle s \rangle^L$, where $\langle s \rangle = 2p_2 + 3p_3$. Some sentence lengths can be achieved via multiple tree topologies, $N_t(d, L)$ for length $d$ and depth $L$. Close to the typical length, in particular, this number is exponential in the sentence length, $N_t(\langle s \rangle^L, L) \sim 2^{(\langle s \rangle^L - 1)/(\langle s \rangle - 1)}$ as shown in App. A. In addition, due to the combinatorial structure of the generative model, even for fixed length $d$ and tree topology, the number of possible sentences grows exponentially with $d$ (Cagnetta et al., 2024).

### 2.2. Local Ambiguity

Due to the sampling without replacement of the rules (2.3, *iii*)), the individual rules are unambiguous: any valid pair or triple of symbols can only be generated by one nonterminal symbol. Nonetheless, the multiple branching ratios lead to ambiguity. For instance, fix a triple of terminals $(abc)$ such that $(z \to abc) \in \mathcal{R}^{(l)}$ for some nonterminal $z$. There is a probability $f_2$ over rules sampling that some binary rule produces $(ab)$, e.g. $(z' \to ab) \in \mathcal{R}^{(l)}$. Then the substring $(abc)$ of level-$\ell$ symbols admits at least two distinct derivations with nonzero probability: (i) directly via $(z \to abc)$, and (ii) via $(z' \to ab)$ followed by a rule whose expansion begins with $c$, which occurs with probability $O(1/v)$. The overall probability of the fortuitous occurrence is then $O(p_2/(m_2 v))$: when this is comparable with the ternary rule probability $p_3/m_3$, the grammar is *locally* ambiguous.

### 2.3. Notation & Learning Task

A Varying-Tree RHM generates a probability distribution over trees of depth $L$ with branching ratios of 2 or 3. We denote the random variables corresponding to nonterminal

and terminal symbols with capital letters, using lowercase $(a, b, c)$ for specific realizations. We indicate the level of variables and rules with a superscript, e.g. $X^{(\ell)}$.

We consider a *root classification* problem, where the learner is given sequences of terminals, or sentences, $\boldsymbol{X} \equiv \boldsymbol{X}^{(L)}$, and tasked to predict the single level-0 nonterminal $Y \equiv X^{(0)}$ at the root of the derivation. The training dataset consists of $P$ sentence-root pairs $(\boldsymbol{X}, Y)$, sampled i.i.d. from a fixed instance of the Varying-Tree RHM.

Given knowledge of the rules, the Bayes-optimal algorithm for performing root classification given a sentence is the inside parsing algorithm (Baker, 1979), which can be seen as the probabilistic (sum-product) instantiation of a general parser.

**Definition 2.4** (Rule-based parser for layered PCFG with binary and ternary rules). A parser is a dynamic program that fills a parsing chart $M$, defined for each span $(i, \lambda)$ (start position $i$, length $\lambda$) and nonterminal of the grammar. In a layered PCFG, given the the rule tensors [1] $R^{(\ell)}$, the rule-based parser computes the level-$(\ell-1)$ chart $M^{(\ell-1)}$ following the recursion

$$M_{i,\lambda}^{(\ell-1)}[z] = \sum_{\lambda', a, b} R_{z \to (a,b)}^{(\ell)} \times M_{i,\lambda'}^{(\ell)}[a] \times M_{i+\lambda', \lambda-\lambda'}^{(\ell)}[b]$$
$$+ \sum_{\lambda', \lambda'', a, b, c} R_{z \to (a,b,c)}^{(\ell)} \times$$
$$M_{i,\lambda'}^{(\ell)}[a] \times M_{i+\lambda', \lambda''}^{(\ell)}[b] \times M_{i+\lambda'+\lambda'', \lambda-\lambda'-\lambda''}^{(\ell)}[c].$$

The parser is sketched in Fig. 2 (top). For each sentence $x_1 \ldots x_d$, the chart is initialized at the $L$-th (terminal) level as $M_{i,1}^{(L)}[z] = \delta_{x_i, z}$. Above, we have expressed the parser in its sum-product version (inside algorithm), so that the chart stores inside probabilities of substrings, $M_{i,\lambda}^{(\ell)}[z] = Pr(x_i, \ldots, x_{i+\lambda-1} \mid z)$ with $z \in \mathcal{N}_\ell$. Different choices of operation recover other parsing algorithms. In particular, replacing $(+, \times)$ by $(\vee, \wedge)$ (Boolean parsing) recovers recognition/CYK (Younger, 1967). All such algorithms can be unified under the semiring parser formulation (Goodman, 1999).

**Proposition 2.5** (Bayes-optimal label prediction). *Consider a layered PCFG with prior $p(y)$ over class label symbols $y \in \mathcal{N}^{(0)}$. Let $\mathbf{x} = (x_1, \ldots, x_d)$ be an observed sentence, and let $M_{1,d}^{(0)}[y]$ be the output of the inside algorithm for the chart entry corresponding to a full span. Then $M_{1,d}^{(0)}[y] = Pr(\mathbf{x} \mid y)$, and the Bayes-optimal predictor of the class*

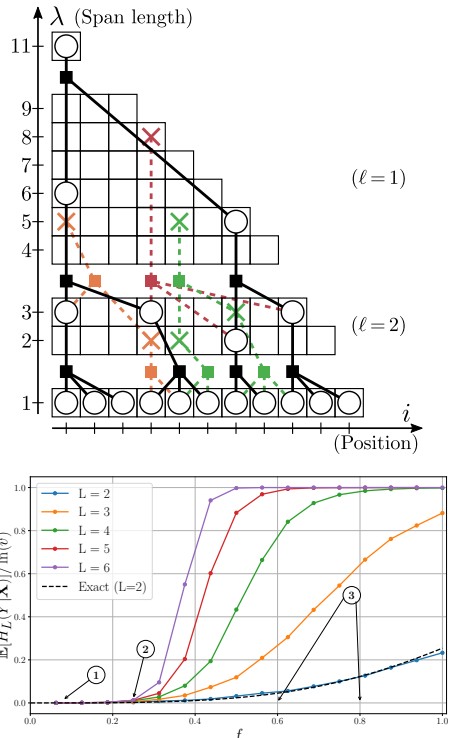

*Figure 2.* Top: illustration of the rule-based parser and the appearance of *spurious nonterminals* due to *local ambiguity*. Depicted are the charts $M_{i,\lambda}^{(\ell)}[z]$ created while parsing a sentence of length $d=11$ and depth $L=3$. The third axis corresponding to nonterminals is not shown explicitly; instead we only distinguish between nonterminals belonging to the true parse tree (circles) and spurious ones (crosses). Bottom: global ambiguity as measured by the inside algorithm. Shown is the expected class entropy, normalized by $\ln(v)$. We consider $v = 16$, representative of the finite $v$ values considered in our experiments. Black dashed lines show $\mathbb{E}[H_{L=2}(Y|\boldsymbol{X})]$, which can be computed exactly (App. A). Pointers indicate the three regimes of $f$ where we measure sample complexity of deep nets.

*label is given by*

$$\hat{y}(\mathbf{x}) = \arg \max_{y \in \mathcal{N}^{(0)}} p(y) M_{1,d}^{(0)}[y].$$

Beyond label prediction, the inside algorithm also yields the conditional class entropy $H(Y \mid \boldsymbol{X})$, which we additionally average over grammar instances and generated sentences to obtain $\mathbb{E}_{\mathcal{G}} \left[ \mathbb{E}_{\boldsymbol{X} \sim P_{\mathcal{G}}} [H(Y \mid \boldsymbol{X})] \right]$. This constitutes a measure of *global ambiguity* of a typical sentence, as well as the information-theoretic lower-bound on the cross-entropy loss of any classifier. For finite $v \ll N_t$, the model displays a crossover with increasing $f$ [2] to a globally ambiguous regime where $H = \mathcal{O}(\ln(v))$, shown in Fig. 2 (bottom):

---

[1] The tensor entries are set to the rule probabilities: $p_2/m_2$ (resp. $p_3/m_3$) for every grammatical binary $z \to (a, b)$ (resp. ternary $z \to (a, b, c)$) rule. We refer to App. B for details on the tensorial formulation and our parallelization scheme.

[2] In our numerical analyses, we will set $f_2 = f_3 = f$ to have a single control parameter for ambiguity (note that our predictions for sample complexity will apply to general $f_2, f_3$), as well as $p_2 = p_3 = 1/2$.

this is controlled by a competition between the fraction of grammatical data and the number of tree topologies (see App. A). In the absence of global ambiguity, i.e. if each sentence $\mathbf{x}$ is compatible with at most one class label $y \in \mathcal{N}^{(0)}$, Boolean parsing is sufficient for label prediction, in which case we will use the indicator notation $N_{i,\lambda}^{(l)} \in \{0, 1\}$, and the prediction $\hat{y}(\mathbf{x})$ is the unique $y$ such that $N_{1,d}^{(0)}(y) = 1$.

## 3. Learning Algorithm and Sample Complexity

As outlined above, the task we propose to the learner is a *root classification* problem. We stress, however, that performance on the classification task for the Varying-Tree RHM *implies* parsing performance: learning strategies which do not re-build the latent tree structure *cannot* achieve polynomial sample complexity, as shown in detail for the fixed-tree RHM (a particular and simpler case of our more general model) in (Cagnetta et al., 2024). Our key question of whether and how deep nets learn to parse can therefore be phrased as follows:

**Q.** *Can deep networks learn the classification task with Bayes-optimal performance from a polynomial number of training samples?*

Our strategy to address this question will be to propose a provably correct *algorithm*, which uses only low-order moments of the data, allowing for a sample complexity which will only be polynomial in the typical sentence size as $L$ increases.

This algorithm, outlined in 1, and sketched in Fig. 3, is of dual nature. On the one hand, we will refer to it as a *parser*, in the sense of being a program as defined in 2.4 which iteratively fills in parsing charts (indicators) from layer to layer. In contrast to the rule-based parser, however, it does not possess prior knowledge of the rules, but will rather infer these from statistical regularities (clustering) across sentences as it progresses from layer to layer. In this second sense, therefore, it is an algorithm for *rule inference*. If the rules are inferred accurately, the parser achieves by construction Bayes-optimal performance on the classification task. We will later conjecture that deep neural networks also learn the root classification task by inferring rules from local statistics via the clustering mechanism: we will test this conjecture empirically, finding that they indeed display precisely the sample complexity predicted theoretically from our algorithm.

In our theoretical analysis, we will consider the limit $v \to \infty$, and focus on identification of the rules, as required for Boolean parsing (CYK). As alluded to above, this is Bayes-optimal when there is at most one label per sentence; in the limit considered, however, it can achieve better-than-chance

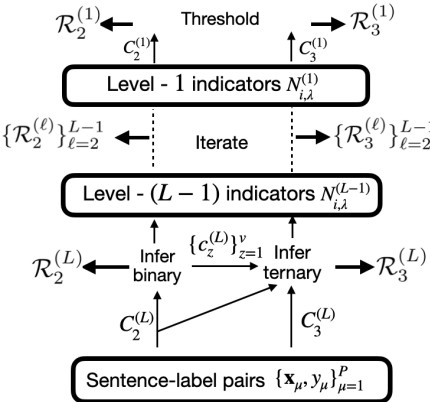

*Figure 3.* Top: Schematic of Algorithm 1, which, given sentence-label pairs, outputs the rule sets $\{\mathcal{R}_2^{(\ell)}, \mathcal{R}_3^{(\ell)}\}_{\ell=1}^L$. Plots visualizing and quantifying clustering performance of the empirically implemented algorithm on $P$ training data can be found in App. E.

performance for $\forall f$, given [3] that $H(Y \mid \mathbf{X}) \ll \ln(v)$ for $v \to \infty$.

Rule inference relies essentially on a *method of moments*, which uses low-order (root-to-pair and root-to-triple) statistics to infer one level of the grammar rules, before applying the rules to predict the next layer of nonterminals. The possibility to reconstruct rules from root-to-pair and root-to-triple covariances is granted by the context-free property of the generative model (Cagnetta et al., 2024). If two adjacent tokens $(X_i, X_{i+1})$ are children of the same nonterminal, their covariance with the label $Y$ depends on that nonterminal. However, local ambiguity hinders the inference of the grammar rules, as *(i)* the spans of the nonterminals are unknown, i.e. we don't know if a given pair/triple of tokens is generated by a single nonterminal; *(ii)* some rules overlap, i.e. some pairs of tokens $(a, b)$ can correspond either to a valid binary sibling pair or to a subsequence of a valid triple of ternary siblings $(a, b, c)$ or $(c, a, b)$.

**Binary rules.** This effect is illustrated in Fig. 4: the tokens can be binary siblings (probability $p_{\text{bin}}$), but also part of a triple of ternary siblings $(X_{I-1}, X_I, X_{I+1})$ or $(X_I, X_{I+1}, X_{I+2})$ (probability $p_{\text{ter}}$ each), or sit across the boundary between two spans (probability $p_{\text{cross}}$). The root-to-pair covariance from Eq. 1 takes contributions from all these cases, as is evident from a linear decomposition based on the law of total covariance, also illustrated in Fig. 4 (diagrams i) to iv), see subsection C.2 for details)). Nevertheless, in the asymptotic vocabulary size limit, $C_2^{(L)}$ is dominated by the contribution of binary siblings (diagram i). Therefore, the following algorithm 2 provably recovers binary rules (see 3.1, with proof in subsection C.2).

---

[3]The entropy is bounded by the number of tree topologies, which remains finite for bounded depth.

**Algorithm 1** Learning algorithm

**Input:** $P$ sentence-label samples $\{\boldsymbol{x}_\mu, y_\mu\}_{\mu=1}^P$
**Pair Cov:** Compute root-to-pair covariance (the average includes the pair position $I$)

$$C_2^{(L)} \;:=\; \mathrm{Cov}\left[(X_I, X_{I+1}), X^{(0)}\right] \in \mathbb{R}^{v \times v \times v} \quad (1)$$

**Triple Cov:** Compute root-to-triple covariance

$$C_3^{(L)} \;:=\; \mathrm{Cov}\left[(X_I, X_{I+1}, X_{I+2}), X^{(0)}\right] \in \mathbb{R}^{v^{\times 4}} \quad (2)$$

**for** $\ell = L$ **to** $2$ **do**
  **Binary rules:** $\mathcal{R}_2^{(\ell)} \leftarrow InferBinary(C_2^{(\ell)})$
  **Ternary rules:** $\mathcal{R}_3^{(\ell)} \leftarrow InferTernary(C_2^{(\ell)}, C_3^{(\ell)})$
  **Nonterminals:**
    **1.** Build candidate nonterminal indicators $N_{i,\lambda}^{(\ell-1)}(a)$ by applying the inferred level-$\ell$ rules (1 if the symbols from $i$ to $i + \lambda - 1$ can be produced by nonterminal $a$ and 0 otherwise)
    **2.** Form adjacent candidate pair/triple indicators by multiplication:

$$N_{i,\lambda,\lambda'}^{(\ell-1)}(a, b) = N_{i,\lambda}^{(\ell-1)}(a) N_{i+\lambda,\lambda'}^{(\ell-1)}(b) \quad (3)$$

  **if** $\ell > 2$ **then**
    **Covariances:** Average the covariance of pair and triple indicators with the root label over all valid positions and span lengths to obtain $C_2^{(\ell-1)}$ and $C_3^{(\ell-1)}$
  **else**
    **Last step:** Obtain $C_2^{(1)}$ and $C_3^{(1)}$ from the covariance between root and complete-span indicators, i.e. $N_{1,\lambda,\lambda'}(a, b)$ s.t. $\lambda + \lambda' = d$ (similarly for triples with $\lambda + \lambda' + \lambda'' = d$). These identify valid rules $(z \to ab) \in \mathcal{R}_2^{(1)}$ (resp. $(z \to abc) \in \mathcal{R}_3^{(1)}$), as they are the only to display an above-noise covariance, of order $O(1/(m_2 v))$ (resp. $O(1/(m_3 v))$).
  **end if**
**end for**
**Output:** rule sets $\{\mathcal{R}_2^{(\ell)}, \mathcal{R}_3^{(\ell)}\}_{\ell=1}^L$.

---

**Proposition 3.1** (Correctness of INFERBINARY from population root-to-pair covariances). *Consider the varying-tree RHM in the asymptotic regime $v \to \infty$, $m_2 = f_2 v$, $m_3 = f_3 v^2$ and fix $\ell \in \{2, \ldots, L\}$. For $(a, b) \in \mathcal{N}_\ell^2$, let*

$$u_{ab} := C_2^{(\ell)}((a, b), :) \in \mathbb{R}^v,$$

*and let $\mathcal{B}_\ell$ be the set of valid level-$\ell$ binary sibling pairs. Define*

$$\tau_2 := \gamma\, v^{-1} \left( \sum_{a,b} \|u_{ab}\|_2^2 \right)^{1/2},$$

*with $\gamma \in (0, 1)$. Then, with high probability over the ran-*

**Algorithm 2** INFERBINARY$(C_2^{(\ell)}, \tau_2)$

**Input:** root-to-pair covariance tensor $C_2^{(\ell)}$
**Output:** binary rule set $\mathcal{R}_2^{(\ell)}$ and cluster centroids $\{C_z^{(\ell)}\}_{z=1}^v$
**Pair slices:** For each pair $(a, b) \in [v]^2$, define

$$u_{ab} \;:=\; C_2^{(\ell)}((a, b), :) \in \mathbb{R}^v \quad (4)$$

**Valid pairs:** Keep only pairs $(a, b)$ such that $\|u_{ab}\|$ exceeds the threshold $\tau_2$
**Clustering:** Cluster the normalized vectors $\widehat{u}_{ab} := u_{ab}/\|u_{ab}\|$ of the valid pairs into $v$ groups
**Binary rules:** For each cluster $z$, define its centroid $c_z^{(\ell)}$ and, for every pair $(a, b)$ in that cluster, add the rule $(z \to ab)$ to $\mathcal{R}_2^{(\ell)}$

---

*dom grammar and for all sufficiently large $v$, one has:*

*(i) $\|u_{ab}\|_2 > \tau_2$ if and only if $(a, b) \in \mathcal{B}_\ell$;*

*(ii) for every valid pair $(a, b) \in \mathcal{B}_\ell$, the normalized vector $u_{ab}/\|u_{ab}\|_2$ converges to the normalized row of $C_1^{(\ell-1)}$ associated with its unique parent $z(a, b)$;*

*(iii) Consequently, thresholding at level $\tau_2$ selects exactly the valid binary sibling pairs, and clustering the surviving normalized vectors recovers their partition by parent. In particular, INFERBINARY reconstructs the level-$\ell$ binary rules up to permutation of parent labels.*

**Ternary rules.** The root-to-triple covariance is also affected by the same sources of ambiguity as $C_2^{(L)}$, as illustrated in Fig. 4 (diagrams v), vi) and vii)). In addition, even in the asymptotic limit, the contribution of ternary siblings (diagram v)) competes with those where two of the three tokens form a pair of binary siblings (vi) and vii)). This competition requires the subtraction of the spurious binary contributions, as illustrated in the following algorithm 3, which provably recovers ternary rules (see 3.2, with proof in subsection C.3).

**Proposition 3.2** (Correctness of INFERTERNARY from population root-to-pair and root-to-triple covariances). *Consider the varying-tree RHM in the asymptotic regime $v \to \infty$, $m_2 = f_2 v$, $m_3 = f_3 v^2$ and fix $\ell \in \{2, \ldots, L\}$. For $(a, b, c) \in \mathcal{N}_\ell^3$, let*

$$\begin{aligned} w_{abc} := &C_3^{(\ell)}((a, b, c), :) \\ &- \frac{1}{v}\left( C_2^{(\ell)}((a, b), :) + C_2^{(\ell)}((b, c), :) \right) \in \mathbb{R}^v. \quad (7) \end{aligned}$$

*Let $(c_z)_{z=1}^v$ denote the normalized cluster centroids obtained via INFERBINARY$(C_2^{(\ell)})$, and let $\mathcal{T}_\ell(z)$ be the set*

**Algorithm 3** INFERTERNARY($C_2^{(\ell)}, C_3^{(\ell)}, \{c_z^{(\ell)}\}_{z=1}^v, \tau_3$)

**Input:** root-to-pair and -triple covariance tensors $C_2^{(\ell)}$, $C_3^{(\ell)}$, binary centroids $\{c_z^{(\ell)}\}_{z=1}^v$

**Output:** ternary rule set $\mathcal{R}_3^{(\ell)}$

**Pair-corrected triple slices:** For each triple $(a, b, c) \in [v]^3$, define

$$
\begin{aligned}
w_{abc} := {} & C_3^{(\ell)}((a,b,c),:) - \frac{1}{v}C_2^{(\ell)}((a,b),:) \\
& - \frac{1}{v}C_2^{(\ell)}((b,c),:) \in \mathbb{R}^v
\end{aligned} \tag{5}
$$

**Alignment:** For each triple $(a, b, c)$ and centroid $c_z^{(\ell)}$, compute the alignment score

$$
A_z(a,b,c) := \frac{\langle w_{abc}, c_z^{(\ell)} \rangle}{\|w_{abc}\| \|c_z^{(\ell)}\|} \tag{6}
$$

**Ternary rules:** If $\max_z A_z(a,b,c)$ exceeds the threshold $\tau_3$, assign $(a,b,c)$ to the maximizing cluster $z^\star$ and add the rule $(z^\star \to abc)$ to $\mathcal{R}_3^{(\ell)}$

*of valid level-$\ell$ ternary siblings generated by level-$(\ell-1)$ nonterminal $z$. Define*

$$
\tau_3 := \gamma\, v^{-3} \sum_{a,b,c} \max_z \langle w_{abc}, c_z \rangle, \tag{8}
$$

*with $\gamma \in (0,1)$. Then, with high probability over the random grammar and for all sufficiently large $v$, one has:*

$$
\langle w_{abc}, c_z \rangle > \tau_3 \quad \text{if and only if} \quad (a,b,c) \in \mathcal{T}_\ell(z).
$$

*Consequently, INFERTERNARY reconstructs the level-$\ell$ ternary rules within the same permutation of parent labels as INFERBINARY.*

**Nonterminal covariances.** Having identified the level-$L$ rules, we can build a detector for candidate nonterminals, $N_{i,\lambda}^{(L-1)}(a) = 1$ if the terminals $(X_i, \ldots, X_{i+(\lambda-1)})$ can be produced by nonterminal $a$, and 0 othwerwise. Pair and triple detectors can also be built by multiplication. Notice, however, that not all candidates correspond to true nonterminals, as there is a probability $\approx f$ that a sequence of adjacent terminals that are *not* binary/ternary siblings is compatible with some rule—we refer to these cases as *spurious*. All indicators introduced can be expressed as the sum of a true and a spurious indicator. The covariances between the root and pairs or triples of candidate nonterminals decompose into true and spurious contributions, too. Nevertheless, as proved in subsection C.4, spurious contributions are negligible in the asymptotic vocabulary size limit, allowing for the iteration of the inference steps.

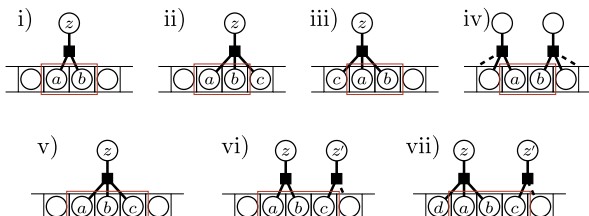

*Figure 4.* Decomposition of root-to-pair and root-to-triple covariances by the law of total covariance according to the local topology. Diagrams i) to iv) contribute to the root-to-pair covariance, while diagrams v), vi) and vii) to the root-to-triple one.

So far, we have proved the correctness of our iterative algorithm 1 for rule inference with exact moments. We next address how the empirical versions of the moments converge to their population limit, allowing us to obtain the sample complexities for moment estimation. Underlying this is a signal-to-noise argument: intuitively, the sample complexity corresponds to the number of data (sentences) at which the task-relevant correlations become detectable with respect to the finite sampling noise.

### 3.1. Sample Complexity of Neural Networks

**Proposition 3.3** (Sample complexity of root-to-pair and root-to-triple covariance estimation). *Consider data generated by a varying-tree RHM in the asymptotic vocabulary size limit $v \to \infty$ with $m_2 = f_2 v$ and $m_3 = f_3 v^2$. Fix a level $\ell \in \{1, \ldots, L\}$. Define $u_{ab} \in \mathbb{R}^v$ as in Eq. 143 (rows of the root-to-pair covariance) and $w_{abc} \in \mathbb{R}^v$ as in Eq. 143 (rows of the root-to-triple covariance). Let $\widehat{u}_{ab}$ and $\widehat{w}_{abc}$ be the empirical rows built from $P$ i.i.d. training samples, and define the errors*

$$
E_{2,ab} := \|\widehat{u}_{ab} - u_{ab}\|_2, \quad E_{3,abc} := \|\widehat{w}_{abc} - w_{abc}\|_2. \tag{9}
$$

*For any $\delta \in (0,1)$, there is a constant $c_\delta$ such that, if*

$$
P > c_\delta\, P_{2,\ell} \quad \text{with} \quad P_{2,\ell} := (p_2^2/2)^{1-\ell}\, v m_2 m_2^{\ell-1}, \tag{10}
$$

*then, with probability larger than $1 - \delta$, $E_{2,ab} < \|u_{ab}\|$ for any of the $v m_2$ valid pairs in $\mathcal{B}_\ell$. Analogously, if*

$$
P > c_\delta\, P_{3,\ell} \quad \text{with} \quad P_{3,\ell} := (p_2^2/2)^{1-\ell}\, v m_3 m_2^{\ell-1}, \tag{11}
$$

*then, with probability larger than $1 - \delta$, $E_{3,ab} < \|w_{ab}\|$ for any of the $v m_3$ valid triples in $\mathcal{T}_\ell$.*

The proposition follows from a simple vector Bernstein bound on the errors,

$$
E_s \le \gamma_s \left( \sqrt{\frac{\log 2/\delta}{v m_s P}} + \frac{\log 2/\delta}{P} \right) \tag{12}
$$

with probability $\geq 1 - \delta$ for some constant $\gamma_s$ and $s = 2, 3$, and from the asymptotic row norms via C.3,

$$\|u_{ab}\|_2 \to \frac{\|r_{z(a,b)}^{(\ell-1)}\|_2}{m_2} \to \frac{1}{m_2 v}\sqrt{\frac{(p_2^2/2)^{\ell-1}}{m_2^{\ell-1}}}, \qquad (13)$$

$$\|w_{abc}\|_2 \to \frac{\|r_{z(a,b,c)}^{(\ell-1)}\|_2}{m_3} \to \frac{1}{m_3 v}\sqrt{\frac{(p_2^2/2)^{\ell-1}}{m_2^{\ell-1}}}.$$

We now argue, and later confirm empirically, that deep neural networks solve the root classification task with a similar mechanism to 1, thus present the following conjecture for their sample complexity.

**Conjecture 3.4** (Sample complexity of deep networks on the root classification task). *Let*

$$P_{\ell,s} \asymp (p_2^2/2)^{1-\ell}\, v\, m_s\, m_2^{\ell-1} \quad for \quad s = 2, 3 \qquad (14)$$

*be the sample complexity scales required to estimate the rows of the level-$\ell \in \{1, \ldots, L\}$ root-to-pair and root-to-triple covariance matrices with sufficient accuracy for rule recovery. Then, the sample complexity of a properly trained deep neural network of depth $\geq L$ trained on root classification is*

$$P_{\mathrm{NN}}^{\star} \asymp \max_{\ell \in \{1,\ldots,L\},\, s \in \{2,3\}} P_{\ell,s}. \qquad (15)$$

*In particular, in the asymptotic limit $v \to \infty$ with $m_s = f_s v^s$ for $s = 2, 3$, where $m_3 \gg m_2$,*

$$P_{\mathrm{NN}}^{\star} \asymp (p_2^2/2)^{1-L}\, v\, m_3\, m_2^{L-1}. \qquad (16)$$

The key property required of the network is depth $\geq L$, which is essential to hierarchically exploit low-order statistics of the data and achieve polynomial sample complexity. We further note that the scaling (16) has a simple interpretation as stemming from the "easiest" branches: since $m_3 \gg m_2$, correlations are strongest when the number of ternary productions is minimal (i.e. all binary until the last layer).

## 4. Empirical measurements of $P^*$

The learning algorithm we proposed is inspired by CNNs: (i) for the first iteration at least, the clustering based on correlations can be achieved with one step of gradient descent (Malach & Shalev-Shwartz, 2018; Cagnetta et al., 2024) and (ii) acting on the correlation between substrings and task across the entire sentence is automatic with weight sharing. We thus first test our predictions on sample complexity in CNNs, before showing that they hold more generally, including in transformers. Our code is publicly available at https://github.com/jackparley/learn_to_parse.

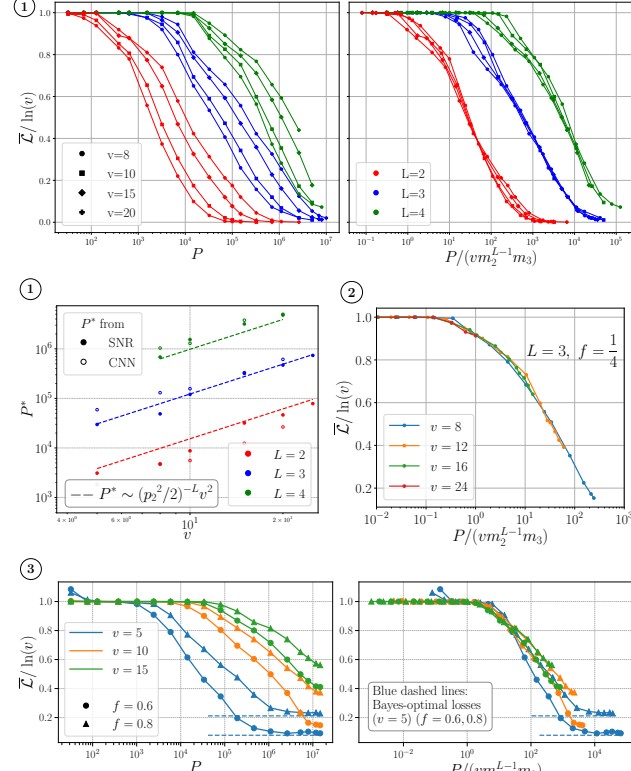

*Figure 5.* Experiments on CNNs, testing our theoretical prediction (16). We show separately the three cases of ambiguity indicated in Fig. 2b: **(1)** low ambiguity ($f \ll 1$), where rescaling by the predicted $P^* \sim v^2$ yields a collapse of the learning curves, and the extracted $P^*(v, L)$ agrees with both theory and empirical signal-to-noise estimates; **(2)** intermediate ambiguity ($f = 1/4$) and $L = 3$, where the predicted scaling $P^* \sim v^5$ is confirmed by a similar collapse; **(3)** high ambiguity ($f = 0.6, 0.8$), where the loss saturates at the Bayes-optimal value due to finite class entropy, while remaining consistent with the predicted scaling.

### 4.1. Tests on CNNs

We consider hierarchical CNNs of depth $L$, with filters of size 4 and stride 2 such that both rule types are covered (see App. D). These are trained by standard SGD on the classification cross-entropy loss, using $P$ sentence-label training samples $\{\boldsymbol{x}_\mu, y_\mu\}_{\mu=1}^{P}$ from our varying tree dataset. For clarity, we will test our predictions separately in the three representative regimes of ambiguity marked in Fig. 2: **(1)** $f \ll 1$ (low ambiguity), **(2)** intermediate $f$ (finite local but no global ambiguity), **(3)** large $f$ (finite global ambiguity).

**(1)** To test scaling in the low ambiguity regime, we set $f = 1/v$ and consider increasing $v$ so that $P^* \sim v\, m_2^{L-1} m_3 \sim v^2$. Fig. 5 (top left) shows the raw learning curves. Rescaling the training set size $P$ by the predicted $P^*$ leads to a remarkable *collapse* shown in the right panel. We additionally test the depth dependence: as $L$ increases, the "easy branches" occur with a smaller probability, lead-

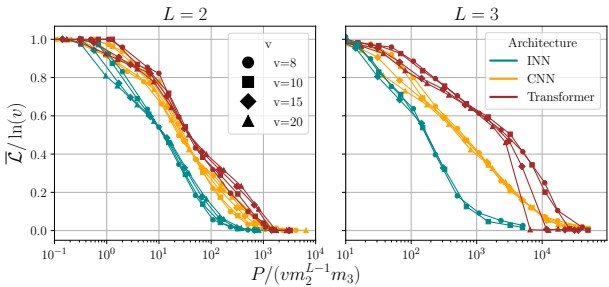

*Figure 6.* Average normalized test loss $\overline{\mathcal{L}}$ vs. rescaled training samples $P$, comparing the sample efficiency of the INN, CNN and transformer for $L=2$ and $L=3$. We consider the setting of increasing $v$ with $f=1/v$, so that $P^*{\sim}v\, m_2^{L-1}\, m_3{\sim}v^2$, a scaling which is found to be consistent with all three architectures.

ing to a factor $(p_2^2/2)^{1-L}$ in Eq. 16. To test this prediction, we extract $P^*(v, L)$ values from the raw learning curves by setting a threshold, and plot the $v-$dependence in the middle left panel. We additionally compare these to predictions from an empirical signal-to-noise ratio measured directly from the data (see App. F for details), finding excellent agreement.

**(2)** We next consider (middle right) intermediate $f = 1/4$ and increasing $v$, with $L = 3$. Eq.16 then leads to $P^*{\sim}v^5$. Again, we find an excellent collapse by rescaling the training set size $P$ accordingly.

**(3)** Finally, we consider (bottom) high ambiguity $f = 0.6$, $0.8$ and $L = 2$. This corresponds to a regime with finite class entropy, which acts as a lower bound on the cross-entropy loss of any classifier. In addition to testing the predicted scaling again, we confirm that the loss of deep networks converges to the analytically predicted Bayes-optimal loss (dashed lines).

### 4.2. Other architectures

We now consider (a) a vanilla encoder-only transformer, and (b) an alternative hierarchical convolutional network, which we refer to as Inside Neural Network (INN) (see App. D for architecture details). The INN assumes prior knowledge of the branching ratios, and implements a separate binary and ternary filter at each layer to create feature maps which precisely mirror the parsing charts in (2.4). This prior introduces a strong bias towards processing sentences according to their underlying tree topology (similar ideas have been applied to improve sample efficiency of syntactic generalization (Nandi et al., 2025)).

Fig. 6(a,b) displays training curves for $L=2$ and $L=3$ respectively. In both cases, sample size was rescaled by our prediction for $P^*$, and shows an excellent collapse for all architectures considered- illustrating the broad applicability of our theory. Although sample complexity scales simi-

larly, in terms of pre-factors the INN displays (for $L = 3$) a massive gain in sample efficiency with respect to the transformer, while the CNN is intermediary, as expected from the inductive bias of these architectures.

## 5. Conclusion

We have provided a theory for how deep nets learn to parse based on local correlations, leading to a prediction for sample complexity confirmed in a variety of architectures. This approach can explain how the representation of abstract semantic notions invariant to syntactic details of the sentence - modeled here as latent variables that can generate many possible visible substrings - can be learned in deep nets.

Note that although we have focused on a supervised task, it would be interesting to extend our results to to next-token prediction. In that case, the key signals are the correlation between the last token and pairs or triples of tokens further away in the sentence. As the training set-size increases, we expect that longer-range correlations can be resolved, and latent variables of higher level can be constructed (Cagnetta & Wyart, 2024; Cagnetta et al., 2025). Extending our sample complexity treatment to that case would be very interesting.

Our theory of learning is based on a proposed algorithm, and not on a mathematical description of gradient descent (GD) that is well beyond the state of the field for deep architectures. That said, our approach puts forward certain mechanisms at play during training, and it would be interesting to establish how they can be realized with GD. That includes: (i) the disambiguating step of the algorithm that clusters triplet strings together after removing the signal of spurious pairs. Although it is well-known that clustering based on correlations can be achieved with one GD step, the possibility that the new procedure may be achievable in a few steps of GD is an interesting question for the future. (ii) Maintaining track of the spans of latent variables, and building strings of adjacent ones (as needed to iterate our learning algorithm across depth). On this front, the question of how CNNs build adjacency from their internal representations is less clear. We expect that again, clustering by correlation is at play, since non-adjacent strings of latent variables will display very weak correlations with the task in comparison to adjacent ones.

## Limitations

One limitation is the assumption of uniform depth. Important works for the future include extending to (i) syntax where some production rules can be used recursively, leading to non-uniform depth (Schulz et al., 2025) and (ii) more structured syntax such as attribute (Knuth, 1968) or dependency grammars that LLMs encode in intriguing ways (Diego-Simón et al., 2024).

## Impact Statement

This work advances theoretical understanding of the interaction between data structure and deep learning. We do not foresee ethical concerns or negative societal impacts.

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

# A. Model Properties: top-down master equations

We give here additional details of our synthetic grammar, starting with the distribution of sentence lengths. To simplify the notation, we consider directly $p_2 = p_3 = 1/2$, i.e. equally probable branchings, and $f_2 = f_3 = f$, choices made for all the numerics shown in the main text.

## A.1. Distribution of sentence lengths

We denote the distribution of sentence lengths $d$ at depth $L$ by $P(d, L)$. This follows the following top-down master equation, initialized at $P(d, L = 0) = \delta_{d,1}$:

$$P(d, L + 1) = \sum_{d' = \lceil d/3 \rceil}^{\lfloor d/2 \rfloor} \frac{d'!}{(3d' - d)!(d - 2d')!} \left(\frac{1}{2}\right)^{d'} P(d', L) \tag{17}$$

where the combinatorial factor accounts for the number of ways of choosing binary/ternary branchings for the sequence of $d'$ parent nodes given the target length $d$, and the $1/2$ reflects equally probable branchings.

To understand how the fluctuations of this multiplicative process scale, we can reason as follows. The sentence length at the next depth $L + 1$ is given by the product $d_{L+1} = r_L \times d_L$, so that, at depth $L$, $\log(d_L) = \sum_{L'=0}^{L-1} \log(r_{L'})$. Here, $r_L$ is a random variable, which will clearly have mean $\langle r_L \rangle = 2.5$. In general, its value will be set by a weighted average of 2 and 3, where the weights correspond to the fraction of $d_L$ symbols which give rise respectively to 2 and 3 descendants. That is

$$r_L = \frac{2n(r = 2) + 3(d_L - n(r = 2))}{d_L} \tag{18}$$

where $n(r = 2) \sim \text{Binom}(d_L, 1/2)$. For large $d_L$, we can approximate this as a Gaussian $n(r = 2) \sim \mathcal{N}(d_L/2, d_L/4)$. $r_L$ is simply a linear combination of Gaussians, and one can therefore calculate its mean and variance. $\langle r_L \rangle = 2.5$, while $\sigma_r^2 = 13/(4d_L)$.

We may equivalently consider the evolution of the log-sentence lengths, $\log(d_{L+1}) = \log(r_L) + \log(d_L)$. The mean and variance of $\log(r_L)$ can be estimated using the Delta method: the mean of $\log(r_L)$ is $\log(2.5)$ up to a $1/d_L \sim 1/2.5^L$ correction, and, more importantly, its variance goes as $13/(2.5^2 \, 4d_L) \sim 1/d_L$.

Therefore, for $L > 3$ we can neglect the fluctuations of $r_L$ around $\log(2.5)$, and the distribution is just translated by a fixed amount from one layer to the next (Fig. 7). Equivalently, in probability space (Fig. 8), we find convergence to a scaling function, once we rescale to $(d - 2.5^L)/2.5^L$ The shape of this fixed distribution remains practically identical beyond $L = 3$, which is in turn just a slightly smoothed version of the distribution at $L = 2$.

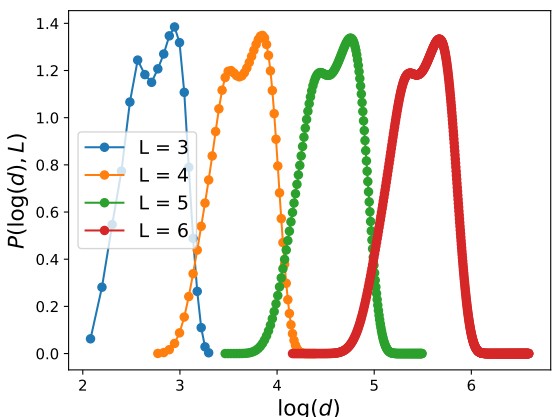

*Figure 7.* In log-space, the multiplicative process converges to a constant translation by $\log(2.5)$ beyond $L = 3$.

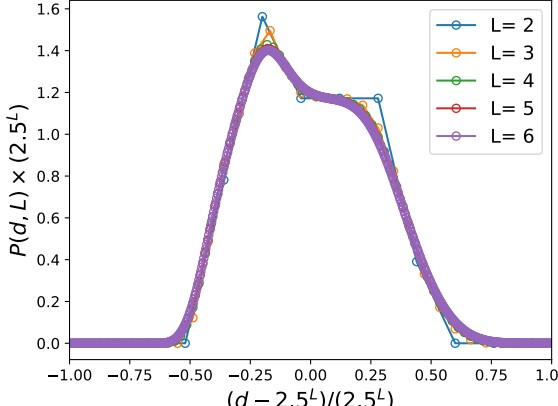

*Figure 8.* Scaling function in probability space. Asymptotically, this corresponds to a $\times 2.5$ dilation of a fixed distribution (in practice this holds already for $L > 3$).

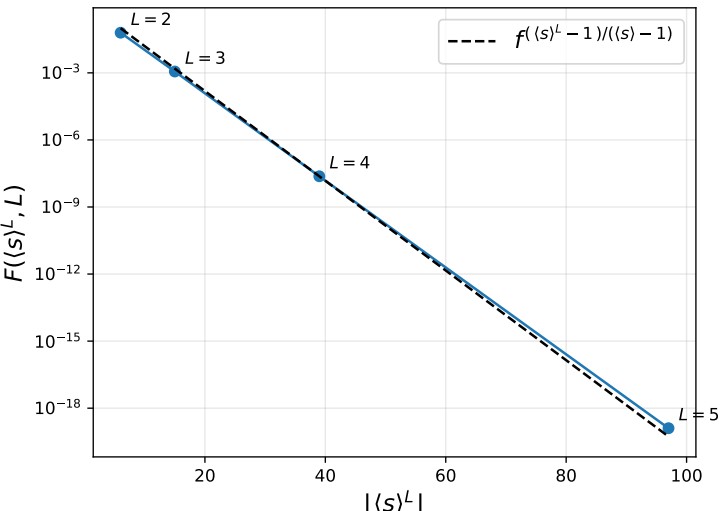

*Figure 9.* Numerical implementation of (Eq. 20) to obtain the fraction of grammatical data, $F(d, L)$, with $f=1/2$ (the $f < 1$ value can be chosen arbitrarily). Evaluating at the typical sentence length, we confirm the expected scaling $\sim f^{(\langle s \rangle^L - 1)/(\langle s \rangle - 1)}$.

## A.2. Number of tree topologies and fraction of grammatical data

We turn next to two additional important quantities, the number of topologically distinct trees for a sentence length $d$ at depth $L$, and the fraction of generated data with respect to the total possible sentences. The master equation for the first reads

$$N_t(d, L+1) = \sum_{d'=\lceil d/3 \rceil}^{\lfloor d/2 \rfloor} \frac{d'!}{(3d'-d)!(d-2d')!} N_t(d', L) \tag{19}$$

initialized as $N_t(d, L=1) = \delta_{d,2} + \delta_{d,3}$. By applying Stirling's formula to the factorials, it is simple to show that this scales asymptotically as $N_t(\langle s \rangle^L, L) \sim 2^{(\langle s \rangle^L - 1)/(\langle s \rangle - 1)}$.

We turn next to the fraction $F$ of grammatical sentences with respect to the total possible $v^d$ given a tree topology, for a given sentence length $d$ and depth $L$, averaged over the possible tree topologies compatible with $(d, L)$. It is easy to show that for a fixed topology (Cagnetta et al., 2024)), a factor $f$ is accumulated for each branching made within the tree. With a fixed branching ratio $s$, this leads to the result that at depth $L$, $F = f^{(s^L - 1)/(s-1)}$, where $(s^L - 1)/(s - 1)$ is the number of internal nodes of the tree. In the varying tree case, we are interested in the equivalent quantity, but which takes into account fluctuations of the tree topology once we fix a value $(d, L)$. This will be given by

$$F(d, L+1) = \frac{\sum_{d'=\lceil d/3 \rceil}^{\lfloor d/2 \rfloor} \frac{d'!}{(3d'-d)!(d-2d')!} \left(\frac{1}{2}\right)^{d'} f^{d'} F(d', L)}{\sum_{d'=2^L}^{3^L} \frac{d'!}{(3d'-d)!(d-2d')!} \left(\frac{1}{2}\right)^{d'}} \tag{20}$$

with $F(d, L=1) = f(\delta_{d,2} + \delta_{d,3})$. The meaning of this formula is to take the fraction $F$ values at the previous layer for each $d'$, and multiply this by $f^{d'}$ (recall that $f$ must be raised to the number of branchings). Finally, this has to be weighed by the probability of having an incoming branching from $(d', L)$, leading to the normalization term in the denominator. We have shown in A.1 that after the first layers fluctuations around the multiplicative factor $\langle s \rangle$ can be neglected: one therefore expects also here that, this fraction of data given a tree topology, scales, for typical sentences, asymptotically with the typical number of internal nodes as $F(\langle s \rangle^L, L) \sim f^{(\langle s \rangle^L - 1)/(\langle s \rangle - 1)}$. We confirm this in Fig. 9, where we implement numerically Eq. 20.

## A.3. Top-down approach to the class entropy

We present here a top-down derivation of the class entropy, which gives the exact prediction for $L = 2$ shown in Fig. 2. For the expected class entropy $\mathbb{E}_{\mathcal{G}} \left[ \mathbb{E}_{\boldsymbol{X} \sim P_{\mathcal{G}}} \left[ H(Y \mid \boldsymbol{X}) \right] \right]$ given a sentence $\boldsymbol{X}$, we can evaluate this by decomposing firstly into

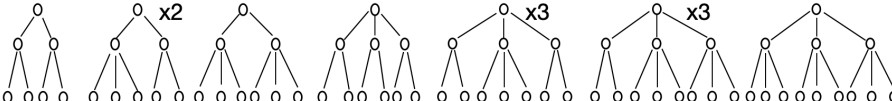

*Figure 10.* Sketch showing the 12 distinct tree topologies at $L = 2$. Note that for $d = 5$ there are two permutations of the tree, and 3 permutations for $d = 7, 8$. The probability of each individual topology is simply given by $p_2 = p_3 = 1/2$ raised to the number of branchings.

sentence lengths (we omit the subscripts on the expectation to ease notation)

$$\mathbb{E}[H_L(Y|\boldsymbol{X})] = \sum_{d'} \mathbb{E}[H_L(Y|\boldsymbol{X}; D = d')] P(d', L) \tag{21}$$

For $L = 2$, one can then derive the expression (shown in Fig. 2b of the main text)

$$\mathbb{E}[H_2(Y|\boldsymbol{X})](f) = \frac{1}{4} f^3 H(1/2, 1/2) + \frac{3}{16} \left( \frac{2}{3} f^3 + \frac{1}{3} f^4 \right) H(1/3, 2/3) + \frac{3}{8} \left( 2f^4(1 - f^4) H(1/2, 1/2) + f^8 H(1/3, 1/3, 1/3) \right) \tag{22}$$

where we use the notation $H(\{p_i\}) = -\sum_i p_i \ln(p_i)$. For each value of $d' = 4, \dots, 9$ we use the value of $F(d', L{=}2)$ as the probability of success of finding an additional parse tree deriving the sentence, among the $N_t(d, L{=}2) - 1$ possible topologies (the 12 topologies at $L = 2$ are sketched in Fig. 10), weighing this by the probability $P(d', L)$. If an additional tree is found, the entropy is then given by the set of probabilities $\{p_i\}$ of the topologies. For $d = 7, 8$ (last term), there are $N_t = 3$ topologies, so that either one more or two more can be found. Note furthermore that for $d = 6$ (second term), the $2{-}2{-}2$ topology is a factor 2 less probable than the $3{-}3$ topology, due to the extra branching. Eq. 23 is valid for $v \gg 1$, as we have assumed that each additional parse tree corresponds to a different label; accounting for coinciding labels leads to corrections of $\mathcal{O}(1/v)$. We stress here that, as pointed out in the main text, the class entropy is always bounded by the number of tree topologies. At $L = 2$, for example, in the maximally ambiguous case $f = 1$ Eq. 23 takes the finite value

$$\mathbb{E}[H_2(Y|\boldsymbol{X})](f = 1) = \frac{1}{4} H(1/2, 1/2) + \frac{3}{16} H(1/3, 2/3) + \frac{3}{8} H(1/3, 1/3, 1/3) \tag{23}$$

which is $\ll \ln(v)$ for $v \to \infty$.

Extending the reasoning behind (23) to deeper grammars, $L > 2$, fails, for the following reason. In deriving (23) we implicitly make an independence assumption, namely that the probability of finding an additional parse tree is independent of the ground truth (or planted) tree. As discussed in the main text (see e.g. Fig. 2a), however, spurious latents can be formed from latents on the ground truth tree, so that the later acts as an additional source leading potentially to additional parse trees. The reason (23) nonetheless holds for $L = 2$ are geometric constraints for forming a full span: for $L = 2$, additional class labels can only be built using purely spurious latents at level-1. For example for $d = 5$, where there are two possible topologies $3 - 2$ or $2 - 3$, if the ground truth is $3 - 2$ and a spurious latent is generated for the pair at the start of the sentence, this spurious latent cannot build a complete span together with the original true latent of the second pair, as this would only sum up to a total span length of 4.

## B. Inside algorithm and bottom-up approach to the crossover

### B.1. Tensorial form of the inside algorithm

We provide here firstly the full form of the tensorial, layered inside algorithm. We recall that the teriminal-level inside tensor for each sentence is initialized as $M_{i,\lambda}^{(L)}[z] = \delta_{x_i, z}$. To build layer by layer the tensors at other levels, we follow

$$M_{i,\lambda}^{(l-1)}[z] = \sum_{a,b=1}^{v} \sum_{q}^{(\mathcal{I}_l)} R_{z \to (a,b)}^{(l)} M_{i,q}^{(l)}[a] M_{i+q,\lambda-q}^{(l)}[b]$$

$$+ \sum_{a,b,c=1}^{v} \sum_{q,r}^{(\mathcal{I}_l)} R_{z \to (a,b,c)}^{(l)} M_{i,q}^{(l)}[a] M_{i+q,r}^{(l)}[b] M_{i+q+r,\lambda-q-r}^{(l)}[c] \tag{24}$$

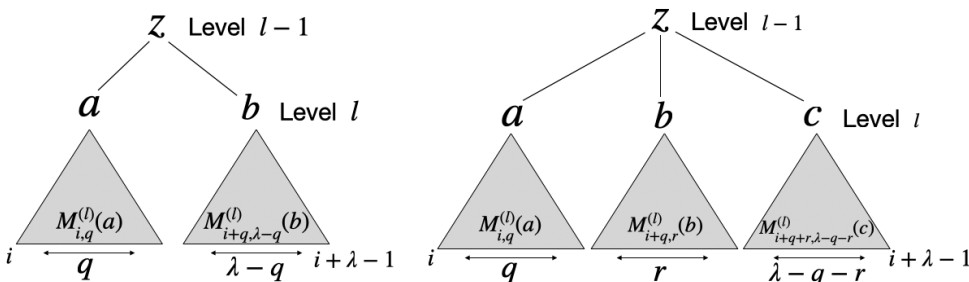

*Figure 11.* Sketch illustrating the notion of binary (left) and ternary (right) splits in the inside recursion, corresponding respectively to the first and second term in (24).

where we recall the level-$l$ rule probability tensors are equal to $R^{(l)}_{z \to (a,b)} = 1/(2m_2)$ and $R^{(l)}_{z \to (a,b,c)} = 1/(2m_3)$ if such rules are grammatical. We define the integer interval $\mathcal{I}_l = [2^{L-l}, 3^{L-l}]$: the span length $\lambda$, pertaining to level-$(l-1)$, satisfies $\lambda \in I_{l-1}$, i.e. the integer interval $[2^{L-\ell+1}, 3^{L-\ell+1}]$. The summation over splits (illustrated in the sketch of Fig. 11) instead is restricted in the following manner, which we denote by the superscript $(I_l)$:

$$\sum_q^{(\mathcal{I}_l)} f(q) := \sum_{\substack{q \in \mathbb{Z} \\ 2^{L-l} \le q \le 3^{L-l} \\ 2^{L-l} \le \lambda-q \le 3^{L-l}}} f(q) \tag{25}$$

$$\sum_{q,r}^{(\mathcal{I}_l)} g(q,r) := \sum_{\substack{q,r \in \mathbb{Z} \\ 2^{L-l} \le q,r \le 3^{L-l} \\ 2^{L-l} \le \lambda-q-r \le 3^{L-l}}} g(q,r) \tag{26}$$

For example, when building the level-$(L-2)$ tensor, for $\lambda = 6$ there is a single valid binary ($q = 3$) and a single valid ternary ($q = 2, r = 2$) split, corresponding respectively to the two ways of splitting a span of length 6 into segments of length 2 or 3, 3-3 or 2-2-2.

Filling in the whole inside tensor following a naive implementation of (24) quickly runs into computational limits, given that the loop over start positions, span lengths $\lambda$ and splits leads to an $\mathcal{O}(d^3)$ scaling for binary rules, and $\mathcal{O}(d^4)$ for ternary. From a computational standpoint, the layerwise structure is indeed essential, as it enables efficient parallelization. The $\mathcal{O}(d^4)$ time complexity scaling would make a naive implementation unfeasible already for $L > 3$; by parallelizing, we are able to compute numerically up to $L = 6$ (where $d \sim \langle s \rangle^L \sim 244$ and (!) $N_t \sim 10^{38}$) the expected class entropies $\mathbb{E}_{\boldsymbol{X}, \mathcal{G}}[H_L(Y|\boldsymbol{X})]$.

To achieve this, we tensorize (24) (for fixed $\lambda$) and parallelize the computation to fill in the next-level inside tensor. We can do this by defining a $v \times v \times v$ size tensor for storing binary rule probabilities $R^{(l)}_{z \to (a,b)}$, and a $v \times v \times v \times v$ for ternary $R^{(l)}_{z \to (a,b,c)}$. We further precompute the set of valid binary $\{q\}$ and ternary $\{(q,r)\}$ splits for each $\lambda$, the total number of which we can denote by $N_{\text{bin}}(\lambda)$ and $N_{\text{ter}}(\lambda)$ respectively. These are expected to grow as $N_{\text{bin}}(\lambda) \sim \lambda$ and $N_{\text{ter}}(\lambda) \sim \lambda^2$. We confirm this in Fig. 12. Defining the reversed level index $\tilde{l} = L - l$, so that $\tilde{l} = 0 \dots L$ from tokens to root, we show the median of the number of splits required to fill in the level-$\tilde{l}$ inside tensor, across the span lengths $\lambda = 2^{\tilde{l}}, \dots 3^{\tilde{l}}$. The median is dominated by the maximum span length $\sim 3^{\tilde{l}}$.

The parallel computation involves (in the ternary case) building a tensor of shape $d \times N_{\text{ter}}(\lambda) \times v \times v \times v \times v$, which we sum over all but the first and third dimensions, leaving the $d \times v$ elements needed to fill in the entries of the entries of the inside tensor at fixed span length for $i = 1 \dots d$ and $z = 1 \dots v$. Due to memory constraints, for large $\tilde{l}$ it can be prohibitive to parallelize simultaneously over the $d \sim \langle s \rangle^L$ start positions and the $N_{\text{ter}} \sim 3^{2\tilde{l}}$ splits. As we illustrate in Fig. 12 for $L = 6$, beyond a certain $\tilde{l}$ ($\tilde{l} = 4$ in this case) $N_{\text{ter}}$ surpasses $d$, so that we in fact maintain the loop over start positions and only parallelize across splits. We note finally that for large $L > 4$ we implement the inside algorithm (24) in log-probability space, to prevent underflow.

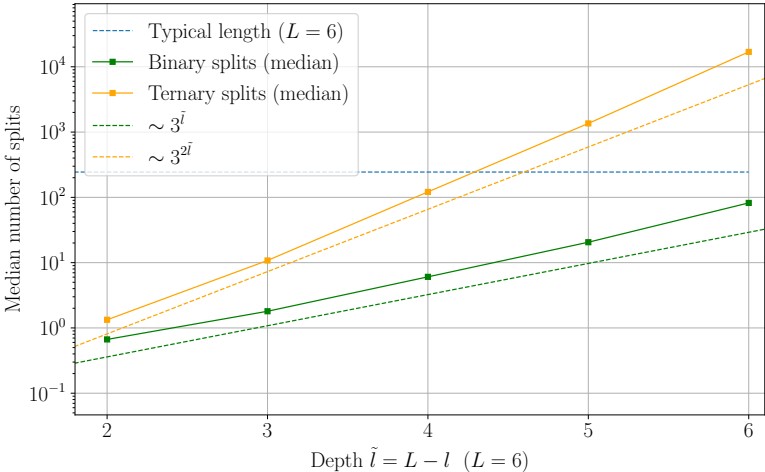

*Figure 12.* Median number of binary and ternary splits, required to build the inside tensor at depth $\tilde{l}$, across the span lengths $\lambda \in [2^{\tilde{l}}, 3^{\tilde{l}}]$. The medians are dominated by the tail, so that these grow as $\sim 3^{\tilde{l}}$ for the binary and $\sim 3^{2\tilde{l}}$ for the ternary case, as shown by dashed lines. For the example case of $L = 6$, $N_{\text{ter}}$ surpasses the sentence length $d \sim \langle s \rangle^L$ beyond $\tilde{l} = 4$.

### B.2. Qualitative account of the finite-$v$ crossover

To give a qualitative understanding of the class entropy crossover for finite $v$, we consider a "coarse-grained" form of the inside algorithm (24), based on two simplifications. Firstly, we neglect fluctuations across the input dimension $d$, as well as boundary effects, which is a safe assumption for large enough $L$ and hence $d$. Secondly, we get rid of the fine structure of the rules, which as shown in Sec. 3 is essential for learning, and instead simply keep track of how many candidate latents (both genuine and spurious) the inside algorithm builds on average across the vocabulary, as it proceeds up the hierarchy. More precisely, we consider the quantity

$$n_\lambda^{(l)} = \frac{1}{d} \sum_{i=1}^{d} \sum_{z=1}^{v} N_{i,\lambda}^{(l)}(z) \qquad (27)$$

defined from the Boolean inside tensors $N_{i,\lambda}^{(l)}$. It is important to note that, although we continue to work in the $v \to \infty$ limit, we *do not* scale (27) by $v$, and yet it is $\mathcal{O}(1)$. This is due to how the inside tensor is initialized. Indeed, the token-level inside tensor is initialized as $N_{i,\lambda=1}^{(L)}(z) = \delta_{x_i,z}$, so there is precisely one latent symbol across the vocabulary dimension, hence $n_{\lambda=1}^{(L)} = 1$.

We now consider the evolution of the average number of latents (27), both genuine and spurious, as the inside algorithm proceeds to build these layer by layer. This will be given by

$$n_\lambda^{(l-1)} = f_2 \sum_q^{(\mathcal{I}_l)} n_q^{(l)} n_{\lambda-q}^{(l)} + f_3 \sum_{q,r}^{(\mathcal{I}_l)} n_q^{(l)} n_r^{(l)} n_{\lambda-q-r}^{(l)} \qquad (28)$$

which simply reflects the fact that, the way we have defined the model, once a binary split $q$ is fixed, the pair of latents occupying that split correspond to a grammatical rule with probability $f_2$, and analogously in the ternary case. As in the main text, we consider for simplicity $f_2=f_3=f$.

In the case $f=1$, where all pairs/triples of candidate latents correspond to a grammatical rule, one expects that (28) will build as many latents as there are distinct tree topologies. In other words, at "depth" $\tilde{l} = 0, \ldots L$, i.e. $\tilde{l} = L - l$, and sentence length $d = \lambda$, we expect the following equality $n_{\lambda=d}^{\tilde{l}=L} = N_t(d, L)$. In Fig. 13 we confirm this equality numerically, by implementing Eq. 28 with $f=1$ and comparing to $N_t$ computed top-down from (19).

With $f < 1$ instead, each time we try to place a new latent we weigh this by the probability of the pair/triple of level-$l$ latents to actually constitute a grammatical rule. Without the inclusion of additional terms, (28) predicts a bifurcation around $f_c = 1/2$ (see Fig. 14). This value can be obtained also by a simple top-down argument: the number of trees grows

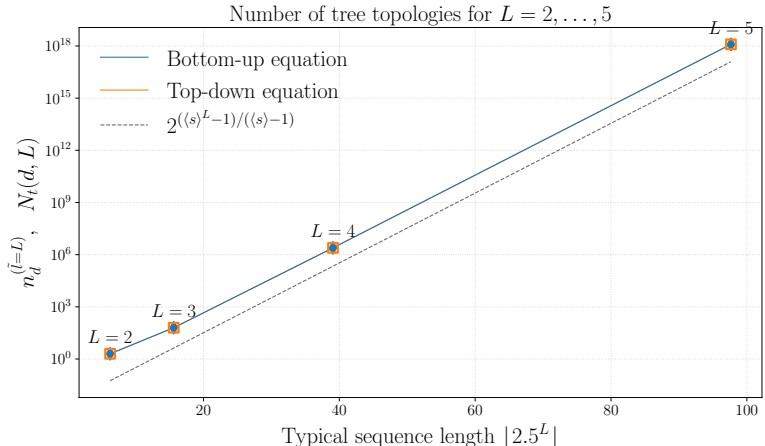

*Figure 13.* Comparison of $n_{\lambda=d}^{(\tilde{l}=L)}$, computed bottom-up from (28), and $N_t(d, L)$, obtained top-down from (19), evaluated at the typical sentence length $d = \lfloor \langle s \rangle^L \rfloor$, for $L = 2, 3, 4, 5$. These match exactly, and grow asymptotically as $\sim 2^{(\langle s \rangle^L - 1)/(\langle s \rangle - 1)}$ (dashed lines).

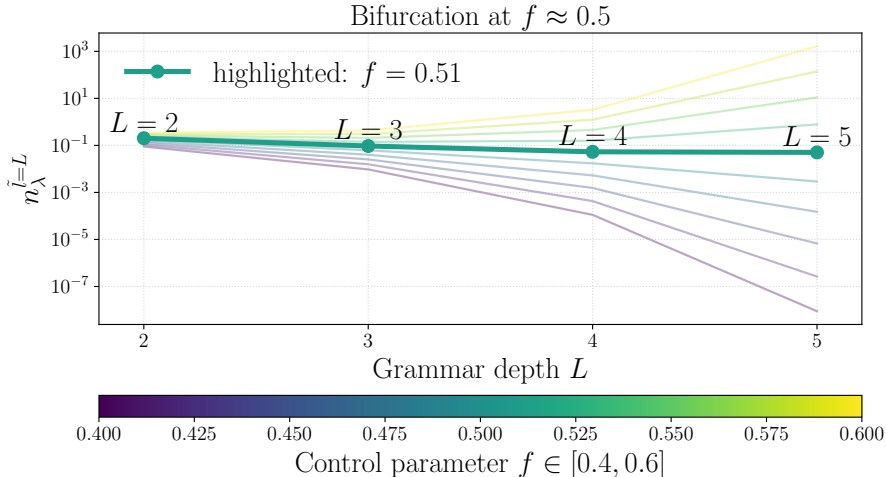

*Figure 14.* Numerical implementation of the iterative mapping (28) for $n_\lambda^{(l)}$, from the initial condition $n_{\lambda=1}^{(L)} = 1$ and evaluated for each depth at the typical sentence length $\lambda = \lfloor \langle s \rangle^L \rfloor$. A bifurcation is found around $f_c = 1/2$, which can also be recovered from a simple argument (see text).

exponentially as $N_t \sim 2^{(\langle s \rangle^L - 1)/(\langle s \rangle - 1)}$. As shown in App. A, the fraction $F(d, L)$ of grammatical data (out of the total $v^d$ which could be formed) given a typical parse tree at depth $L$ *decays* exponentially as $\sim f^{(\langle s \rangle^L - 1)/(\langle s \rangle - 1)}$, where we recall $f < 1$. Naively, one expects a transition at $FN_t \sim 1$ or $f_c = 1/2$: below $f_c$, the probability of finding two or more grammatical parse trees for the same sentence vanishes for large $L$, whereas for $f > f_c$ this probability approaches one.

Both arguments given above for the crossover are mean-field in nature, and neglect the additional effect of planted latents, i.e. present on the true derivation tree, which in particular do not follow the dynamics of Eq.(28). We stress furthermore, that although we believe these arguments to capture the crux of the finite-$v$ crossover shown in Fig. 2b, they do not constitute a theory for the class entropy per se. Finally, as pointed out in the main text, the class entropy is always bounded by the number of tree topologies, so that the crossover in the class entropy disappears completely for sufficiently large $v$.

## C. Iterative learning algorithm

In this section, we prove that, in the asymptotic vocabulary size limit ($v \to \infty$ with $m_s = f_s v^{s-1}$ for $s = 1, 2$), [1] infers the rules of a varying-tree RHM using root-to-pair and root-to-triple covariances. The proof is based on the analysis of the

various contributions to root-to-pair and root-to-triple covariances in the asymptotic limit. Many of these contributions are indeed negligible in the limit, which allows us to isolate the contributions of the individual rules and thus infer the generative model layer by layer. Throughout this section, we assume knowledge of the exact moments of the data distribution.

**Notation:** We denote a single instance of the Varying-Tree RHM, corresponding to a specific choice of production rules, with $\mathcal{G}$, and identify expectations with respect to the associated probability distribution with a subscript ($\mathbb{P}_{\mathcal{G}}\{.\}$, $\mathbb{E}_{\mathcal{G}}[.]$, $\mathrm{Cov}_{\mathcal{G}}[.]$). With no subscript, $\mathbb{P}\{.\}$, $\mathbb{E}[.]$, $\mathrm{Var}[.]$ and $\mathrm{Cov}[.]$ denote averages over draws of the random rules (i.e., the draw of the specific Varying-Tree RHM instance $\mathcal{G}$).

### C.1. Asymptotic statistics of Varying-Tree RHM data

This subsection collects useful results on the scales of various statistics for data generated by the Varying-Tree RHM in the asymptotic vocabulary size limit. We focus in particular on properties that are asymptotically independent of the specific realisation of the random production rules.

#### C.1.1. ASYMPTOTIC PARENT-TO-CHILD TRANSITION MATRICES

The parent-to-child transition matrices store, for each tree level $\ell$, branching factor $s = 2, 3$ and child index $i = 1, \ldots, s$, the probability of the $i$-th child given the parent along a specific branch of the tree,

$$\left(P_{i,s}^{(\ell)}\right)_{a,b} := \mathbb{P}_{\mathcal{G}}\left\{X_i^{(\ell)} = a \middle| X^{(\ell-1)} = b\right\}. \tag{29}$$

For each instance of the Varying-Tree RHM, the specific entries of such matrices depend on the production rules. Namely, $\left(P_{i,s}^{(\ell)}\right)_{a,b} = (m_s^{-1})\left(N_{i,s}^{(\ell)}\right)_{a,b}$, where $\left(N_{i,s}^{(\ell)}\right)_{a,b}$ denotes the number of occurrences of child symbol $a$ in position $i$ of the $s$-ary, level-$\ell$ rules with parent symbol $b$. For a fixed parent symbol $b$ (and level, arity and child position, omitted for clarity), the $(N_{a,b})_{a=1}^v$ are *multivariate hypergeometric* counts from a population of size $v^s$ (total number of possible $s$-tuples), split into $v$ groups of size $v^{s-1}$ (tuples with a given symbol in position $i$), with $m_s$ draws. This distribution has known factorial moments: with the expectation symbol denoting the average over draws of the random rules, and $(x)_n = x(x-1)\ldots(x-n+1)$,

$$\mathbb{E}\left[\prod_{a=1}^v (N_{a,b})_{n_a}\right] = \frac{(m_s)_{\sum_a n_a}}{(v^s)_{\sum_a n_a}} \prod_{a=1}^v \left(v^{s-1}\right)_{n_a}. \tag{30}$$

Without loss of generality, we write

$$P_{i,s}^{(\ell)} = J + \Delta_{i,s}^{(\ell)}, \tag{31}$$

where

$$J = \mathbb{E}[P_{i,s}^{(\ell)}] = \tfrac{1}{v}\mathbf{1}\mathbf{1}^\top, \quad \mathbb{E}[\Delta_{i,s}^{(\ell)}] = 0, \quad \mathrm{Var}\left[\Delta_{i,s}^{(\ell)}(a,b)\right] = \frac{1}{vm_s}\left(1 - \frac{1}{v}\right)\frac{v^s - m_s}{v^s - 1} \to \frac{1}{fv^s} + O\!\left(\frac{1}{v^{s+1}}\right), \tag{32}$$

The expectation of column $\ell_2$-norm and Frobenius norm of the $\Delta_{i,s}$' are (layer index $\ell$ dropped to ease notation)

$$\mathbb{E}\left[\|(\Delta_{i,s})_{:,b}\|^2\right] = v\,\mathrm{Var}\left[\Delta_{i,s}(a,b)\right] \to \frac{1}{fv^{s-1}}, \quad \mathbb{E}\left[\|\Delta_{i,s}\|_F^2\right] = v^2\,\mathrm{Var}\left[\Delta_{i,s}(a,b)\right] \to \frac{v^{2-s}}{f}. \tag{33}$$

To understand fluctuations about the expected norms, note that the $\Delta_{i,s}$ are centered and scaled hypergeometric counts, hence they are sub-Gaussian random variables with variance proxy $\mathrm{Var}\left[\Delta_{i,s}(a,b)\right]$. Hence, the 4-th moment is bounded by the squared variance proxy $\mathrm{Var}\left[\Delta_{i,s}(a,b)\right]^2 = O((m_s v)^{-2})$. Cross-moments such as $\mathrm{Cov}\left[\Delta_{i,s}(a,b)^2, \Delta_{i,s}(a',b)^2\right]$ obey the same bound via Cauchy-Schwarz $\mathrm{Cov}[X,Y] \le \sqrt{\mathrm{Var}[X]}\sqrt{\mathrm{Var}[Y]}$. Therefore,

$$\mathrm{Var}\left[\|(\Delta_s)_{:,b}\|^2\right] = \mathrm{Var}\left[\sum_a \Delta_s(a,b)^2\right] = \sum_a \mathrm{Var}\left[\Delta_s(a,b)^2\right] + \sum_{a'\neq a}\mathrm{Cov}\left[\Delta_s(a,b)^2, \Delta_s(a',b)^2\right] = O\left(\frac{1}{m_s^2}\right). \tag{34}$$

The Frobenius norm is the sum of the $\ell_2$-norms of the columns, hence we can bound its variance with the sum of the column norms' variances (the real variance is even smaller by the anti-correlations of the columns),

$$\text{Var}\left[\|\Delta_{i,s}\|_F^2\right] \leq \sum_v \text{Var}\left[\|(\Delta_{i,s})_{:,b}\|^2\right] = O\left(\frac{v}{m_s^2}\right). \tag{35}$$

This implies concentration since $\mathbb{E}\left[\|\Delta_{i,s}\|_F^2\right] \to v/m_s$, thus $\text{Var}\left[\|\Delta_{i,s}\|_F^2\right] / \left(\mathbb{E}\left[\|\Delta_{i,s}\|_F^2\right]\right)^2 = O(1/v)$. Sub-gaussianity of the entries with variance proxy $O(1/(m_s v))$ also gives the operator norm via the circular law for the distribution of the eigenvalues of a $v \times v$ random matrix with independent and identically distributed entries in the limit $v \to \infty$,

$$\boxed{\|\Delta_{i,s}\|_{\text{op}}^2 \to \frac{1}{m_s}, \quad \|\Delta_{i,s}\|_F^2 \to \frac{v}{m_s}.} \tag{36}$$

Note that the operator norm has the same scale as the $\ell_2$-norm of the columns, while it is smaller than the Frobenius norm by a factor of $v$: due to the symmetry for exchanging symbols, there is no preferred direction, thus the spectrum of the $\Delta_{i,s}$ is approximately flat.

**Average over child position.** Let us define the parent-to-child transition matrix averaged over arities and child position,

$$\widetilde{\Delta}^{(\ell)} := p_2\left(\frac{1}{2}\sum_{i=1}^2 \Delta_{i,2}^{(\ell)}\right) + p_3\left(\frac{1}{3}\sum_{i=1}^3 \Delta_{i,3}^{(\ell)}\right). \tag{37}$$

The different $\Delta$'s appearing in this formula are all independent, because binary and ternary rules are sampled independently, and there are no correlations between the symbols in different positions on the right-hand side of a production rule. In addition, the statistics of the transition matrices are independent of the child index. Therefore,

$$\text{Var}\left[\widetilde{\Delta}(a,b)\right] = \left(\frac{p_2}{2}\right)^2 2\text{Var}\left[\Delta_{i,2}(a,b)\right] + \left(\frac{p_3}{3}\right)^2 3\text{Var}\left[\Delta_{i,3}(a,b)\right] \to \frac{p_2^2}{2}\frac{1}{fv^2} + O\left(\frac{1}{v^3}\right). \tag{38}$$

All the conclusions on sub-gaussianity, Frobenius and operator norm apply to this case as well, with Eq. 38 variance proxy.

### C.1.2. ASYMPTOTIC MARGINALS OF TERMINALS AND NONTERMINALS

**Lemma C.1** (Asymptotic symbol marginals). *In the limit $v \to \infty$ with $m_s = fv^{s-1}$ for $s = 2, 3$ and finite $f \in (0, 1)$, the marginal probability of a single hidden (nonterminal) or visible (terminal) symbol converges to the uniform probability $1/v$.*

Let $\pi \in \Delta^{v-1}$ be the distribution of the root $X^{(0)}$. For a fixed terminal position $i$ and level $\ell = 1, \ldots, L$, let $s_\ell(i)$ denote the child-index/branching-factor pair of the $\ell$-th step along the root-to-leaf path. The marginal distribution of the $i$-th terminal reads

$$p_i^{(L)} = \left(\prod_{\ell=1}^L P_{s_\ell(i)}^{(\ell)}\right)\pi = \left(\prod_{\ell=1}^L \left(J + \Delta_{s_\ell(i)}^{(\ell)}\right)\right)\pi \in \mathbb{R}^v. \tag{39}$$

Expanding the product and using $J\Delta = 0$, $J\pi = u$ (uniform distribution $\mathbf{1}/v$), $Ju = u$,

$$p_i^{(L)} = u + \sum_{r=2}^L \left(\Delta_{s_L(i)}^{(L)} \cdots \Delta_{s_r(i)}^{(r)}\right)u + \left(\Delta_{s_L(i)}^{(L)} \cdots \Delta_{s_1(i)}^{(1)}\right)\pi. \tag{40}$$

**Ensemble mean (uniform).** By independence of the levels and $\mathbb{E}[P_{s_\ell}^{(\ell)}] = J$,

$$\mathbb{E}[p_i^{(L)}] = \left(\prod_{\ell=1}^L \mathbb{E}[P_{s_\ell(i)}^{(\ell)}]\right)\pi = J^L\pi = \frac{1}{v}\mathbf{1}. \tag{41}$$

**Concentration.** Write $p_i^{(\ell)}$ for the leaf distribution after $\ell$ steps on the path, with $p^{(0)} = \pi$ and $\delta_i^{(\ell)} = p_i^{(\ell)} - u$. Using $Jp = u$ for any distribution $p \in \Delta^{v-1}$ (second equality),

$$\delta_i^{(\ell)} = P_{s_\ell(i)}^{(\ell)} p_i^{(\ell-1)} - u = \left(P_{s_\ell(i)}^{(\ell)} - J\right) p_i^{(\ell-1)} = \Delta_{s_\ell(i)}^{(\ell)} u + \Delta_{s_\ell(i)}^{(\ell)} \delta_i^{(\ell-1)}. \tag{42}$$

Conditioning on $\delta^{(\ell-1)}$ and using independence of the levels, (path index $i$ omitted for simplicity)

$$\mathbb{E}\left[\|\delta^{(\ell)}\|_2^2 \,\big|\, \delta^{(\ell-1)}\right] = p^{(\ell-1)\top}\mathbb{E}\left[\Delta_s^{(\ell)\top}\Delta_s^{(\ell)}\right] p^{(\ell-1)} = \left(u + \delta^{(\ell-1)}\right)^\top G_s \left(u + \delta^{(\ell-1)}\right). \tag{43}$$

The statistics of the centred transition probabilities, $\Delta_{a,b} = P_{a,b} - \mathbb{E}[P_{a,b}]$, are invariant for permutations of the indices. As a result,

$$G_s := \mathbb{E}\left[\Delta_s^\top \Delta_s\right] := \alpha_s I_v + \beta_s \left(\mathbf{1}\mathbf{1}^\top - I_v\right), \tag{44}$$

where $\alpha_s$ denotes the diagonal entries and $\beta_s$ the off-diagonal ones,

$$\alpha_s := \sum_{a=1}^v \mathrm{Var}\big(P_s(a,c)\big) = \frac{1}{m_s}\left(1 - \frac{1}{v}\right)\frac{v^s - m_s}{v^s - 1} \qquad\qquad = \frac{1}{fv^{s-1}} + O\left(\frac{1}{v^s}\right), \tag{45}$$

$$\beta_s := \sum_a \mathrm{Cov}\big(P_s(a,c), P_S(a,c')\big) = -\frac{1}{v}\frac{v-1}{v^s - 1} \qquad\qquad = -\frac{1}{v^s} + O\left(\frac{1}{v^{s+1}}\right). \tag{46}$$

$G_s$ is a rank-1 perturbation of the identity, hence we can derive its spectrum by decomposing it along the rank-1 component $J = \mathbf{1}\mathbf{1}^\top/v$ and the orthogonal subspace spanned by $I_v - J$,

$$\begin{aligned} G_s &= (\alpha_s - \beta_s) I_v + v\beta_s J = (\alpha_s - \beta_s)(I_v - J) + (\alpha_s + (v-1)\beta_s)J \\ &= \lambda_s^{(\|)} J + \lambda_s^{(\perp)}(I_v - J), \end{aligned} \tag{47}$$

whence

$$\lambda_s^{(\perp)} = \alpha_s - \beta_s = \frac{1}{fv^{s-1}} + O\left(\frac{1}{v^s}\right), \quad \lambda_s^{(\|)} = \alpha_s + (v-1)\beta_s = \frac{1-f}{f\,v^{s-1}} + O\left(\frac{1}{v^s}\right). \tag{48}$$

Hence, after averaging over $\delta^{(\ell-1)}$ (and reinstating the dependence on the branching factor of the level $s$),

$$\mathbb{E}\left[\|\delta^{(\ell)}\|_2^2\right] = \lambda_{s,\perp}\mathbb{E}\left[\|\delta^{(\ell-1)}\|_2^2\right] + \frac{\lambda_{s,\|}}{v}, \tag{49}$$

which yields

$$\mathbb{E}\|\delta^{(L)}\|_2^2 = \left(\prod_{\ell=1}^L \lambda_{s_\ell}^{(\perp)}\right)\|\pi - u\|_2^2 + \frac{1}{v}\sum_{r=1}^L \lambda_{s_r}^{(\|)} \prod_{\ell=r+1}^L \lambda_{s_\ell}^{(\perp)}. \tag{50}$$

As $v \to \infty$ for fixed $f$ ($m_s = fv^{s-1}$),

$$\lambda_s^{(\perp)} = \frac{1}{fv^{s-1}} + O(v^{-s}), \quad \lambda_s^{(\|)} = \frac{1-f}{f}v^{-(s-1)} + O(v^{-s}) \tag{51}$$

$$\Rightarrow S = \sum_{\ell=1}^L (s_\ell - 1): \quad \prod_{\ell=1}^L \lambda_{s_\ell}^{(\perp)} = \frac{1}{f^L v^S}\left(1 + O(v^{-1})\right), \tag{52}$$

$$\Rightarrow \mathbb{E}\|\delta^{(L)}\|_2^2 = \frac{\|\pi - u\|_2^2}{f^L v^S} + \frac{1-f}{f}v^{-s_L} + O\left(v^{-\min\{S+1,\, s_L+1\}}\right). \tag{53}$$

When the root prior is uniform, fluctuations are controlled by the branching factor of the last level $s_L$: $v^{-2}$ or $v^{-3}$. By invariance of the rules' statistics for permutations of the indices,

$$\mathbb{E}\left[\left(\left(p_i^{(L)}\right)_a - \tfrac{1}{v}\right)^2\right] = \frac{\mathbb{E}\left[\|\delta_i^{(L)}\|_2^2\right]}{v} \to \frac{1-f}{f}\frac{1}{v^{s_L+1}}. \tag{54}$$

The variance decays faster than the squared mean $v^{-2}$ with $v$, independently of the branching factor. Therefore, the marginals concentrate around the uniform distribution. Considering a random leaf instead of a well-defined path with specific arities $s_\ell$ at all levels does not change the result, since the marginals converge to the uniform distribution regardless.

C.1.3. ROOT-TO-TERMINAL AND ROOT-TO-NONTERMINAL COVARIANCES

In this section, we study the properties of the covariance between the root $X^{(0)}$ and a single nonterminal or terminal symbol. The root-to-terminal covariance (with terminal position $i$) is defined as

$$C^{(L)}(i; a, b) := \mathbb{P}_{\mathcal{G}}\left\{X_i = a, X^{(0)} = b\right\} - \mathbb{P}_{\mathcal{G}}\left\{X_i = a\right\}\mathbb{P}_{\mathcal{G}}\left\{X^{(0)} = b\right\}$$

$$= \sum_c \mathbb{P}_{\mathcal{G}}\left\{X_i = a \Big| X^{(0)} = c\right\}\left(\delta_{c,b}\mathbb{P}_{\mathcal{G}}\left\{X^{(0)} = b\right\} - \mathbb{P}_{\mathcal{G}}\left\{X^{(0)} = c\right\}\mathbb{P}_{\mathcal{G}}\left\{X^{(0)} = b\right\}\right)$$

$$\Rightarrow C^{(L)}(i) = P_{s_L(i)}^{(L)} \cdots P_{s_1(i)}^{(1)} \Sigma_\pi \in \mathbb{R}^{v \times v}, \qquad \text{with} \quad \Sigma_\pi = \text{diag}(\pi) - \pi\pi^\top \in \mathbb{R}^{v \times v}. \tag{55}$$

After decomposing each $P$ as $J + \Delta$ and noting that $\mathbf{1}^\top \Sigma_\pi = 0$, thus $J\Sigma_\pi = 0$,

$$C^{(L)}(i) = \left(\Delta_{s_L(i)}^{(L)} \cdots \Delta_{s_1(i)}^{(1)}\right)\Sigma_\pi. \tag{56}$$

We now define the root-to-terminal covariance with randomized terminal position $i$. By construction, choosing a position uniformly among all terminals is equivalent to, at each level $\ell$: *(i)* drawing the branching factor $s_\ell \in \{2, 3\}$ (according to the model's branching probabilities $p_2$ and $p_3$), then *(ii)* drawing the child index $i$ uniformly from $\{1, \ldots, s_\ell\}$,

$$\widetilde{\Delta}^{(\ell)} := p_2\left(\frac{1}{2}\sum_{i=1}^2 \Delta_{i,2}^{(\ell)}\right) + p_3\left(\frac{1}{3}\sum_{i=1}^3 \Delta_{i,3}^{(\ell)}\right). \tag{57}$$

Consequently,

$$C_1^{(L)} = \left(\prod_{\ell=1}^L \widetilde{\Delta}^{(\ell)}\right)\Sigma_\pi. \tag{58}$$

**Lemma C.2** (Asymptotic root-to-terminal covariance)**.** *In the limit $v \to \infty$ with $m_s = fv^{s-1}$ for $s = 2, 3$ and finite $f \in (0, 1)$, the operator and Frobenius norm of the root-to-terminal covariance matrix converge to deterministic values independent of the sampling of the rules,*

$$\boxed{\|C_1^{(L)}\|_{\text{op}}^2 \to \frac{1}{v^2}\frac{(p_2^2/2)^L}{(fv)^L}, \qquad \|C_1^{(L)}\|_F^2 \to \frac{1}{v}\frac{(p_2^2/2)^L}{(fv)^L}.} \tag{59}$$

For $L = 1$, these scales agree with those of $\widetilde{\Delta}_s$ from Eq. 37 (recall that, by the asymptotic uniformity of the marginal distributions, approximately equal to $C_1^{(1)} \to \widetilde{\Delta}^{(1)}/v$). As is the case for $\widetilde{\Delta}_s$, the operator norm is smaller than the Frobenius norm by a factor $v$, and it can be shown to coincide asymptotically with the $\ell_2$-norm of the columns, signaling a flat spectrum (reasonable due to the symmetry of the rules' distribution for exchanging symbols).

We first compute the expectation of the squared Frobenius norm over grammars,

$$\mathbb{E}\left[\|C_1^{(L)}\|_F^2\right] = \text{Tr}\left(\Sigma_\pi \mathbb{E}\left[\left(\widetilde{\Delta}^{(L)}\cdots\widetilde{\Delta}^{(1)}\right)^\top\left(\widetilde{\Delta}^{(L)}\cdots\widetilde{\Delta}^{(1)}\right)\right]\Sigma_\pi\right). \tag{60}$$

After unfolding the transpose and recalling that the $\Delta$'s of different levels are independent, we get

$$\mathbb{E}\left[\|C_1^{(L)}\|_F^2\right] = \text{Tr}\left(\Sigma_\pi \mathbb{E}\left[\left(\widetilde{\Delta}^{(L-1)}\cdots\widetilde{\Delta}^{(1)}\right)^\top \widetilde{G}\left(\widetilde{\Delta}^{(L-1)}\cdots\widetilde{\Delta}^{(1)}\right)\right]\Sigma_\pi\right), \tag{61}$$

where $\widetilde{G} = \mathbb{E}\left[\widetilde{\Delta}^\top\widetilde{\Delta}\right]$. All the $\Delta$'s appearing in $\widetilde{\Delta}$ are independent of each other. Thus,

$$\widetilde{G} = \left(\frac{p_2}{2}\right)^2\left(\sum_{i=1}^2 \mathbb{E}\left[\Delta_{2,i}^\top\Delta_{2,i}\right]\right) + \left(\frac{p_3}{3}\right)^2\left(\sum_{i=1}^3 \mathbb{E}\left[\Delta_{3,i}^\top\Delta_{3,i}\right]\right) = \frac{p_2^2}{2}G_2 + \frac{p_3^2}{3}G_3$$

$$= \tilde{\lambda}_\| J + \tilde{\lambda}_\perp (I_v - J), \tag{62}$$

whence

$$\tilde{\lambda}_\perp = \frac{p_2^2}{2}\lambda_2^{(\perp)} + \frac{p_3^2}{3}\lambda_3^{(\perp)} = \frac{p_2^2}{2}\frac{1}{fv} + O\left(\frac{1}{v^3}\right), \quad \tilde{\lambda}_\| = \frac{p_2^2}{2}\lambda_2^{(\|)} + \frac{p_3^2}{3}\lambda_3^{(\|)} = \frac{p_2^2}{2}\frac{1-f}{fv} + O\left(\frac{1}{v^3}\right). \tag{63}$$

The $J$ part of $\widetilde{G}$ does not contribute because each $\Delta$ has zero column and row sums, thus

$$\mathrm{Tr}\left(\Sigma_\pi \, \mathbb{E}\left[\left(\widetilde{\Delta}^{(L-1)}\cdots\widetilde{\Delta}^{(1)}\right)^\top \widetilde{G}\left(\widetilde{\Delta}^{(L-1)}\cdots\widetilde{\Delta}^{(1)}\right)\right]\Sigma_\pi\right) =$$

$$\tilde{\lambda}_\perp \mathrm{Tr}\left(\Sigma_\pi \, \mathbb{E}\left[\left(\widetilde{\Delta}^{(L-1)}\cdots\widetilde{\Delta}^{(1)}\right)^\top \left(\widetilde{\Delta}^{(L-1)}\cdots\widetilde{\Delta}^{(1)}\right)\right]\Sigma_\pi\right) \Rightarrow$$

$$\mathbb{E}\left[\|C_1^{(L)}\|_F^2\right] = \left(\tilde{\lambda}_\perp\right)^L \|\Sigma_\pi\|_F^2. \tag{64}$$

With uniform root prior $\pi$, $\Sigma_\pi$ has a single eigenvalue $\lambda_\pi = 1/v$ with multiplicity $v-1$, thus $\|\Sigma_\pi\|_F^2 \to 1/v$. Using Eq. 63 for $\tilde{\lambda}_\perp$,

$$\mathbb{E}\left[\|C_1^{(L)}\|_F^2\right] = \left(\tilde{\lambda}_\perp\right)^L \|\Sigma_\pi\|_F^2 \to \frac{1}{v}\frac{(p_2^2/2)^L}{(fv)^L}. \tag{65}$$

Concentration about the mean Frobenius norm follows from the sub-gaussianity of the $\widetilde{\Delta}$'s and the resulting concentration of the $\widetilde{\Delta}$'s Frobenius norms due to the independence of the levels.

For the operator norm, we can get an upper bound by submultiplicativity,

$$\|C_1^{(L)}\|_{\mathrm{op}} \le \|\Sigma_\pi\|_{\mathrm{op}} \prod_{\ell=0}^{L-1} \|\widetilde{\Delta}^{(\ell)}\|_{\mathrm{op}} \to \frac{1}{v}\frac{(p_2^2/2)^{L/2}}{(fv)^{L/2}}, \tag{66}$$

where we used $\|\Sigma_\pi\|_{\mathrm{op}} \to 1/v$ and $\|\widetilde{\Delta}\|_{\mathrm{op}}^2 \to \tilde{\lambda}_\perp$. In fact, there is a matching lower bound obtained directly by the property $\|C\|_{\mathrm{op}} \ge \|C\|_F/\sqrt{\mathrm{rank}(C)}$, or

$$\|C_1^{(L)}\|_{\mathrm{op}}^2 \ge \max_b \|(C_1^{(L)})_{:,b}\|_2^2 \ge \frac{1}{v}\sum_v \|(C_1^{(L)})_{:,b}\|_2^2 = \frac{\|C_1^{(L)}\|_F^2}{v} \to \frac{1}{v^2}\frac{(p_2^2/2)^L}{(fv)^L}. \tag{67}$$

**Lemma C.3** (Orthogonality of the root-to-terminal covariance rows). *Denote the rows of the root-to-terminal covariance matrix with $r_z^{(L)} := C_1^{(L)}(z,:) \in \mathbb{R}^v$. In the limit $v \to \infty$ with $m_s = fv^{s-1}$ for $s = 2,3$ and finite $f \in (0,1)$, the squared norms of the row vectors converge to a deterministic value independent of the sampling of the rules and of $z$,*

$$\boxed{\|r_z^{(L)}\|_2^2 \to \frac{1}{v^2}\frac{(p_2^2/2)^L}{(fv)^L} = \|C_1^{(L)}\|_{\mathrm{op}}^2 = \frac{\|C_1^{(L)}\|_F^2}{v}.} \tag{68}$$

*In addition, the row vectors become orthogonal to each other,*

$$\boxed{\cos\left(r_z^{(L)}, r_{z'}^{(L)}\right) = O_P\left(\frac{1}{\sqrt{v}}\right) \quad \text{for fixed} \quad z \ne z', \quad \max_{z \ne z'} |\cos\left(r_z^{(L)}, r_{z'}^{(L)}\right)| = O_P\left(\frac{\ln v^{A/2}}{\sqrt{v}}\right),} \tag{69}$$

*for some constant $A$, where the notation $a = O_P(b)$ means that the probability of $a = O(b)$ over draws of the random rules converges to 1.*

We prove this lemma by induction on the levels $L$.

**norm, $L = 1$.** For $L = 1$, by the asymptotic uniformity of the marginal C.1,

$$C_1^{(1)}(z,r) \to \frac{1}{v}\widetilde{\Delta}^{(1)}(z,r) \Rightarrow r_z^{(1)} \to \frac{1}{v}\widetilde{\Delta}^{(1)}(z,:) \in \mathbb{R}^v. \tag{70}$$

To ease notation, we replace $\widetilde{\Delta}^{(1)}$ with a binary transition matrix with fixed child index $i$, e.g. $\Delta_{1,2}^{(1)}$, then omit the layer superscript and the child index and arity subscripts. Since the moments of the $\widetilde{\Delta}^{(1)}$'s have the same scaling in $v$ as the moments of the $\Delta_{1,2}^{(1)}$, all the following results can be easily extended to the $\widetilde{\Delta}$'s after replacing the variance of $\Delta$ ($1/(m_2 v)$) with that of $\widetilde{\Delta}$ ($(p_2^2/2)/(m_2 v)$). Thus, the norm of the row vectors is

$$\|r_z\|_2^2 = \frac{1}{v^2} \sum_{r=1}^v \Delta(z,r)^2. \tag{71}$$

As $\mathbb{E}[\Delta(z,r)^2] \to (m_2 v)^{-1}$, the expectation of the squared norm converges to $(v^2 m_2)^{-1}$. The variance reads

$$\mathrm{Var}\left[\|r_z\|_2^2\right] = \frac{1}{v^4} \sum_{r,r'} \left(\mathbb{E}\left[\Delta(z,r)^2 \Delta(z,r')^2\right] - \mathbb{E}\left[\Delta(z,r)^2\right]\mathbb{E}\left[\Delta(z,r')^2\right]\right)$$

$$= \frac{1}{v^4}\left(v\mathrm{Var}\left[\Delta(z,r)^2\right] + v(v-1)\mathrm{Cov}\left[\Delta(z,r)^2, \Delta(z,r')^2\right]\right). \tag{72}$$

$\mathrm{Var}\left[\Delta(z,r)^2\right] = O(v^{-4})$ by sub-Gaussianity of the $\Delta$'s with variance proxy $O(v^{-2})$, resulting in a $O(v^{-7})$ contribution to $\mathrm{Var}\left[\|r_z\|_2^2\right]$. For the covariance term, we first express the elements of the $\Delta$ matrix via the occurrences of the child symbol $z$ in the $m_2$ production rules assigned to parent symbol $r$,

$$\Delta(z,r) = \frac{N_{z,r} - f_2}{m_2}, \quad (N_{z,r})_z \sim \mathrm{Hypergeom}\left(N = v^2, (K_i = v)_{i=1}^v, n = m_2\right), \tag{73}$$

so that, using $N^2 = (N)_2 + N$,

$$\mathrm{Cov}\left[\Delta(z,r)^2, \Delta(z,r')^2\right] = \frac{1}{m_2^4}\mathrm{Cov}\left[(N_{z,r})_2 + (1-2f_2)N_{z,r}, (N_{z,r'})_2 + (1-2f_2)N_{z,r'}\right]. \tag{74}$$

For $r \neq r'$, the pair $(N_{z,r}, N_{z,r'})$ is a two-cell multivariate hypergeometric count with the same total and target populations,

$$\mathbb{E}\left[(N_{z,r})_n (N_{z,r'})_k\right] = \frac{(m_2)_n (m_2)_k (v)_{n+k}}{(v^2)_{n+k}}. \tag{75}$$

Hence, all the contributions to the covariance have the form

$$\mathbb{E}\left[(N_{z,r})_n (N_{z,r'})_k\right] - \mathbb{E}\left[(N_{z,r})_n\right]\mathbb{E}\left[(N_{z,r'})_k\right] = \frac{(m_2)_n (m_2)_k (v)_{n+k}}{(v^2)_{n+k}} - \frac{(m_2)_n (v)_n}{(v^2)_n}\frac{(m_2)_k (v)_k}{(v^2)_k} =$$

$$(m_2)_n (m_2)_k \left[\frac{(v)_{n+k}}{(v^2)_{n+k}} - \frac{(v)_n}{(v^2)_n}\frac{(v)_k}{(v^2)_k}\right] = -\frac{nk}{v} + O\left(\frac{1}{v^2}\right), \tag{76}$$

where, in the last equality, we used the expansion $(v)_n = v^n(1 - v^{-1})\ldots(1 - (n-1)v^{-1}) = v^n(1 - (n(n-1)/2)v^{-1} + O(v^{-2}))$. Multiplying by the overall $m_2^{-4}$ factor yields $\mathrm{Cov}\left[\Delta(z,r)^2, \Delta(z,r')^2\right] = O(v^{-5})$, resulting in another $O(v^{-7})$ contribution to $\mathrm{Var}\left[\|r_z\|_2^2\right]$. Comparison with the $O(v^{-6})$ scale of the squared expectation of the squared norm proves that $\|r_z\|_2^2$ converges to its expectation in probability.

**cosines, $L=1$.** Consider the scalar product between distinct row vectors,

$$\langle r_z, r_{z'} \rangle = \frac{1}{v^2} \sum_{r=1}^v \Delta(z,r)\Delta(z',r). \tag{77}$$

with $z' \neq z$. Due to the 0-sum constraint $\sum_z \Delta(z,r) = 0$, $\mathbb{E}\left[\Delta(z,r)\Delta(z',r)\right] = -(v-1)^{-1}\mathbb{E}\left[\Delta(z,r)^2\right]$, hence $\mathbb{E}\left[\langle r_z, r_{z'} \rangle\right] \to -v^{-1}\mathbb{E}\left[\|r_z\|_2^2\right]$. After dividing by the norms and using their convergence to a deterministic limit from the previous paragraph, we get

$$\mathbb{E}\left[\cos(r_z, r_{z'})\right] \to \frac{\mathbb{E}\left[\langle r_z, r_{z'} \rangle\right]}{\|r_z\|_2 \|r_{z'}\|_2} \to -\frac{1}{v}. \tag{78}$$

To understand fluctuations, we look at the high-order even moments of the scalar product,

$$\mathbb{E}\left[(\langle r_z, r_{z'}\rangle)^{2q}\right] = \frac{1}{v^{4q}} \sum_{r_1,\ldots,r_{2q}} \mathbb{E}\left[\Delta(z, r_1)\Delta(z', r_1)\ldots\Delta(z, r_{2q})\Delta(z', r_{2q})\right]. \tag{79}$$

After setting $\Gamma(r) := \Delta(z, r)\Delta(z', r)$, we can group the summands by the number $k \leq 2q$ of distinct unique root indices $r_i$ appearing. Denoting with $n_i$ the multiplicity of the index $r_i$, such that $\sum_i n_i = 2q$,

$$\mathbb{E}\left[(\langle r_z, r_{z'}\rangle)^{2q}\right] = \frac{1}{v^{4q}} \sum_{k=1}^{2q} (v)_k \sum_{n_1+\cdots+n_k=2q} \frac{(2q)!}{n_1!\ldots n_k!} \mathbb{E}\left[\prod_{i=1}^{k}\Gamma(r_i)^{n_i}\right]. \tag{80}$$

Each $\Gamma(r)$ contains a factor of $m_2^{-2}$ times $(N_{z,r} - f_2)(N_{z,r'} - f_2)$. Furthermore,

$$(N_{z,r} - f_2)^n = \sum_{t=0}^{n}\binom{n}{t}(-f_2)^{n-t}N_{z,r}^t = \sum_{t=0}^{n}\binom{n}{t}(-f_2)^{n-t}\sum_{a=0}^{t}S(t,a)(N_{z,r})_a$$

$$:= \sum_{a=0}^{n}c_a(n)(N_{z,r})_a, \tag{81}$$

where we used the Stirling numbers of the second kind $S(t, a)$ to express the powers of $N_{z,r}$ as sums over falling factorials. As a result,

$$\mathbb{E}\left[\prod_{i=1}^{k}\Gamma(r_i)^{n_i}\right] = \frac{1}{m_2^{4q}} \sum_{a_1,b_1=0}^{n_1} \cdots \sum_{a_k,b_k=0}^{n_k}\left(\prod_{i=1}^{k}c_{a_i}(n_i)c_{b_i}(n_i)\right)\mathbb{E}\left[\prod_{i=1}^{k}(N_{z,r})_{a_i}(N_{z',r})_{b_i}\right] \tag{82}$$

Since

$$\mathbb{E}\left[\prod_{i=1}^{k}(N_{z,r})_{a_i}(N_{z',r})_{b_i}\right] = \frac{\left(\prod_{i=1}^{k}(m_2)_{a_i+b_i}\right)(v)_{\sum_i a_i}(v)_{\sum_i a_i}}{(v^2)_{\sum_i(a_i+b_i)}} = f^{\sum_i(a_i+b_i)}\left(1 + O(v^{-1})\right), \tag{83}$$

The scaling in $v$ is always bounded by the $m_2^{-4q} = O(v^{-4q})$ factor. Thus, after bounding $(v)_k$ with $v^k$ and all the remaining multiplicity factors with $(Cq)^{Cq}$ for some constant $C \geq 1$, we get

$$\frac{1}{v^{4q}}(v)_k\frac{(2q)!}{n_1!\ldots n_k!}\mathbb{E}\left[\prod_{i=1}^{k}\Gamma(r_i)^{n_i}\right] \leq (Cq)^{Cq}v^{k-8q}. \tag{84}$$

The highest possible $k$ is $2q$. However, every $n_i = 1$ results in an additional cancellation that reduces the order of $\mathbb{E}\left[\prod_{i=1}^{k}\Gamma(r_i)^{n_i}\right]$ by 1. This is because each term with $n_i = 1$ results in a factor $(N_{z,r} - f_2)(N_{z',r} - f_2) = N_{z,r}N_{z',r} + f_2^2 - f_2 N_{z,r} - f_2 N_{z',r}$ inside the expectation, times additional factors of $(N_{z,r'})_a(N_{z',r'})_b$ with $r' \neq r$. For all these 4 contributions, the leading-order expectation is identical, reducing the order by 1. In the case $k = 2q$, since $n_i = 1$ for all $i = 1, \ldots, 2q$, this effect reduces the order by $2q$. The largest contribution comes from the case $k = q$ and $n_i = 2$ for all $i$, where there are no extra cancellations due to $n_i = 1$, thus

$$\mathbb{E}\left[(\langle r_z, r_{z'}\rangle)^{2q}\right] \leq (Cq)^{Cq}v^{-7q} \tag{85}$$

By Markov's inequality, over draws of the random rules,

$$\mathbb{P}\left\{\langle r_z, r_{z'}\rangle \geq \varepsilon_v\right\} \leq \frac{\mathbb{E}\left[(\langle r_z, r_{z'}\rangle)^{2q}\right]}{\varepsilon_v^{2q}} \leq \left(\frac{(Cq)^{C/2}}{v^{7/2}\epsilon_v}\right)^{2q} \Rightarrow$$

$$\mathbb{P}\left\{\frac{\langle r_z, r_{z'}\rangle}{\|r_z\|_2\|r_{z'}\|_2} \geq \eta_v\right\} \leq \left(\frac{(Cq)^{C/2}}{v^{1/2}\eta_v}\right)^{2q} \Rightarrow$$

$$\mathbb{P}\left\{\max_{z\neq z'}\frac{\langle r_z, r_{z'}\rangle}{\|r_z\|_2\|r_{z'}\|_2} \geq \eta_v\right\} \leq v^2\left(\frac{(Cq)^{C/2}}{v^{1/2}\eta_v}\right)^{2q} \tag{86}$$

where we used $\|r_z\|_2 \to (v^2 m_2)^{-1/2} = O(v^{-3/2})$ and absorbed a factor of $v^3$ in $\eta_v$. Setting $q = c \log v$ and $\eta_v = Av^{-1/2}(Cq)^{C/2}$,

$$\mathbb{P}\left\{ \max_{z \neq z'} \frac{\langle r_z, r_{z'} \rangle}{\|r_z\|_2 \|r_{z'}\|_2} \geq A\sqrt{\frac{(C \log v)^{C/2}}{v}} \right\} \leq v^2 A^{-2c \log v} = v^{2(1-c \log A)}, \tag{87}$$

which tends to 0 iff $c \log A > 1$.

**Induction step.** The induction step is simple. For the norms,

$$\|r_z^{(L)}\|_2^2 = \sum_{a,b} \Delta^{(L)}(z,a)\Delta^{(L)}(z,b) r_a^{(L-1)} r_b^{(L-1)} = \sum_a (\Delta^{(L)}(z,a))^2 \|r_a^{(L-1)}\|_2^2 + \sum_{a \neq b} \Delta^{(L)}(z,a)\Delta^{(L)}(z,b) \left\langle r_a^{(L-1)}, r_b^{(L-1)} \right\rangle \tag{88}$$

Using the convergence of the level-$L-1$ squared norms to their mean, the diagonal term gives

$$\mathbb{E}\left[ \|r_a^{(L-1)}\|_2^2 \right] \sum_a (\Delta^{(L)}(z,a))^2 \to \frac{\mathbb{E}\left[ \|r_a^{(L-1)}\|_2^2 \right]}{m_2}, \tag{89}$$

since the $\Delta's$ are i.i.d. across levels and we just proved convergence of the squared norms of the rows for $\Delta^{(1)}$. For the off-diagonal term, we use asymptotic orthogonality:

$$\left\langle r_a^{(L-1)}, r_b^{(L-1)} \right\rangle = o_P(1)\mathbb{E}\left[ \|r_a^{(L-1)}\|_2^2 \right] \tag{90}$$

uniformly in $a \neq b$, thus

$$\sum_{a \neq b} \Delta^{(L)}(z,a)\Delta^{(L)}(z,b) \left\langle r_a^{(L-1)}, r_b^{(L-1)} \right\rangle = o(1)\mathbb{E}\left[ \|r_a^{(L-1)}\|_2^2 \right] \left( \sum_{a \neq b} \Delta^{(L)}(z,a)\Delta^{(L)}(z,b) \right) \tag{91}$$

In addition, due to the 0-sum constraint $\sum_a \Delta^{(L)}(z,a) = 0$,

$$\left( \sum_{a \neq b} \Delta^{(L)}(z,a)\Delta^{(L)}(z,b) \right) = \left( \sum_a \Delta^{(L)}(z,a) \right)\left( \sum_b \Delta^{(L)}(z,b) \right) - \sum_a (\Delta^{(L)}(z,a))^2 = -\sum_a (\Delta^{(L)}(z,a))^2. \tag{92}$$

Thus, the off-diagonal term is bounded by the diagonal term times an $o_p(1)$ coefficient that vanishes asymptotically. For the angles, the mechanism is similar: First, separate the iterative definition into diagonal and off-diagonal terms,

$$\left\langle r_z^{(L)}, r_{z'}^{(L)} \right\rangle = \sum_a \Delta^{(L)}(z,a)\Delta^{(L)}(z',a)\|r_a^{(L-1)}\|_2^2 + \sum_{a \neq b} \Delta^{(L)}(z,a)\Delta^{(L)}(z',b) \left\langle r_a^{(L-1)}, r_b^{(L-1)} \right\rangle \tag{93}$$

Secondly, use the asymptotic orthogonality of the previous-level rows to bound the off-diagonal term with the diagonal term times a $o_P(1)$ factor. Finally, notice that the $\Delta^{(L)}$ are i.i.d. across the layers, so the asymptotic orthogonality of the rows of $\Delta^{(1)}$ implies that of the rows of $\Delta^{(L)}$.

**Remark.** Notice the hierarchy of scales between the elements of the covariance matrix, whose scale is given by their variance over draws of the rules,

$$C_1^{(L)}(a,b)^2 = O_{\mathbb{P}}\left( \frac{1}{v^3} \frac{(p_2^2/2)^L}{(fv)^L} \right), \tag{94}$$

then the squared norm of the rows, sum of the squared elements over $a$ hence $\approx v C_1^{(L)}(a,b)^2$, and the Frobenius norm, sum of the squared elements over $a$ and $b$, hence $\approx v^2 C_1^{(L)}(a,b)^2$.

### C.2. Binary rules inference from the root-to-pair covariance

In this section, we prove that, in the asymptotic vocabulary size limit, 2 correctly infers the set of level-$\ell$ binary rules of the Varying-Tree RHM from the level-$\ell$ root-to-pair covariance tensor. The proof relies on the decomposition of the $v$-dimensional vector $u_{ab} := C_2^{(\ell)}((a,b),:)$ into the contribution of binary production rules and those of the other possible local configurations. Non-binary contributions are always negligible in the asymptotic limit, while the binary contribution is proportional to the row vector of the root-to-level-$(\ell-1)$ covariance $C_1^{(\ell-1)}(z,:)$ for all the pairs $(a,b)$ generated by the lower-level nonterminal $z$. By C.3, these row vectors are orthogonal, hence clustering the $u_{ab}$'s by their direction recovers the level-$\ell$ binary rules.

For any level $\ell = 2, \ldots, L$, consider the sequence of level-$\ell$ nonterminals $\mathbf{X}^{(\ell)} = (X_1^{(\ell)}, \ldots, X_T^{(\ell)})$ and a uniformly random index $I \in \{1, \ldots, T-1\}$. For a fixed pair $(a,b) \in \mathcal{V}^2$, define

$$X_{a,b}^{(\ell,\text{pair})} := \mathbb{E}_{\mathcal{G}}\left[ \mathbf{1}\{(X_I^{(\ell)}, X_{I+1}^{(\ell)}) = (a,b)\} \,\Big|\, \mathbf{X}^{(\ell)} \right] = \frac{1}{T-1}\sum_{i=1}^{T-1}\mathbf{1}\{(X_i^{(\ell)}, X_{i+1}^{(\ell)}) = (a,b)\}. \tag{95}$$

We study the covariance $C_2^{(\ell)}$ of $X_{a,b}^{(\ell,\text{pair})}$ with the root one-hot vector, $X^{(0)}$ over the Varyng-Tree RHM distribution . Given $(a,b)$, this covariance is a vector in $\mathbb{R}^v$ with one component per possible root symbol $x^{(0)}$. With root prior probability $\pi$,

$$C_2^{(\ell)}((a,b),:) := \text{Cov}_{\mathcal{G}}(X_{a,b}^{(\ell,\text{pair})}, X^{(0)}) = \mathbb{E}_{\mathcal{G}}\big[X_{a,b}^{(\ell,\text{pair})}X^{(0)\top}\big] - \mathbb{E}_{\mathcal{G}}[X_{a,b}]\pi^\top \quad (\in \mathbb{R}^v). \tag{96}$$

To average over trees and positions, we partition by the type of the lowest common ancestor (LCA) of $(X_I^{(\ell)}, X_{I+1}^{(\ell)})$, as illustrated in Fig. 4: (bin) level-$(\ell-1)$ followed by a binary rule (i.e. $(X_I^{(\ell)}, X_{I+1}^{(\ell)})$ are binary siblings, as in diagram *i)* of the figure), (ter) level-$(\ell-1)$ followed by a ternary rule (i.e. $(X_I^{(\ell)}, X_{I+1}^{(\ell)})$ are ternary siblings, as in diagrams *ii)* and *iii)*) (cross, $\ell'$) LCA of level $0 \leq \ell' \leq \ell-2$ (i.e. $(X_I^{(\ell)}, X_{I+1}^{(\ell)})$ sit across the boundary between input tuples, as in diagram *iv)*). By the law of total covariance,

$$\text{Cov}_{\mathcal{G}}(X_{a,b}^{(\ell,\text{pair})}, X^{(0)}) = \mathbb{E}_{\mathcal{G}}\left[\text{Cov}_{\mathcal{G}}(X_{a,b}^{(\ell,\text{pair})}, X^{(0)})\Big|\text{type(LCA)}\right] +$$
$$\text{Cov}_{\mathcal{G}}\left(\mathbb{E}_{\mathcal{G}}\left[X_{a,b}^{(\ell,\text{pair})}\Big|\text{type(LCA)}\right], \mathbb{E}_{\mathcal{G}}\left[X^{(0)}\Big|\text{type(LCA)}\right]\right). \tag{97}$$

Since the root is independent of $\text{type(LCA)}$, $\mathbb{E}\left[X^{(0)}\Big|\text{type(LCA)}\right]$ equals the root prior deterministically, and the second term on the right-hand side above vanishes. For the binary and both ternary terms, the type of the last production rule is fixed. Hence, these terms factorize into the product of the level-$\ell$ rules tensor and the root-to-level-$(\ell-1)$-nonterminal covariance,

$$\text{Cov}_{\mathcal{G}}\left(X_{a,b}^{(\ell,\text{pair})}, X^{(0)}\Big|\text{type(LCA)} = \text{bin}\right) = \sum_{z=1}^v R_{z\to(a,b)}^{(\ell)}C_1^{(\ell-1)}(z,:),$$

$$\text{Cov}_{\mathcal{G}}\left(X_{a,b}^{(\ell,\text{pair})}, X^{(0)}\Big|\text{type(LCA)} = \text{ter}\right) = \sum_{z=1}^v\sum_{c=1}^v\left(R_{z\to(a,b,c)}^{(\ell)} + R_{z\to(c,a,b)}^{(\ell)}\right)C_1^{(\ell-1)}(z,:).$$

Finally, denoting with $p_{\text{bin}}$, $p_{\text{ter}}$ and $p_{\text{cross}}$ the probabilities of the different LCA types,

$$\text{Cov}_{\mathcal{G}}(X_{a,b}^{(\ell,\text{pair})}, X^{(0)}) = p_{\text{bin}}\sum_{z=1}^v R_{z\to(a,b)}^{(\ell)}C_1^{(\ell-1)}(z,:)$$
$$+ p_{\text{ter}}\sum_{z=1}^v\sum_{c=1}^v\left(R_{z\to(a,b,c)}^{(\ell)} + R_{z\to(c,a,b)}^{(\ell)}\right)C_1^{(\ell-1)}(z,:)$$
$$+ p_{\text{cross}}C^{\text{cross}}((a,b),:), \tag{98}$$

(bin) The binary contribution is equal to the matrix product between the $v^2$-by-$v$ binary rules tensor $R_{z\to(a,b)}^{(\ell)}$ and the root-to-level-$\ell$ covariance matrix $C_1^{(\ell-1)}(z,r)$. For every pair $(a,b)$ of binary siblings, $R_{z\to(a,b)}^{(\ell)}$ selects the unique

$z(a, b)$ that generates the pair via a binary rule:

$$C_2^{(\ell,\text{bin})}((a,b),:) = \begin{cases} \frac{1}{m_2} C_1^{(\ell-1)}(z(a,b),:) & \text{if} \quad \exists\, z \in \mathcal{N}_{\ell-1} \text{ s.t. } (z \to ab) \in \mathcal{R}_2^{(\ell)}, \\ 0 & \text{otherwise.} \end{cases} \tag{99}$$

By Eq. 68, for all pairs $(a, b)$ of binary siblings,

$$\|C_2^{(\ell,\text{bin})}((a,b),:)\|_2^2 = \frac{\|r_{z(a,b)}^{(\ell-1)}\|_2^2}{m_2^2} \to \frac{1}{v^2 m_2^2} \frac{(p_2^2/2)^{\ell-1}}{m_2^{\ell-1}}. \tag{100}$$

(ter) The ternary contribution can be written as the product of a level-$\ell$ kernel $K_{(a,b),z}^{(\text{ter})}$ and the root-to-level-$(\ell-1)$ covariance rows $C_1^{(\ell-1)}(z,:)$. The kernel reads

$$\begin{aligned} K_{(a,b),z}^{(\text{ter})} &= \sum_c \frac{1}{2} \left( \mathbb{P}\left\{ (X_U^{(\ell)}, X_{U+1}^{(\ell)}, X_{U+2}^{(\ell)}) = (a,b,c) \,\middle|\, X_U^{(\ell-1)} = z \right\} + \right. \\ &\qquad\qquad \left. \mathbb{P}\left\{ (X_{U-1}^{(\ell)}, X_U^{(\ell)}, X_{U+1}^{(\ell)}) = (c,a,b) \,\middle|\, X_U^{(\ell-1)} = z \right\} \right) \\ &= \frac{1}{2} \left( \left( P_{(12),3}^{(\ell)} \right)_{(a,b),z} + \left( P_{(23),3}^{(\ell)} \right)_{(a,b),z} \right), \end{aligned} \tag{101}$$

where $P_{(ij),3}^{(\ell)}$ denotes the ternary rules parent-to-child $v^2 \times v$ transition matrix with children positions $i$ and $j$. By generalising Eq. 29 and Eq. 32,

$$P_{(ij),3}^{(\ell)} = \frac{1}{v^2} \mathbf{1}_{v^2} \mathbf{1}_v^\top + \Delta_{(ij),3}^{(\ell)}, \quad \text{with} \quad \mathbb{E}[\Delta_{(ij),3}^{(\ell)}] = 0,$$

$$\text{Var}\left[ \Delta_{(ij),3}^{(\ell)} \right] = \frac{1}{v^2 m_3} \left( 1 - \frac{1}{v^2} \right) \frac{v^3 - m_3}{v^3 - 1} \to \frac{1}{f_3 v^4} + O\left( \frac{1}{v^5} \right). \tag{102}$$

By the same mechanism of the induction step in the proof of C.3, in the asymptotic vocabulary size limit

$$\|C_2^{(\ell,\text{ter})}((a,b),:)\|_2^2 \to \left( \sum_z \left( \Delta_{(ij),3}^{(\ell)}((a,b),z) \right)^2 \right) \|r_z^{(\ell-1)}\|_2^2 \to \frac{\|r_z^{(\ell-1)}\|_2^2}{v m_3}. \tag{103}$$

Since $m_3 = O(v^2)$, this norm is smaller than that of the binary contribution by a factor of $1/v$,

$$\|C_2^{(\ell,\text{ter})}((a,b),:)\|_2^2 = O\left( \frac{\|C_2^{(\ell,\text{bin})}((a,b),:)\|_2^2}{v} \right) \tag{104}$$

(cross) All the other contributions consist of a root-to-nonterminal covariance over $\ell' \le \ell - 2$ layers, then a binary transition and two branches from the children of this transition to the terminal symbols. This term is necessarily smaller than the others because, even with $\ell' = \ell - 2$, it contains an extra transition at the last level due to the extra branch, adding at least another factor of $(m_2)^{-1}$ to the norms.

We are now ready to state the full proposition about the correctness of 2 (cf. 3.1 of the main text).

**Proposition C.4** (Correctness of INFERBINARY from population root-to-pair covariances). *Consider data generated by a varying-tree RHM in the asymptotic vocabulary size limit $v \to \infty$ with $m_2 = f_2 v$ and $m_3 = f_3 v^2$. Fix a level $\ell \in \{2, \dots, L\}$. For each pair $(a, b) \in \mathcal{N}_\ell^2$, let*

$$u_{ab} := C_2^{(\ell)}((a,b),:) \in \mathbb{R}^v, \tag{105}$$

*and let $\mathcal{B}_\ell \subset \mathcal{N}_\ell^2$ denote the set of valid level-$\ell$ binary sibling pairs,*

$$\mathcal{B}_\ell = \left\{ (a,b) \in \mathcal{N}_\ell^2 \mid \exists z \in \mathcal{N}_{\ell-1} \text{ s.t. } (z \to ab) \in \mathcal{R}_2^{(\ell)} \right\}. \tag{106}$$

*Define*

$$\tau_2 := \gamma\, v^{-1} \Big( \sum_{a,b} \|u_{ab}\|_2^2 \Big)^{1/2}, \qquad \gamma \in (0,1). \tag{107}$$

*Then, with high probability over the random grammar, for all sufficiently large $v$:*

1. *$\|u_{ab}\|_2 > \tau_2$ if and only if $(a,b) \in \mathcal{B}_\ell$;*

2. *for every valid pair $(a,b) \in \mathcal{B}_\ell$,*

$$\left\| \frac{u_{ab}}{\|u_{ab}\|_2} - \frac{C_1^{(\ell-1)}(z(a,b),:)}{\|C_1^{(\ell-1)}(z(a,b),:)\|_2} \right\|_2 \to 0; \tag{108}$$

3. *consequently, thresholding the slices $u_{ab}$ with $\tau_2$ selects exactly the valid binary sibling pairs, and clustering the normalized surviving vectors $u_{ab}/\|u_{ab}\|_2$ recovers exactly their partition by parent nonterminal.*

*In particular,* INFERBINARY *reconstructs the full set of level-$\ell$ binary rules, up to permutation of parent labels.*

The proof follows from the analysis of the root-to-pair covariance. Indeed, by Eq. 98, one has

$$u_{ab} = \begin{cases} s_{ab} + e_{ab} & \text{for} \quad (a,b) \in \mathcal{B}_\ell, \\ e_{ab} & \text{otherwise,} \end{cases} \tag{109}$$

with $s_{ab} = m_2^{-1} C_1^{(\ell-1)}(z(a,b),:)$, and

$$\|s_{ab}\|_2 \to S_v^{(\ell)} := \frac{p_{\mathrm{bin}}}{m_2 v} \sqrt{\frac{(p_2^2/2)^{\ell-1}}{m_2^{\ell-1}}}, \qquad \|e_{ab}\|_2^2 = o\Big( (S_v^{(\ell)})^2 \Big). \tag{110}$$

Hence, recalling that there are $vm_2$ pairs $(a,b)$ in $\mathcal{B}_\ell$, $\tau_2 \to \gamma\sqrt{f_2} S_v$. For $\gamma \in (0,1)$ and sufficiently large $v$, this is smaller than all $\|u_{ab}\|_2$'s with $(a,b) \in \mathcal{B}_\ell$, while bigger that $\|u_{ab}\|_2 = \|e_{ab}\|_2$ when $(a,b) \notin \mathcal{B}_\ell$. The normalized rows of $C_1^{(\ell-1)}$ are asymptotically orthogonal by C.3, which ensures the validity of the clustering step.

### C.3. Ternary rules inference from root-to-pair and root-to-triple covariances

In this section, we prove that, in the asymptotic vocabulary size limit, 3 correctly infers the set of level-$\ell$ ternary rules of the Varying-Tree RHM from the level-$\ell$ root-to-pair and root-to-triple covariance tensors. As in the previous section, the proof relies on decomposing the root-to-triple covariance tensor into contributions from the different possible local tree topologies. In this case, the contribution of genuine ternary siblings competes with other spurious contributions, but these spurious contributions can be removed using the root-to-pair covariance tensor and the knowledge of level-$\ell$ binary rules from 2.

As in subsection C.2, consider a sequence $\mathbf{X}^{(\ell)}$ of level-$\ell$ nonterminals and a random index $I \in \{1, \ldots, T-2\}$. For a fixed triple $(a,b,c) \in \mathcal{V}^3$, define

$$X_{a,b,c}^{(\ell,\mathrm{triple})} := \mathbb{E}_{\mathcal{G}}\left[ \mathbf{1}\left\{ (X_I^{(\ell)}, X_{I+1}^{(\ell)}, X_{I+2}^{(\ell)}) = (a,b,c) | \mathbf{X}^{(\ell)} \right\} \right] = \frac{1}{T-2} \sum_{i=1}^{T-2} \mathbf{1}\left\{ (X_i^{(\ell)}, X_{i+1}^{(\ell)}, X_{i+2}^{(\ell)}) = (a,b,c) \right\}, \tag{111}$$

The root-to-triple covariance is the $v \times v \times v \times v$ covariance tensor between $X^{(\ell,\mathrm{triple})}$ and the root one-hot vector $X^{(0)}$, or a vector in $\mathbb{R}^v$ per triple $(a,b,c)$. As in subsection C.2, we use the law of total covariance to decompose the covariance into contributions corresponding to different local topologies (see Fig. 4 for an illustration). The three terminal symbols selected can share the same LCA at the previous level $\ell-1$ (i.e. they are ternary siblings, probability $p_{\mathrm{ter}}$) or only two of the three (either $(a,b)$ or $(b,c)$ with the same probability) come from the same level-$(\ell-1)$ nonterminal. In the latter case, the two

terminals sharing the LCA are either binary (probability $p_{\text{bin}}$) or ternary siblings (probability $p_{\text{ter}}$). Therefore,

$$C_3^{(\ell)}((a,b,c),:) := \text{Cov}_{\mathcal{G}}\left(X_{a,b,c}^{(\ell,\text{triple})}, X^{(0)}\right) = p_{\text{ter}} \sum_{z=1}^{v} R_{z\to(a,b,c)}^{(\ell)} C_1^{(\ell-1)}(z,:)$$

$$+ \sum_{z=1}^{v}\left(p_{\text{bin}} R_{z\to(a,b)}^{(\ell)} + p_{\text{ter}} \sum_{d=1}^{v} R_{z\to(d,a,b)}^{(\ell)}\right)\sum_{z'=1}^{v}\widetilde{P}_{c,z'}^{(\ell)} C_2^{(\ell-1)}((z,z'),:)$$

$$+ \sum_{z=1}^{v}\widetilde{P}_{a,z}^{(\ell)}\sum_{z'=1}^{v}\left(p_{\text{bin}} R_{z'\to(b,c)}^{(\ell)} + p_{\text{ter}} \sum_{d=1}^{v} R_{z'\to(b,c,d)}^{(\ell)}\right) C_2^{(\ell-1)}((z,z'),:), \tag{112}$$

where $\widetilde{P}_{a,z}^{(\ell)}$ denotes the parent-to-child transition matrix from Eq. 29, averaged over arities and child positions. [4]

(ter) The pure ternary contribution consists of the matrix product between the $v^3$-by-$v$ ternary rules tensor $R_{z\to(a,b,c)}^{(\ell)}$ and the root-to-level-$(\ell-1)$ covariance rows $C_1^{(L-1)}(z,:)$. As in the binary contribution of the root-to-pair covariance tensor,

$$C_3^{(\ell,\text{ter})}((a,b,c),:) = \begin{cases} \frac{1}{m_3} C_1^{(\ell-1)}(z(a,b,c),:) & \text{if}\quad \exists\, z \in \mathcal{N}_{\ell-1} \text{ s.t. } (z\to abc) \in \mathcal{R}_3^{(\ell)}, \\ 0 & \text{otherwise.} \end{cases} \tag{113}$$

Thus, for all triples of ternary siblings,

$$\|C_3^{(\ell,\text{ter})}((a,b,c),:)\|_2^2 = \frac{\|r_{z(a,b,c)}^{(\ell-1)}\|_2^2}{m_3^2} \to \frac{1}{v^2 m_3^2}\frac{(p_2^2/2)^{\ell-1}}{m_2^{\ell-1}}. \tag{114}$$

(cross) Here we focus on the term corresponding to $(a,b)$ coming from the same level-$(\ell-1)$ nonterminal and $c$ from a different one, as the other term has identical statistics by symmetry:

$$C_3^{(\ell,\text{cross})}((a,b,c),:) := \sum_{z=1}^{v}\left(p_{\text{bin}} R_{z\to(a,b)}^{(\ell)} + p_{\text{ter}}\sum_{d} R_{z\to(d,a,b)}^{(\ell)}\right)\sum_{z=1}^{v}\widetilde{P}_{c,z'}^{(\ell)} C_2^{(\ell-1)}((z,z'),:). \tag{115}$$

By writing $\widetilde{P}^{(\ell)}$ as $J + \widetilde{\Delta}^{(\ell)}$, since $\sum_{z'} C_2^{(\ell-1)}((z,z'),:) = C_1^{(\ell-1)}(z,:)$, we get

$$C_3^{(\ell,\text{cross})}((a,b,c),:) = \sum_{z}\left(p_{\text{bin}} R_{z\to(a,b)}^{(\ell)} + p_{\text{ter}}\sum_{d} R_{z\to(d,a,b)}^{(\ell)}\right)\frac{1}{v} C_1^{(\ell-1)}(z,:)$$

$$+ \sum_{z}\left(p_{\text{bin}} R_{z\to(a,b)}^{(\ell)} + p_{\text{ter}}\sum_{d} R_{z\to(d,a,b)}^{(\ell)}\right)\sum_{z'}\widetilde{\Delta}_{c,z'}^{(\ell)} C_2^{(\ell-1)}((z,z'),:)$$

$$:= C_3^{(\ell,\text{cross},1)}((a,b),:) + C_3^{(\ell,\text{cross},\Delta)}((a,b,c),:) \tag{116}$$

– The first term $C_3^{(\ell,\text{cross},1)}$ is independent of the symbol $c$ and very similar to the expansion of the root-to-pair covariance from Eq. 98: there is the same binary term, and one of the two ternary terms (no cross term), times an extra factor of $v^{-1}$. As proved in the previous section (subsection C.2), the binary term dominates the norm for all valid pairs of binary siblings $(a,b)$. Hence, for all $(a,b) \in \mathcal{B}_\ell$,

$$\|C_3^{(\ell,\text{cross},1)}((a,b),:)\|_2^2 \to \frac{\|C_2^{(\ell,\text{bin})}((a,b),:)\|_2^2}{v^2} \to \frac{1}{v^4 m_2^2}\frac{(p_2^2/2)^{\ell-1}}{m_2^{\ell-1}}. \tag{117}$$

As $v^4 m_2^2 = O(v^6) = O(v^2 m_3^2)$, this term has the same norm scaling as the contribution of ternary siblings (cf. Eq. 114). If, instead, $(a,b) \notin \mathcal{B}_\ell$, then the squared norm includes another factor of $v^{-1}$, hence the term can be neglected. In other words,

$$C_3^{(\ell,\text{cross},1)}((a,b),:) \to \frac{1}{v} C_2^{(\ell)}((a,b),:), \tag{118}$$

hence it is removed asymptotically by subtracting the scaled root-to-pair covariance $v^{-1} C_2^{(\ell)}((a,b),:)$.

---

[4]Note that the topology enforces the condition that the symbol $c$ (resp. $a$) is located on the left (resp. right) of its span, fixing the child position in the tensor. However, this property does not affect the conclusions of this section, and we will ignore it.

- The second term $C_3^{(\ell,\text{cross},\Delta)}$ depends on all three symbols. As for the root-to-pair correlations, this term is the largest for valid binary pairs $(a, b)$. In this case,

$$C_3^{(\ell,\text{cross},\Delta)}((a,b,c),:) \to \frac{p_{\text{bin}}}{m_2} \sum_{z'} \widetilde{\Delta}_{c,z'}^{(\ell)} C_2^{(\ell-1)}((z(a,b),z'),:). \tag{119}$$

The squared norm of $C_2^{(\ell-1)}$ is $O(v^{-2}m_2^{-\ell})$ and the application of $\widetilde{\Delta}$ adds another factor of $m_2^{-1}$. Adding the extra factor of $m_2^{-1}$ results in a squared norm of order $v^{-2}m_2^{-\ell-3} = O(v^{-\ell-5})$, the same size as that of the pure ternary contribution by Eq. 114.

We can now prove the correctness of 3 (cf. 3.2 of the main text).

**Proposition C.5** (Correctness of INFERTERNARY from population root-to-pair and root-to-triple covariances)**.** *Consider data generated by a varying-tree RHM in the asymptotic vocabulary size limit $v \to \infty$ with $m_2 = f_2 v$ and $m_3 = f_3 v^2$. Fix a level $\ell \in \{2, \ldots, L\}$. For each triple $(a, b, c) \in \mathcal{N}_\ell^3$, let*

$$w_{ab} := C_3^{(\ell)}((a,b,c),:) - v^{-1}C_2((a,b),:) - v^{-1}C_2((b,c),:) \in \mathbb{R}^v. \tag{120}$$

*Let $(c_z)_{z=1,\ldots,v}$ denote the unit-norm cluster centroids obtained via INFERBINARY($C_2^{(\ell)}$). For all $z \in \mathcal{N}_{\ell-1}$, let $\mathcal{T}_\ell(z) \subset \mathcal{N}_\ell^3$ denote the set of level-$\ell$ ternary sibling triples generated by level-$(\ell-1)$ nonterminal $z$,*

$$\mathcal{T}_\ell(z) = \left\{ (a,b,c) \in \mathcal{N}_\ell^3 \mid (z \to abc) \in \mathcal{R}_3^{(\ell)} \right\}. \tag{121}$$

*Define*

$$\tau_3 := \gamma v^{-3} \sum_{a,b,c} \max_z \langle w_{abc}, c_z \rangle, \qquad \gamma \in (0,1). \tag{122}$$

*Then, with high probability over the random grammar, for all sufficiently large $v$:*

$$\langle w_{abc}, c_z \rangle > \tau_3 \quad \text{if and only if} \quad (a,b,c) \in \mathcal{T}_\ell(z).$$

*Consequently, INFERTERARY reconstructs the full set of level-$\ell$ ternary rules within the same permutation of parent labels selected by INFERBINARY.*

The proof follows from the analysis of the root-to-triple covariance. Indeed, by Eq. 112, one has

$$w_{abc} = \begin{cases} s_{abc}(z) + e_{abc} & \text{for} \quad (a,b,c) \in \mathcal{T}_\ell(z), \\ e_{abc} & \text{otherwise,} \end{cases} \tag{123}$$

where $s_{abc}(z) = p_{\text{ter}} m_3^{-1} C_1^{(\ell-1)}(z(a,b,c),:)$, while

$$e_{abc} \to \frac{p_{\text{bin}}}{m_2} \sum_{z'} \left( \widetilde{\Delta}_{c,z'}^{(\ell)} C_2^{(\ell-1)}((z(a,b),z'),:) + \widetilde{\Delta}_{a,z'}^{(\ell)} C_2^{(\ell-1)}((z',z(b,c)),:) \right). \tag{124}$$

Since, by C.4, the cluster centroids $c_z$ are asymptotically parallel to the row vectors $r_z^{(\ell-1)} = C_1^{(\ell-1)}(z,:)$,

$$\langle w_{abc}, c_z \rangle = \begin{cases} \frac{p_{\text{ter}}}{m_3} \|C_1^{(\ell-1)}(z,:)\|_2 + \frac{\langle e_{abc}, C_1^{(\ell-1)}(z,:) \rangle}{\|C_1^{(\ell-1)}(z,:)\|_2} & \text{for } (a,b,c) \in \mathcal{T}_\ell(z), \\ \frac{p_{\text{ter}}}{m_3} \frac{\langle C_1^{(\ell-1)}(z',:), C_1^{(\ell-1)}(z,:) \rangle}{\|C_1^{(\ell-1)}(z,:)\|_2} + \frac{\langle e_{abc}, C_1^{(\ell-1)}(z,:) \rangle}{\|C_1^{(\ell-1)}(z,:)\|_2} & \text{for } (a,b,c) \in \mathcal{T}_\ell(z'), \ z' \neq z, \\ \frac{\langle e_{abc}, C_1^{(\ell-1)}(z,:) \rangle}{\|C_1^{(\ell-1)}(z,:)\|_2} & \text{otherwise,} \end{cases} \tag{125}$$

By Eq. 124 (we consider only one of the two terms as the other behaves identically),

$$\left\langle e_{abc}, C_1^{(\ell-1)}(z,:) \right\rangle = \frac{p_{\text{bin}}}{m_2} \sum_{z'=1}^{v} \widetilde{\Delta}_{c,z'} \left\langle C_2^{(\ell-1)}((z(a,b),z'),:), C_1^{(\ell-1)}(z,:) \right\rangle. \tag{126}$$

Let us focus on the $z'$ in the sum such that $(z, z')$ is a valid pair of level-$(\ell - 1)$ binary siblings. There are $O(m_2) = O(v)$ such $z'$. As proved in subsection C.2, for such $z'$, $C_2^{(\ell-1)}((z(a, b), z'), :) \to (p_{\text{bin}}/m_2)C_1^{(\ell-2)}(z'', :)$, where $z''$ is uniquely determined by $z(a, b)$ and $z'$. Other values of $z'$, where $(z, z')$ is not a valid pair of binary siblings, the norm of $C_2$ is smaller by a factor of $1/v$, hence they yield subdominant contributions to $\left\langle e_{abc}, C_1^{(\ell-1)}(z, :) \right\rangle$. After writing $C_1^{(\ell-1)}(z, :) = \sum_y \widetilde{\Delta}_{z,y}^{(\ell-1)} C_1^{(\ell-2)}(y, :)$, we get

$$\left\langle e_{abc}, C_1^{(\ell-1)}(z, :) \right\rangle = \frac{p_{\text{bin}}^2}{m_2^2} \sum_{z'=1}^{v} \widetilde{\Delta}_{c,z'}^{(\ell)} \sum_y \widetilde{\Delta}_{z,y}^{(\ell-1)} \left\langle C_1^{(\ell-2)}(z''(a, b, z'), :), C_1^{(\ell-2)}(y, :) \right\rangle$$

$$= \frac{p_{\text{bin}}^2}{m_2^2} \sum_{z'=1}^{v} \widetilde{\Delta}_{c,z'}^{(\ell)} \left( \widetilde{\Delta}_{z,z''}^{(\ell-1)} \| C_1^{(\ell-2)}(z'', :) \|_2^2 + \sum_{y \neq z''} \widetilde{\Delta}_{z,y}^{(\ell-1)} \left\langle C_1^{(\ell-2)}(z''(a, b, z'), :), C_1^{(\ell-2)}(y, :) \right\rangle \right).$$

(127)

Both terms in brackets can be bound with the same scale in $v$. Since $y \neq z''$, the scalar product is $O_P(\| C_1^{(\ell-2)}(y, :) \|_2^2 / \sqrt{v})$ by C.3, while the multiplication by $\widetilde{\Delta}_{z,y}^{(\ell-1)}$ adds at most a factor of $1/\sqrt{m_2}$ (or $1/m_2$ to the squared norm), resulting in $O_P(\| C_1^{(\ell-2)}(y, :) \|_2^2 / \sqrt{m_2 v})$. The other term respects the same bound as $\Delta_{z,z''}$ is $O_P(1/\sqrt{m_2 v})$. Thus, considering also the multiplication by $\widetilde{\Delta}^{(\ell)}$ outside the bracket and the additional factor of $m_2^{-2}$, and using $\| C_1^{(\ell-2)}(y, :) \|_2^2 = O(v^{-\ell})$ from C.3,

$$\left\langle e_{abc}, C_1^{(\ell-1)}(z, :) \right\rangle = O_P\left( \frac{\| C_1^{(\ell-2)}(y, :) \|_2^2}{m_2^3 \sqrt{v}} \right) = O_P\left( v^{-7/2 - \ell} \right).$$

(128)

Since $\| C_1^{(\ell-1)} \|_2^2 \to (p_2^2/2)^{\ell-1}/(v^2 m_2^{\ell-1}) = O(v^{-1-\ell})$, we have that, in the asymptotic vocabulary size limit

$$\langle w_{abc}, c_z \rangle \to \begin{cases} \frac{p_{\text{ter}}}{m_3} \| C_1^{(\ell-1)}(z, :) \|_2 (1 + O_P(v^{-1/2})) & \text{for } (a, b, c) \in \mathcal{T}_\ell(z), \\ O_P\left( \frac{\| C_1^{(\ell-1)}(z, :) \|_2}{m_3 \sqrt{v}} \right) & \text{otherwise,} \end{cases}$$

(129)

Since there are $vm_3$ valid triples $(a, b, c)$ ($m_3$ for each $\mathcal{T}_\ell(z)$), $\tau_3 \to \gamma \sqrt{f_3} p_{\text{ter}} \| C_1^{(\ell-1)}(z, :) \|_2 / m_3$, smaller than all the $\langle w_{abc}, c_z \rangle$'s with $(a, b, c) \in \mathcal{T}_\ell(z)$ for any $\gamma \in (0, 1)$ and sufficiently large $v$.

### C.4. Iteration via candidate nonterminal indicators

The root-to-pair and root-to-triple covariances used by 2 and 3 are obtained via the pair and triple indicators,

$$X_{a,b}^{(\ell, \text{pair})} := \mathbb{E}_\mathcal{G} \left[ \mathbf{1} \left\{ (X_I^{(\ell)}, X_{I+1}^{(\ell)}) = (a, b) | \mathbf{X}^{(\ell)} \right\} \right], \quad X_{a,b,c}^{(\ell, \text{triple})} := \mathbb{E}_\mathcal{G} \left[ \mathbf{1} \left\{ (X_I^{(\ell)}, X_{I+1}^{(\ell)}, X_{I+2}^{(\ell)}) = (a, b, c) | \mathbf{X}^{(\ell)} \right\} \right].$$

(130)

However, the input to our learning task consists of sentence-root pairs $(\mathbf{X} \equiv \mathbf{X}^{(L)}, X^{(0)})$, hence the indicators above are only available for $\ell = L$ (terminals). In this section, we show in detail how to use the inferred level-$\ell$ rules to build indicator functions for pairs and triples of *candidate* nonterminals, then prove that the covariances between the candidate nonterminal indicators and the root converge asymptotically to the root-to-triple and root-to-pair covariances considered in the previous section. As a result, our iterative learning algorithm 1 can infer all the grammar rules using only sentence-root pairs.

#### C.4.1. CANDIDATE NONTERMINAL INDICATORS

The algorithms INFERBINARY and INFERTERNARY can infer level-$L$ rules directly from sentence-root pairs. Once these rules are identified, we can fill a Boolean inside tensor,

$$N_{i,\lambda}^{(L-1)}(z \mid \mathbf{X}) = \mathbf{1} \left\{ (X_i, \ldots, X_{i+\lambda-1}) = (a_1, \ldots, a_{\lambda-1}) \text{ and } (z \to a_1 \ldots a_{\lambda-1}) \in \mathcal{R}^{(\ell)} \right\}.$$

(131)

Formally, this tensor is obtained via the Boolean form of the inside algorithm (24), i.e. CYK. Let us define $\tilde{R}^{(\ell)}_{z \to (a,b,c)}$ and $\tilde{R}^{(\ell)}_{z \to (a,b,c)}$ as Boolean indicators for binary and ternary level-$\ell$ rules. After initializing the level-$L$ (leaf) indicators,

$$N^{(L)}_{i,1}(a \mid \boldsymbol{X}) = \mathbf{1}\{X_i = a\}, \quad N^{(L)}_{i,\lambda}(a \mid \boldsymbol{X}) = 0 \text{ for all } \lambda \geq 2, \tag{132}$$

we get (see App. B for notation)

$$N^{(\ell-1)}_{i,\lambda}(z) = \sum_{a,b=1}^{v} \sum_{q \in \mathcal{I}_l} \tilde{R}^{(\ell)}_{z \to (a,b)} N^{(\ell)}_{i,q}(a) N^{(\ell)}_{i+q,\lambda-q}(b)$$

$$+ \sum_{a,b,c=1}^{v} \sum_{q,r \in \mathcal{I}_l} \tilde{R}^{(\ell)}_{z \to (a,b,c)} N^{(\ell)}_{i,q}(a) N^{(\ell)}_{i+q,r}(b) N^{(\ell)}_{i+q+r,\lambda-q-r}(c) \tag{133}$$

The initialization on unary splits $\lambda = 1$ and the existence of only binary and ternary rules guarantees that $N^{(\ell-1)}_{i,\lambda}$ can be nonzero only for $\lambda \in I_{l-1}$, i.e. the integer interval $[2^{L-\ell+1}, 3^{L-\ell+1}]$. In the first step of the hierarchy, corresponding to $l = L$ in (133), the equation simplifies, as there is only one valid binary split for $\lambda = 2$, $q = 1$, and one valid ternary split for $\lambda = 3$, $q = r = 1$,

$$N^{(L-1)}_{i,\lambda=2}(z) = \sum_{a,b} \tilde{R}^{(L)}_{z \to (a,b)} N^{(L)}_{i,1}(a) N^{(L)}_{i+1,1}(b),$$

$$N^{(L-1)}_{i,\lambda=3}(z) = \sum_{a,b,c} \tilde{R}^{(L)}_{z \to (a,b,c)} N^{(L)}_{i,1}(a) N^{(L)}_{i+1,1}(b) N^{(L)}_{i+2,1}(c). \tag{134}$$

Notice that, due to local ambiguity, these tensors do not identify the true nonterminals of the parse tree that generated the sentence $\boldsymbol{X}$, but only *candidate* nonterminals. Formally, let $\tau(\boldsymbol{X})$ denote one of the parse trees that generate $\boldsymbol{X}$. For each level $\ell = 0, \ldots, L-1$, the sequence of hidden nonterminals $\boldsymbol{X}^{(\ell)}$ defines a segmentation of the sentence. Let us denote this segmentation with $\{(t_k, s_k, X^{(\ell-1)}_k)\}^K_{k=1}$, where $t_k$ indicates the starting position of the $k$-th block ($t_1 = 1$), $X^{(\ell-1)}_k$ the generating nonterminal, and $s_k \in \{2, 3\}$ the arity of the production rule below. A hit in the candidate nonterminal indicator $N^{(\ell)}_{i,\lambda}(z) = 1$ corresponds to a true nonterminal if and only if there is a parse tree of the sentence such that, in the level-$\ell$ segmentation, there is some $k$ for which $t_k = i$, $t_{k+1} - t_k = \lambda$ and $X^{(\ell)}_k = z$. *Spurious* hits can occur for a correct local segmentation with the wrong label (e.g. $t_k = i$ and $t_{k+1} - t_k = \lambda$ but $X^{(\ell)}_k \neq z$), but also for wrong segmentations (there is no $k$ such that $t_k = i$ and $t_{k+1} - t_k = \lambda$ in any of the true parse trees, but the $(i, \lambda)$ portion of the sentence is still compatible with the grammar rules). In general, we can write

$$N^{(\ell)}_{i,\lambda}(z) = N^{(\ell,\text{true})}_{i,\lambda}(z) + N^{(\ell,\text{spur})}_{i,\lambda}(z), \tag{135}$$

where $N^{(\ell,\text{true})}_{i,\lambda}(z)$ is the indicator function of the true level-$\ell$ nonterminal.

Multiplying Boolean tensors $N^{(\ell-1)}_{i,\lambda}(z)$ that correspond to adjacent blocks, we get indicator functions for pairs and triples of candidate nonterminals,

$$N^{(\ell,\text{pair})}_{i,\lambda,\lambda'}(a,b) := N^{(\ell)}_{i,\lambda}(a) N^{(\ell)}_{i+\lambda,\lambda'}(b),$$

$$N^{(\ell,\text{triple})}_{i,\lambda,\lambda',\lambda''}(a,b,c) := N^{(\ell)}_{i,\lambda}(a) N^{(\ell)}_{i+\lambda,\lambda'}(b) N^{(\ell)}_{i+\lambda+\lambda',\lambda''}(c), \tag{136}$$

whose average over starting positions $i$ and block lengths $\lambda$, $\lambda'$, $\lambda''$ generalize the true indicators $X^{(\ell,\text{pair})}_{a,b}$ and $X^{(\ell,\text{triple})}_{a,b,c}$ to candidate level-$\ell$ nonterminals. In particular, after decomposing each indicator into true and spurious contributions, we get

$$N^{(\ell,\text{pair})}_{a,b} := \left\langle N^{(\ell,\text{pair})}_{i,\lambda,\lambda'}(a,b) \right\rangle_{i,\lambda,\lambda} = X^{(\ell,\text{pair})}_{a,b} + N^{(\ell,\text{spur})}_{a,b}, \tag{137}$$

Hence, by the linearity of the covariance,

$$\text{Cov}_{\mathcal{G}}\left[N^{(\ell,\text{pair})}_{a,b}, X^{(0)}\right] = C^{(\ell)}_2((a,b),:) + C^{(\ell,\text{spur})}_2((a,b),:). \tag{138}$$

The same decomposition holds for triple indicators and the root-to-triple covariance.

### C.4.2. ROOT-TO-CANDIDATE-NONTERMINAL-INDICATORS COVARIANCE CONVERGE TO THE TRUE ROOT-TO-NONTERMINAL COVARIANCES

We now prove that the effect of spurious nonterminals on the level-$\ell$ covariances is negligible in the asymptotic vocabulary size limit. As a result, using the candidate nonterminal indicators instead of the true nonterminal sequences does not impede the inference of level-$\ell$ rules via INFERBINARY and INFERTERNARY. We focus on the indicator function of pairs of candidate nonterminals and the root-to-pair covariance for simplicity, but the same ideas and results apply to triples too.

For each value of $(i, \lambda, \lambda')$, we can analyze the contribution to $C_2^{\text{spur}}$ by conditioning on the local topology of the configuration. Let $I = [i, i + \lambda + \lambda' - 1]$ be the window covered by the two adjacent candidates. Define $\Gamma_i$ as the overlap pattern of the true level-$\ell$ segmentation restricted to $I$, i.e. the ordered list of true blocks intersecting $[i, i + \lambda + \lambda' - 1]$, together with their arities and whether they are truncated at the left/right boundaries. We choose $\Gamma_i$ to be purely topological (i.e. it does not include the identities of nonterminals), so that it is independent of the root label $X^{(0)}$. Then, by the law of total covariance,

$$C_{2,(i,\lambda,\lambda')}^{\text{spur}}((a,b),:) = \mathbb{E}_{\Gamma_i}\Big[\text{Cov}\Big[N_{i,\lambda,\lambda'}^{\text{spur}}(a,b), X^{(0)}\Big|\Gamma_i\Big]\Big], \tag{139}$$

For each configuration, $\Gamma_i = \gamma$, Let $\boldsymbol{k}_\gamma = (k_1, \ldots, k_q)$ be the indices of the true level-$\ell$ blocks intersecting the interval $[i, i + \lambda + \lambda' - 1]$ under $\gamma$, and set $\mathbf{X}_\gamma := (X_{k_1}, \ldots, X_{k_q}) \in [v]^q$. Define the probability

$$p_{\gamma,\boldsymbol{z}}(a,b) := \mathbb{P}_{\mathcal{G}}\Big(N_{i,\lambda,\lambda'}^{\text{spur}}(a,b) = 1 \,\Big|\, \Gamma = \gamma, \, \mathbf{X}_\gamma = \boldsymbol{z}\Big), \qquad \boldsymbol{z} \in [v]^q, \tag{140}$$

for the level-$\ell$ nonterminals $\boldsymbol{z}$ and local topology $\gamma$ to generate a block of terminals compatible with the pair of nonterminals $(a,b)$. The event $\{N_{i,\lambda,\lambda'}^{\text{spur}}(a,b) = 1\}$ is conditionally independent of the root given $(\Gamma = \gamma, \mathbf{X}_\gamma = \boldsymbol{z})$, i.e.

$$\text{Cov}_{\mathcal{G}}\Big[N_{i,\lambda,\lambda'}^{\text{spur}}(a,b), X^{(0)}\Big|\Gamma_i = \gamma, \mathbf{Z}_\gamma = \boldsymbol{z}\Big] = 0. \tag{141}$$

Thus, we may write the conditional spurious covariance as the weighted sum over the configurations of $\mathbf{X}_\gamma$,

$$\text{Cov}_{\mathcal{G}}\Big[N_{i,\lambda,\lambda'}^{\text{spur}}(a,b), X^{(0)}\Big|\Gamma_i = \gamma\Big] = \sum_{\boldsymbol{z} \in [v]^q} p_{\gamma,\boldsymbol{z}}(a,b)\, \text{Cov}\Big[\mathbf{1}\{\mathbf{X}_\gamma = \boldsymbol{z}\}, X^{(0)} \mid \Gamma = \gamma\Big]. \tag{142}$$

Plugging back and averaging over $\gamma$ gives the full spurious term.

**Proposition C.6** (Convergence of root-to-candidate covariance to the true root-to-nonterminal covariance). *Consider data generated by a varying-tree RHM in the asymptotic vocabulary size limit $v \to \infty$ with $m_2 = f_2 v$ and $m_3 = f_3 v^2$. Fix a level $\ell \in \{2, \ldots, L\}$. For each valid pair of binary siblings $(a,b) \in \mathcal{B}_\ell$, and local block configuration $(i, \lambda, \lambda')$ compatible with a pair of level-$\ell$ nonterminals,*

$$\|C_{2,(i,\lambda,\lambda')}^{\text{spur}}((a,b),:)\|_2 = o\left(\frac{1}{m_2}\|C_1^{(\ell-1)}(z(a,b),:)\|_2\right). \tag{143}$$

*Since the scale on the right-hand side corresponds to the true level-$\ell$ root-to-pair covariance of valid pairs, the contribution of spurious nonterminals to the root-to-candidate-pair covariance is negligible, hence it does not impede the inference of binary rules via* INFERBINARY*. In addition, for each valid triple of ternary siblings $(a,b,c)$ such that $\exists z \in \mathcal{N}_{\ell-1}$ and $(abc) \in \mathcal{T}_\ell(z)$, and local block configuration $(i, \lambda, \lambda', \lambda'')$ compatible with a triple of level-$\ell$ nonterminals,*

$$\|C_{3,(i,\lambda,\lambda',\lambda'')}^{(\ell,\text{spur})}((a,b,c),:)\|_2 = o\left(\frac{1}{m_3}\|C_1^{(\ell-1)}(z(a,b,c),:)\|_2\right), \tag{144}$$

*implying that the contribution of spurious nonterminals does not impede the inference of ternary rules via* INFERTERNARY*.*

To prove the proposition above, denote the conditional covariance given the local topology with $u_{\gamma,\boldsymbol{z}}$, so that

$$C_{2,(i,\lambda,\lambda')}^{\text{spur}}((a,b),:) = \sum_\gamma \mathbb{P}_{\mathcal{G}}\{\Gamma_i = \gamma\} \sum_{\boldsymbol{z} \in [v]^q} p_{\gamma,\boldsymbol{z}}(a,b)\, u_{\gamma,\boldsymbol{z}}. \tag{145}$$

The number of local topologies $\gamma$ compatible with a given configuration is finite in $v$, hence the average over topologies does not affect the scaling of the norm with $v$. Each term in the sum is a covariance between the root and $q$ level-$\ell$ adjacent nonterminals. $q \geq 2$, because the sentence portion generated by $\mathbf{X}_\gamma = z$ has to cover the portion generated by $(a, b)$, thus the largest terms in the sum have the size of level-$\ell$ root-to-pair covariances. Therefore, $((i, \lambda, \lambda')$ omitted to ease notation)

$$C_2^{\text{spur}}((a,b),:) = O\left(\sum_{\mathbf{z}\in[v]^2} p_{\gamma,\mathbf{z}}(a,b)\, u_{\gamma,\mathbf{z}}\right) \Rightarrow \|C_2^{\text{spur}}((a,b),:)\|_2^2 = O\left(\sum_{\mathbf{z},\mathbf{z}'} p_{\gamma,\mathbf{z}}(a,b)p_{\gamma,\mathbf{z}'}(a,b)\,\langle u_{\gamma,\mathbf{z}}, u_{\gamma,\mathbf{z}'}\rangle\right).$$

(146)

For all valid pairs $(a, b)$, the scale of $\|u_{\gamma,\mathbf{z}}\|_2^2$ is given by Eq. 100, while the scalar product $\langle u_{\gamma,\mathbf{z}}, u_{\gamma,\mathbf{z}'}\rangle$ for $\mathbf{z} \neq \mathbf{z}'$ is, by C.3, smaller than the squared norm by a factor $O(v^{-1/2})$. Hence,

$$\|C_2^{\text{spur}}((a,b),:)\|_2^2 = O\left(\sum_{\mathbf{z}\in[v]^2} (p_{\gamma,\mathbf{z}})^2 \frac{\|r_c^{(\ell-1)}\|_2^2}{m_2^2}\right) + O\left(\left(\sum_{\mathbf{z}\neq\mathbf{z}'} p_{\gamma,\mathbf{z}} p_{\gamma,\mathbf{z}'}\right)\frac{1}{\sqrt{v}}\frac{\|r_c^{(\ell-1)}\|_2^2}{m_2^2}\right).$$

(147)

$p_{\gamma,\mathbf{z}}(a,b)$ can be thought of as the joint probability of two events: one for the left portion of the span of $\mathbf{z}$ to be compatible with $a$, and one for the right portion to be compatible with $b$. Each event is a union of finitely many events requiring a specific binary (or ternary) tuple to be among the $m_2$ (or $m_3$) rules coming from $a$, having probability $m_2/v^2 = O(v^{-1})$ (or $m_3/v^3 = O(v^{-1})$). Therefore, the joint probability $p_{\gamma,\mathbf{z}}(a,b)$ is $O(v^{-2})$, with constants depending on $(a, b)$ and $\gamma$. In addition, for a fixed topology $\gamma$, each combination of nonterminals $\mathbf{z}$ has a finite probability (over the draw of the random rules) of generating a sequence of terminals compatible with $(a, b)$, hence the number of $\mathbf{z} \in [v]^q$ such that $p_{\gamma,\mathbf{z}}(a,b) > 0$ is $O(v^q)$. As a result,

$$\sum_{\mathbf{z}\in[v]^2} (p_{\gamma,\mathbf{z}})^2 = O\left(v^2 \times (v^{-2})^2\right) = O\left(v^{-2}\right),$$

$$\sum_{\mathbf{z}\neq\mathbf{z}'} p_{\gamma,\mathbf{z}} p_{\gamma,\mathbf{z}'} \leq \left(\sum_{\mathbf{z}\in[v]^2} p_{\gamma,\mathbf{z}}\right)^2 = O\left(1\right),$$

(148)

which, together with Eq. 147, proves Eq. 143. The same strategy can be applied to prove Eq. 144, with $p_{\gamma,\mathbf{z}}(a,b,c) = O(v^{-3})$. For triples, the maximal contribution to the spurious covariance can still come from $q = 2$, as there are topologies where two nonterminals $(z_1, z_2)$ can generate the same span of terminals as nonterminals $(a, b, c)$, but the $O(v^{-3})$ scaling of $p_{\gamma,\mathbf{z}}(a,b,c)$ compensates.

## D. Architectures and Training Details

We present here additional details regarding the neural network architectures and their training. As regards the input data, we represent RHM sequences as a one-hot encoding over the $v$ token symbols. For the CNN and INN, we additionally whiten this $d \times v$ tensor over the $v$ channels, i.e. each of the $d$ pixels has zero mean and unit variance across channels (Cagnetta et al., 2024).

### D.1. Convolutional neural networks (CNNs)

We build deep CNNs in the standard way by stacking convolutional layers. As noted in the main text, we choose a filter of size 4 and stride 2: this ensures that both rule types are always covered (i.e., all patches of length 2 and 3 are read by the filter). Mirroring the layered CFG, we apply a total of $L + 1$ convolutional layers before reading off the label with a linear classifier. To handle sequences of varying length, we pad the input with zeros up to $d_{max}(L) = 3^L$, the maximum sequence length at depth $L$ (for $L = 2$ and $L = 3$, we in fact pad further to $d = 10$ and $d = 30$ respectively, to ensure the length of the feature map stays $> 4$ until the last layer). We have checked that padding with zeros in a symmetric manner, instead of adding zeros to the tail, does not change the results in any appreciable way.

As nonlinear activation function, we use the Rectified Linear Unit (ReLU) (whihout biases, as the input is whitened). The width (number of channels) per layer is set to $H$: we consider throughout the $H \to \infty$, overparametrized limit, with maximal update parametrization (Yang & Hu, 2021) to achieve stable learning. That is, we initialize weights as zero-mean

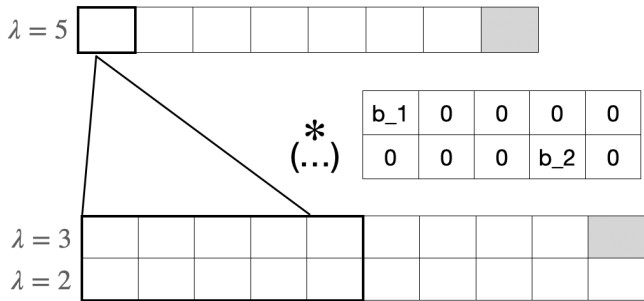

*Figure 15.* Sketch of the dilated filter corresponding to the binary split $q = 3$ for $\lambda = 5$, i.e. splitting the substring of length 5 into two substrings 3-2. This is built by filling in the components of the base binary filter $(b_1, b_2)$ at the appropriate positions. The dilated filter is then convolved with the level-$(L-1)$ feature map to build the $\lambda = 5$ row of the level-$(L-2)$ feature map (note one also has to convolve with the 2-3 filter, not shown, and sum the outputs).

unit-variance Gaussians, scale all hidden layers but the last by $H^{-1/2}$, scale the last layer by $H^{-1}$ (to have vanishing output at initialization), and scale the learning rate with $H$. We find that the $H \to \infty$ limit (in the sense that results are not changed by increasing further) is achieved by scaling according to the total number of ternary rules to be learned, in practice a safe criterion we find is $H \sim 2vm_3 = 2fv^3$. For most parameter values $H = 1024$ more than suffices, while for the largest $fv^3$ cases we increase this up to $H = 2^{11}$ and $2^{12}$.

We consider standard Stochastic Gradient Descent (SGD) training on the cross-entropy loss, without momentum, learning rate set to $H$ and batch size 128. For the general model with varying tree topology, due to the non-vanishing class entropy for any finite $f$, the training loss does not become arbitrarily small. We monitor the test loss and perform early stopping, by detecting a consistent rise or plateauing in the test loss. Once the training is stopped, we record the best model. Test loss data shown in the paper are obtained from averaging over 10 jointly independent realizations of grammar rules, sequences and weight initializations (same for INN, transformer below). The size of the test set is fixed in all cases to 20000: due to the combinatorial explosion of the total number of sequences $P_{\max}$, the probability of encountering a sequence seen during training in the test set is negligible.

### D.2. Inside neural network (INN)

We give here further details on the INN architecture. The basic idea is to use shared separate binary/ternary filters at each layer, and apply these in such a way as to construct feature maps which mirror the inside tensors of the inside algorithm (24). That is, at each level-$l$ we will build feature maps of size $3^L \times (3^{L-l} - 2^{L-l} + 1) \times H$, where the dimensions correspond to start positions (we pad the input with zeros up to $d_{max}(L) = 3^L$), span lengths $\lambda \in \mathcal{I}_l$ (we will denote the total number of span lengths $|\mathcal{I}_l|$ as $N_\lambda(l) = 3^{L-l} - 2^{L-l} + 1$), and finally number of channels $H$. For classification we are interested only in the full span of the sentence, so that at the root level $l = 0$ we consider only the first position $i = 1$ and build a feature map of size $(3^L - 2^L + 1) \times H$. The final linear readout, which yields the logits over the $n_c = v$ classes, is of the same size $H \times n_c$ as in the CNN. This is thanks to the fact that we feed forward the sequence length $d$ of each data point, and use dynamical indexing to read off only at the relevant position in the final feature map. The sequence length $d$ can be calculated from the whitened input $\tilde{x}$ as the sum of squares $d = \sum_{i=1}^{d} \tilde{x}_i^2$.

To build the level-$(l-1)$ feature map from the level-$l$ map following the structure of (24), we will proceed by considering separate each value of the new span length $\lambda \in \mathcal{I}_{l-1}$. For a given $\lambda$ we identify all the corresponding binary/ternary splits, i.e. all possible values $\{q\}$ and $\{q, r\}$ satisfying the constraints expressed in (24). For each binary/ternary split, we will build a *dilated* filter using the shared parameters of the base binary/ternary filter. To illustrate this, consider building the level-$(L-2)$ map from the level-$(L-1)$, in particular for $\lambda = 5$. In this case there are two binary splits, $q = 2$ and $q = 3$, corresponding respectively to splitting 5 as $2-3$ or as $3-2$. If we consider a 2D convolution, we will then want to construct e.g. the second of these ($r = 3$) as a dilated filter by filling in the components of the base binary filter $(b_1, b_2)$ into a larger $2 \times 5$ size filter which slides along the map below (see Fig. 15). We proceed analogously for ternary splits, filling in the base ternary filter $(t_1, t_2, t_3)$ at appropriate positions.

For efficient implementation, at each span length $\lambda \in \mathcal{I}_{l-1}$ we first build all the relevant filters, stack them, perform all the convolutions in parallel and finally sum their outputs. This leaves a single loop over the span lengths $\lambda \in \mathcal{I}_{l-1}$ to build the level-$(l-1)$ feature map. In practice, in the code we first flatten the level-$l$ map into a linear $N_\lambda(l) \times 3^L$ object by

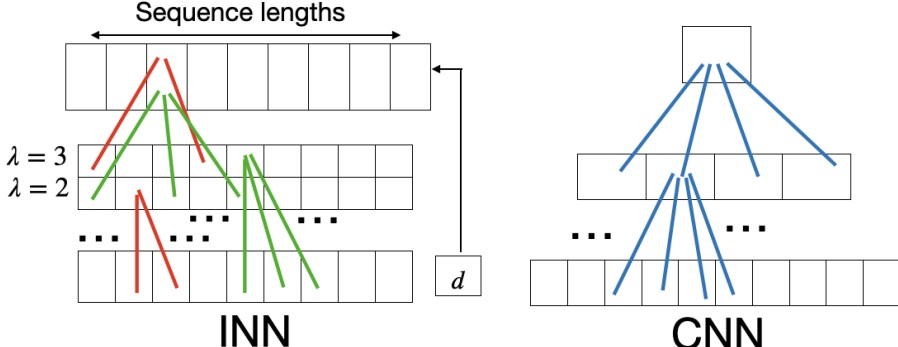

*Figure 16.* Sketch comparing the connectivity of the INN to the CNN for $L = 2$. While in the CNN we consider a single (shared) filter of length 4 and stride 2 at each layer, in the INN we consider separate shared binary/ternary (red/green) filters at each layer, which are applied to precisely build feature maps mirroring the inside tensors (24). The sequence length $d$ is fed forward to the last layer (shown as example is $d = 6$, which receives input from the two parses $3 - 3$ and $2 - 2 - 2$).

interleaving the span lengths, and perform 1D instead of 2D convolutions. In this representation, the kernel length of the dilated filters is $N_\lambda(l) \times \lambda$, and we perform 1D convolution with stride $N_\lambda(l)$; for a binary split defined by $q$, we fill in the base filter $(b_1, b_2)$ at positions $(q - 2^{L-l}, N_\lambda(l) \times q + \lambda - q - 2^{L-l})$ of the dilated filter, while for the ternary split $q, r$ we fill in $(t_1, t_2, t_3)$ at the three positions $(q - 2^{L-l}, N_\lambda(l) \times q + r - 2^{L-l}, N_\lambda(l) \times (q + r) + \lambda - q - r - 2^{L-l})$.

For initialization and training, we follow the same settings as in the CNN. In Fig. 16 we provide a sketch comparing the connectivity of the two architectures.

### D.3. Encoder-only transformer

We build transformers by stacking standard Multi-Head Attention layers (Vaswani et al., 2017) with layer normalization and multi-layer perceptron in between. The one-hot encoded input (padded with zeros up to $3^L$) is firstly mapped into the embedding space of dimension $d_{emb}$ by a learnable linear projection, to which we further add learnable positional embeddings. We choose $d_{emb}$ and number of heads $n_h$ large enough so that increasing them further does not improve performance, in particular $d_{emb} = 512$ and $n_h = 16$. We make standard choices for the per-head width, set to $d_{head} = d_{emb}/n_h$, and the hidden size of the MLP, $d_{mlp} = 4 \times d_{emb}$.

We perform BERT-style classification in the standard way, by pre-pending a dummy [CLS] token to each sequence, which is one-hot encoded as an additional symbol in the vocabulary. To obtain the logits over the $n_c$ class labels, we read out this initial position at the last layer. For training, we consider the same initialization as the CNN, but use instead an adam optimizer with learning rate tuned to $5 \times 10^{-4}$.

## E. Empirical evaluation of the learning algorithm

We here implement the learning algorithm for $L = 2$, and quantify empirically its performance as the number of training sentences $P$ is increased. As explained in the paper, from $P$ sentences we compute the position-averaged binary covariance vectors, and cluster these to identify grammatical binary rules (INFERBINARY, (Algo. 2)), using the Kmeans algorithm with number of clusters $= v$ (we recall there are as many groups of synonymic rules as there are parent nonterminals). Combining binary covariances and the raw ternary ones, we can then construct the whitened ternary covariance vectors, and these can be clustered again with Kmeans (number of clusters $= v$) to identify the ternary rules (INFERTERNARY, (Algo. 3)). As a simple measure of performance, we consider the mean group purity: given that we know the true synonymic groups, for each group we can evaluate the most common predicted cluster, and count which fraction of the true synonymic set are in that majority. We then report the average over all groups. For example, if $v = 16$, and $f = 1/4$ (which we will consider for the figure), $m_3 = 64$, so a purity of 0.9 means that 58 out of 64 rules in that synonymic set landed in the same cluster.

In the top panel of Fig. 17, we show this performance as a function of P, for fixed $f = 1/4$ and increasing $v$, for binary (full) and ternary (dashed) rules. In the bottom panel, we confirm the scalings expected from the theory. Binary rules can be clustered at a sample complexity $(vm_2^L)$, while ternary rules can be clustered at the overall complexity $(vm_2^{L-1}m_3)$.

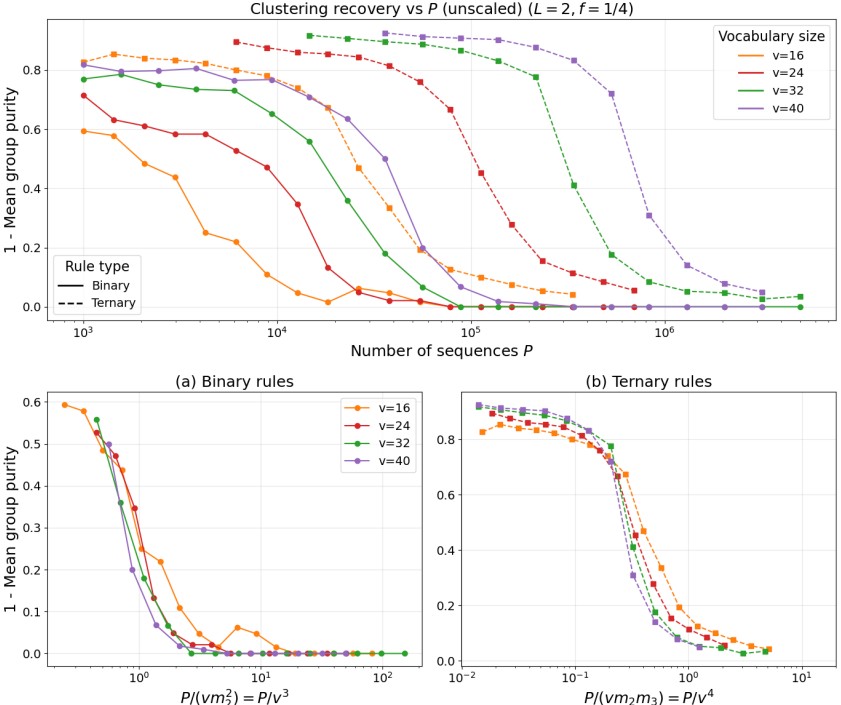

*Figure 17.* Empirical performance of the clustering-based learning algorithm at $L = 2$. **Top:** mean group purity for binary (solid) and ternary (dashed) rules as a function of the number of training samples $P$, at fixed $f = 1/4$ and increasing $v$. **Bottom:** corresponding sample complexity scalings, confirming the theoretical predictions $P^* \sim vm_2^L$ for binary rules and $P^* \sim vm_2^{L-1}m_3$ for ternary rules.

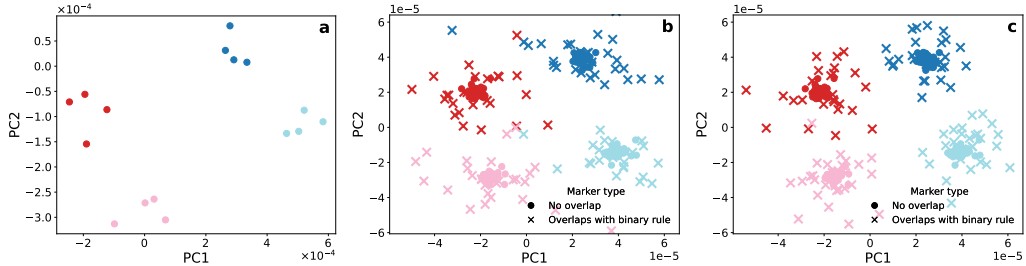

*Figure 18.* Illustration of clustering with parameters $v = 16$, $f = 1/4$ ($m_2 = 4$, $m_3 = 64$) and $L = 2$, acting on a training set of $P=8\times10^6$ sentences. **a** shows the clustering of root-to-pair covariance vectors (here projected on two dimensions with PCA), i.e. INFERBINARY (Algo. 2). For visualization purposes, we show only 4 out of $v$ clusters in all panels; different nonterminals correspond to different colors. **b** highlights that ternary rules that overlap with some grammatical binary rule (denoted by crosses, contrasted with nonoverlapping ternary rules denoted by dots) do not cluster. **c** removes the binary contribution from the root-to-triple covariances, showing how the clusters become cleanly separated (it becomes perfect for $v \to \infty$), i.e. INFERTERNARY (Algo. 3).

In Fig. 18 instead, for a fixed large $P$ we visualize the clustering (via PCA) underlying INFERBINARY and INFERTERNARY, which allow to recover the grammatical binary and ternary rules respectively.

## F. Empirical signal-to-noise ratio (SNR)

We give here further details concerning the empirical signal-to-noise (SNR) measurement, which, as stated in the main text, we use to obtain an alternative $P^*$ prediction which we can compare with the values extracted from CNNs (see Fig. 5(d)). Given that, as predicted by the theory, sample complexity is dominated by the task of grouping synonymic level-$L$ ternary rules, the SNR should reflect how the noisy correlations one would extract from a finite training set of size $P$ approach their clean asymptotic values. Following this logic, we introduce the following quantity based on the conditional probability distribution across the class labels $\alpha = 1, \ldots v$ given a triple $(a, b, c)$ (which contains equivalent information to

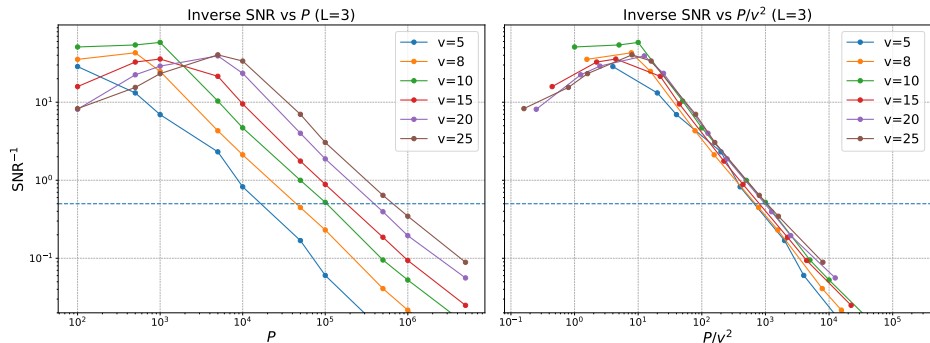

*Figure 19.* Inverse SNR curves $\mathrm{SNR}^{-1}(P)$, computed at $L = 3$ for different $v$, with $f = 1/v$. The dashed lines indicate the threshold $\mathrm{SNR}^{-1}(P^*) = 0.5$ which we set in order to extract the sample complexity prediction shown in Fig. 5(b).

the covariances used in Sec. 3) at a specific position defined by $I$, i.e.

$$\mathrm{SNR}_P\left((a,b,c)\right) = \frac{\|Pr\left(\alpha|(X_I=a, X_{I+1}=b, X_{I+2}=c)\right) - v^{-1}\vec{1}\|^2}{\|\hat{Pr}\left(\alpha|(X_I=a, X_{I+1}=b, X_{I+2}=c)\right) - Pr\left(\alpha|(X_I=a, X_{I+1}=b, X_{I+2}=c)\right)\|^2} \tag{149}$$

where we choose $I = \lfloor\langle s\rangle^L/2\rfloor$, i.e. at the center of the typical sequence length, to avoid boundary effects. $\hat{Pr}$ denotes the empirical estimate of this quantity after $P$ samples. Note that in (149) we have the ratio of squared norms of two $v-$dimensional vectors: in the numerator, the "signal" measures how much the asymptotic values deviate from the uniform distribution $v^{-1}$, which would be the case if correlations were removed making the grammar unlearnable. In the denominator, the "noise" measures how much the empirical estimates deviate from the asymptote due to sampling noise. To obtain the final SNR, we average (149) across all grammatical triples (i.e. ternary rules) $(a, b, c)$.

In Fig. 19, we illustrate how we obtain the SNR $P^*$ prediction for $L = 3$. For a range of training set sizes $P$, we firstly extract the asymptotic probabilities, and for each $P$ calculate (149) averaged across triples. We then fix a lower threshold on the inverse SNR curves, i.e. $\mathrm{SNR}^{-1}(P)$, to obtain a sample complexity. Although there will be a dependence on the precise threshold value, for the reasonable choice $\mathrm{SNR}^{-1}(P^*) = 0.5$ we obtain an excellent agreement with the $P^*$ from CNN data, as shown in Fig. 5(b).

