# OpenReview forum: "Deep networks learn to parse uniform-depth context-free languages from local statistics"
_ICML.cc/2026/Conference — ICML 2026 regular_

### Official Review · Reviewer_j9Nd · 2026-02-26

**Soundness:** 2
**Presentation:** 3
**Significance:** 3
**Originality:** 3
**Overall Recommendation:** 3
**Confidence:** 3

**Summary:**

The paper studies how neural networks learn grammars (e.g. PCFG), which is a hard problem because the PCFG learning itself is NP-hard. To effectively lay the study ground, the paper first introduces a subclass of PCFG where it is uniform depth with production rules.

The authors show that the grammar can be learned in polynomial time by proposing a moment based concrete algorithm and shows its sample complexity (equation 8). Emprically, the author has chosen one setting in this grammar realization and test the learning capability for CNN/transformers/INN, the authors show that under the chosen setting, All the architecture roughly follow the predicted sample complexity with some structural differentce due to their each inductive bias.

**Compliance With Llm Reviewing Policy:**

Affirmed.

**Key Questions For Authors:**

As said in the weakness points, I would like to see more empirical results. This includes but not limited to:
1) different value of f
2) different number of productions for binary/tenary productions (m2/m3)

I would welcome the authors to give different analysis for example for the second study, but even for ranges still obeying the sample complexity according to equation (8) is fine. The idea is to see how much this sample complexity can roughly hold for neural network learning and especially when it breaks.

**Limitations:**

yes

**Strengths And Weaknesses:**

- Strength: The paper tackles a very difficult and arguably long standing important question: how does neural networks learn grammars. The paper offers an interesting angle to tackle this problem. Although the question is generally proven untrackable, the paper constructs 1) a subclass of grammar that can be learned 2) a concrete algorithm where it is possible to analyse the sample complexity. By doing these analysis, the paper can look for the empirical question on if neural networks follow similar learning mechanisms (or at least some neural networks). The paper shows great empirical success in adopting the above mentioned appraoch and found that not only it predicts the CNN (where the theory was inpired on) but also generalizes well on INN and transformer architectures.

- Weakness: I feel that while the approach is novel and the problem is important, the paper does not fully exploit the framework it has laid out. For example, the paper chosen an f=0.25. What happens when f=0.3 making the learning harder? What about f=0.05 where the learning is much easier (is neural networks able to take short-cuts in these scenarios)? I believe for such theoretical papers, it will be of great value to know the boundary that the exposed theory holds. Another question is the binary/tenary distribution where the paper simply takes equal f. Such questions I believe are of value because 1) in real languages, we know that tenary is more scarce than binary 2) it enriches the theory. This missing makes the paper seeming have a great theory and succeed on "one example" but as readers, it is hard to know how much the theory generalizes and when it breaks.

---

> ### Author Rebuttal · Authors · 2026-03-31
>
> We thank the reviewer for finding our work important. Their score, however, is limited, presumably because of the criticism that we did not test our predictions extensively. We provide here the additional tests they asked for, and argue below that some other proposed tests are already present in the paper.  We hope that the referee will be satisfied and raise their mark substantially.
>
> # 1. “$f$ is not varied in the empirical tests”
>
> We do not only consider $f=1/4$ for empirical tests. In Fig.1, we fixed $m_2$, so that $f$ is in fact decaying with increasing $v$ as $f=1/v$. The predicted scaling of the sample complexity with varying grammar parameters (Eq. (8)) holds in both cases. Notice that, after replacing $m_2$ and $m_3$ in Eq. (8) with $f_2 v$ and $f_3 v^2$, respectively, the sample complexity is a growing function of $f_2$ and $f_3$. This mirrors the intuition that higher local ambiguity should make the problem “harder”.
>
> Regarding the comment on increasing $f$ even further, we consider $f=1/4$ already as a large $f$ regime: as can be seen in Fig. 3, $f$ cannot be increased significantly further without developing global ambiguity. Remaining in the globally unambiguous regime is important: (i) it is an assumption for our learning algorithm, and (ii) sentences in natural languages are typically not globally ambiguous.
>
> # 2. Checking theory for different values of $f_2$, $f_3$
>
> As pointed out explicitly in Sec. 2 (line 159), our result on sample complexity, which depends on both $m_2$ and $m_3$,  holds for different values of $f_2$ and $f_3$.  We chose to illustrate and test our more general prediction in the specific case $f=f_2=f_3$, but it is straightforward to test our result when these two parameters differ, as we do now. We have now carried out additional training experiments with CNNs, considering a range of different $v$ and different combinations of values of $f_2, f_3$, where $f_2, f_3 \leq 1$. As can be seen in this [figure](https://figshare.com/s/1fb5b8e710dce591e0b2), our theoretical prediction clearly holds in these new cases. This includes cases with $f_2>f_3$, where, as mentioned by the reviewer, ternary rules are scarcer than binary rules. We thank the reviewer for raising this point, **we will include this figure in the revised manuscript** to strengthen our empirical tests even further.

---

> > ### Author Rebuttal · Reviewer_j9Nd · 2026-04-04
> >
> > I thank the reviewer for the check for sample complexity. First, I think it is also important to show the case where learning is supposedly easy (f << 0.25) and it is good to know when the learning breaks with ambiguity with larger f as I asked above.

---

> > > ### Author Response · Authors · 2026-04-06
> > >
> > > We thank the referee for the discussion:
> > >
> > > - “It is also important to show the case where learning is supposedly easy ($f \ll 0.25$)”
> > >
> > > We agree, but we stress that the $f\ll1$ case is already being shown in the paper. Indeed, besides the fixed $f=0.25$ case, we consider (for Fig. 1) $f=1/v$; given that we vary $v$ up to $v=20$, we are clearly probing the $f\ll1$ regime, and our predictions also hold in this case.
> > >
> > > - “it is good to know when the learning breaks with ambiguity with larger $f$ as I asked above”
> > >
> > > We are now clarifying in detail what occurs in this regime, and will improve the manuscript accordingly. As implied by Fig.3, for larger parameter $f$ a finite class entropy appears. It implies that, even for an optimally trained deep net, the test cross-entropy loss cannot decay to zero (being bounded from below by this finite class entropy). For this reason, we had avoided this regime. However, we expect our predictions for sample complexity to hold for any $f$- only the limiting value of optimal performance will change.
> > >
> > > We establish these two results in the [Figure](https://figshare.com/s/3b9dbde6a9d9a1a21c7a), where we consider $f=0.6$ and $f=0.8$ for a range of values of $v$. Key points are that (i) the curves indeed decay only to the class entropy value expected from the theory. (ii) Curve collapse still occurs before the curves plateau; establishing that our result for sample complexity still holds in the large $f$ regime.
> > >
> > > Ultimately, we have provided very complete tests of our theory. We hope that the referee agrees and will raise their mark significantly.

---

### Official Review · Reviewer_qjFL · 2026-03-05

**Soundness:** 3
**Presentation:** 1
**Significance:** 2
**Originality:** 3
**Overall Recommendation:** 3
**Confidence:** 2

**Summary:**

The paper introduces a class of PCFGs with controllable properties, proposes an algorithms of learning such PCFGs from text, and derives its sample complexity. It also empirically tests the predicted sample complexity with real neural architectures.

**Compliance With Llm Reviewing Policy:**

Affirmed.

**Final Justification:**

I will keep my score with low confidence.

**Key Questions For Authors:**

See my questions regarding section 2.1 above.

**Limitations:**

Yes

**Strengths And Weaknesses:**

I do not have the expertise to fully understand section 4 and will leave its evaluation to other reviewers.

My main concern is, with so many assumptions made regarding the synthetic grammar (e.g., uniform depth, unambiguity, non-overlapping nonterminals of different levels), it is vastly different from natural languages. The proposed algorithm is also very different from the neural architectures used in practice. So I'm not sure how much this study would reveal about real language learning by modern neural architectures.

I had a hard time understanding section 2.1. See below:
- Line 134 left: V_l with 0<l<L is never explained. From the context, I can guess V_l is the nonterminals symbols that can appear at level l of a parse tree. Is it correct?
- Line 141 left: Where do you sample rules from? Are you saying that you have a fixed set of pairs/triples of level (l+1) symbols, from which you can sample rules (or more precisely, RHS)?
- Line 158 left: f_2, f_3 are undefined. And how do you come up with the two equations?
- Line 163 left: Why are choosing tuples (i.e., constructing the rule set of the grammar) and assigning a grammar rule probability related here? I guess you are not referring to grammar rule probabilities, but referring to the probabilities of choosing RHS when constructing the rule set, right?
- Line 124 right: "why" of what?

Line 170 left: If you include terminals in a parse tree (as typically done), then P(x|T)=1 if T is a valid parse tree of x. The product of rule probabilities is P(T), not P(x|T). Accordingly, the formula at line 176 is wrong.

Line 097 right: "Progress in this setting is limited." In fact, there was steady progress in the research of PCFG learning over the past ten years. Examples:
- Compound Probabilistic Context-Free Grammars for Grammar Induction. ACL 2019.
- PCFGs Can Do Better: Inducing Probabilistic Context-Free Grammars with Many Symbols. NAACL 2021.
- Dynamic Programming in Rank Space: Scaling Structured Inference with Low-Rank HMMs and PCFGs. NAACL 2022.

---

> ### Author Rebuttal · Authors · 2026-03-31
>
> # 1. Assumptions make synthetic grammar vastly different to natural language
>
> We refer here to our detailed response to point 1 raised by the 1st Reviewer (gN25), in particular for recent evidence connecting our work to natural languages.
>
> Note that a treatment of the mechanism for learning general PCFGs with deep networks, and its associated sample complexity,  is well beyond the current state of the field. We contend that to move the field forward and obtain the first results on sample complexity, additional assumptions must be made on the structure of the data. They can later be relaxed, step by step, in future works. The essential philosophy here is to provide a minimal setting, where core structural challenges of language (notably ambiguity and latent compositional structure) are present, yet retaining full control over the data-generating process (e.g. full analytical control of all language correlations).
>
> # 2. Proposed algorithm different to NN architectures
>
> Tracking the dynamics of deep CNNs is, once again, beyond the state of the field. But as explained in the paper, the learning algorithm we propose is deeply connected to hierarchical convolutional architectures (CNNs) used in practice. As shown in (Cagnetta et al., ‘24), CNN filters acting on local patches can achieve clustering with a single step of gradient descent, while weight sharing across the input dimension ensures that the statistics for clustering are averaged across positions as in our learning algorithm. Although transformers do not have this architecture, we importantly show empirically that they follow the same sample complexity, which is strong evidence for them exploiting the same learning mechanism/algorithm based on local correlations.
>
> # 3. Sec. 2
>
> We thank the reviewer for his careful reading of the manuscript. As detailed in the response to the 2nd reviewer (Wubc), we will make sure the definition of the model is crystal clear. Below, we respond to each comment:
>
> - 134: yes, $V_l$ is the set of disjoint symbols defined at each level.
>
> - 141: yes, we construct the set of all possible pairs/triples of level $(l+1)$ symbols. These sets are of size $v^2$ and $v^3$ respectively, and we sample uniformly without replacement $m_2$ and $m_3$ rules from these for each level-$l$ nonterminal (LHS).
>
> - 158: This will be stated more clearly, but $f_2$ and $f_3$ are simply introduced as control parameters to define the number of rules. The scaling form, $m_s=f_s v^{s-1}$, is chosen so that local ambiguity remains finite in the large $v$ limit. As explained in detail for the simple case $f_2=f_3=f$, defining the number of rules as $m_2=f v$ and $m_2=f v^2$ implies that, for any $v$, there is always a probability $f$ that an arbitrarily chosen tuple (be it binary or ternary) is grammatical, so that $f$ precisely controls local ambiguity.
>
> - 163: Constructing the rule set of the grammar and assigning rule probabilities are indeed separate. And indeed, what we refer to in this paragraph as occurring with probability $f$ relates to constructing the rule set: when we construct the rule set, there is probability $f$ for an arbitrary tuple to be included in the grammar. $f$ further controls the local ambiguity arising from overlaps between rule types (see next point).
>
> - 124: This sentence explains why, even though the binary/ternary rules are chosen separately without ambiguity, together they do lead to ambiguity because they can overlap. As explained there, the substring corresponding to the grammatical ternary rule $abc$, can appear fortuitously if $ab$ is also grammatical (with probability $f$), and the next symbol in the sequence is $c$.
>
> - 170: instead of $P(x,T)$, it should indeed be $P(T)$. We will correct this in our final version, also in the formula at line 176. We are indeed following the standard convention that terminals are included in the parse tree.
>
> # 4. Progress is limited
>
> We stand by this statement and will clarify precisely what we mean here by “progress”. As acknowledged in the paper, there have been many strong and important works on learning PCFGs. What we mean is that, in the setting where no information on parse trees is given, no one has been able to provide results on polynomial sample complexity. Likewise, no testable predictions for the sample complexity of deep NNs learning PCFGs have ever been made. We will add the interesting references mentioned by the referee, but none of them achieves either of these goals. This is for good reasons, and it is safe to assume that showing the learnability or sample complexity of a general PCFG is an impossible task. This ties into the first point of our rebuttal: progress requires further simplifying assumptions.
>
> We hope that the referee will be satisfied with these answers and will raise their score. Otherwise, we would be happy to answer any remaining questions they may have.

---

> > ### Author Rebuttal · Reviewer_qjFL · 2026-04-02
> >
> > 1. I do agree that assumptions must be made to move forward the study of a difficult problem, but nevertheless that is a weakness of the paper.
> > 2. By "by modern neural architectures", I mean the transformer architecture. I believe that the paper would be more appealing if the theoretical study is conducted on transformers. But I do understand that it would be very difficult.
> > 3. & 4. Thanks for the clarification and discussion. Please incorporate them into the next version of the paper.
> >
> > As I said in my review, I lack the expertise to evaluate section 4 and hence have low confidence in my assessment. AC please feel free to down-weight my evaluation.

---

> > > ### Author Response · Authors · 2026-04-03
> > >
> > > - *I do agree that assumptions must be made to move forward the study of a difficult problem, but nevertheless that is a weakness of the paper.*
> > >
> > > We view our methodology, based on assumptions on the structure of data, as a strength rather than a weakness: it allows us to reach results that were inaccessible for a very long time. This original approach also delineates a path for future progress, by sequentially relaxing these assumptions.
> > >
> > > - *By "by modern neural architectures", I mean the transformer architecture. I believe that the paper would be more appealing if the theoretical study is conducted on transformers. But I do understand that it would be very difficult.*
> > >
> > > Our work, based on proposing an algorithm for learning, can be thought of as a theory for both CNNs and transformers: it makes testable predictions for both architectures, which we find to be satisfied.
> > > The referee seems to imply that general PCFGs as well as transformer dynamics must be treated to warrant publication. Such a high bar would likely preclude the publication of any theoretical work in this field.

---

### Official Review · Reviewer_Wubc · 2026-03-12

**Soundness:** 2
**Presentation:** 1
**Significance:** 3
**Originality:** 3
**Overall Recommendation:** 2
**Confidence:** 4

**Summary:**

This paper study the sample complexity of root label classification in the artificial problem of learning under uniform-depth PCFGs generated data. Thereotically, it predicts the root label via estimated span/root label covarience. It characterized the sample complexity in terms of the depth of PCFGs, number of production rules, and vocbulary size both theoretically (via considering the convergence of moments) and empirically via testing againt several neural network classifier.

Disclaim:  I cannot be sure about the technical contribution due to the extremely unclear writting.

**Compliance With Llm Reviewing Policy:**

Affirmed.

**Final Justification:**

At the recurrent form, I recommend rejection. The paper could have potentially interesting findings. However, the writting is not polished with algorithmatic descriptions interwined with theoretical claims. Most importantly, the justification about the task being studied here can be called parsing is either missing, lacking or too far a stretch.

**Key Questions For Authors:**

1. Does your data consist of (X,X^(0)) pair, that is sentence and root label pair? (seems so from 5.1) If that is the case
2.  Does NNs predict the label directly or all spans will try predict the label?

**Limitations:**

yes

**Strengths And Weaknesses:**

Strength:

1. It provides novel analysis on the sample complexity of learning under PCFG.
2. The theoretical analysis seems to be matched by empirical finding over several neural networks.
3. It has many interesting and indepth analysis.

Weakness:
1. The writting is so bad that I cannot be sure about my understanding.
    - Section 2.1 should be defined very mathematically, remove all the uncessary words, find a text book definition of PCFGs to begin. Be very clear about what are the intrinsic parameters and what is your parameterization/assumption. For example, m_2 is the parameter, and m_2=f_2v is your parameterization/assumption. Leaving all the comments regarding their choice later.
    - Move Section 3 later. At this stage, mentioning there is ambiguity is sufficient. The readers still do not know what you are studying.
    - Define your learning problem formally. Section 4 starts with what is not known and lots of word. You can simply define the dataset in the set theoretical language. Now, you define your algorithm mathematically or in an algorithm box. Your comments are a distraction, they can come in later. In particular, Step 3 can not be understand, you said define then just listing a one side equation. Tips: use C_2^L \in \mathbb{R}^{v\times v \times v} to indicates shape in equation. Also, I will just use Cov instead of C.
     -Now, with parameters and learning problem be clearly defined. You can give your theoretical results, and wrap them as theorems /lemmas .  Then, you give your hypothesis regarding critical parameters, other comments and experimental verification.
2. Lack of empirical study on the analysed moment matching algorithm. I suppose you can make a predictor out of covarience.
3. The task being studied is not parsing
4. No study on whether NNs learn the local statstics. Regarding the title isn't it safer to say Local Statistics Explains Deep Learning of Uniform-Depth Context-Free Languages. (still not accurate)

I recommend rejection at this stage, not because this is not a good work, but simply because this is not ready to publish. On the technical side, due to the mess in writting, I cannot be sure about the results either.

---

> ### Author Rebuttal · Authors · 2026-03-31
>
> All reviewers found our work interesting. Yet, reviewer Wubc (and to some extent qjFL) found our presentation poor and gave a low score. Reviewer gN25, by contrast, found our work well-written and clear. This is revealing of the interdisciplinarity of this field: some referees insist on formal mathematical presentations, while others enjoy more intuitive discussions. Because we believe that this work is important and should speak to a broad audience, **we will modify the text to make it both formal and intuitive**. In particular, **we will implement reviewer Wubc's recommendations as follows:**
>
> # 1. Definitions and model setup
>
> In the revision of Sec. 2.1, we will rewrite the opening of the paper so that it begins with a general definition of a PCFG as a tuple $(\mathcal{N}, \Sigma, \mathcal{R}, S, \mathcal{P})$ (respectively, nonterminals set, terminals set, rules, start symbol and rules probability), followed by the special case of a *layered PCFG*, i.e. a PCFG whose nonterminals are partitioned into levels and whose production rules connect level $\ell$ only to level $\ell+1$. We will then define our varying-tree random-hierarchy model as a particular layered PCFG with only binary and ternary productions.
>
> In addition, we will add a new Sec. 2.2, giving a formal definition of the learning problem: The dataset consists of i.i.d. pairs $(\mathbf{X}, X^{(0)})$, where $\mathbf{X}$ is the terminal sentence and $X^{(0)}$ is the root label. The learning task is to predict the conditional distribution of the root given the sentence. We will make explicit that this is a *root-classification* task rather than supervised parsing.
>
> # 2. Ambiguity
>
> We agree that the current Sec. 3 appears too early. In the revision, we will therefore move the detailed ambiguity analysis to the appendix, and keep in the main text only a short Sec. 2.3 introducing the distinction between *local* and *global* ambiguity.
>
> # 3. Algorithm and theorem-level results
>
> In the revision, we will open Sec. 3 with an algorithm box that defines the method level by level: (i) compute the root-to-pair and root-to-triple covariance tensors; (ii) identify level-$L$ binary rules by clustering the root-to-pair covariance vectors; (iii) identify level-$L$ ternary rules by clustering the *whitened* root-to-triple covariance vectors; (iv) use the inferred rules to build indicators/counts for candidate adjacent nonterminal pairs and triples; (v) iterate the same procedure upward through the hierarchy to recover rules at all levels, until the final candidate root label is obtained. We will also revise the notation so that all tensor shapes and objects are defined explicitly in the statement of the algorithm.
>
> We will then formulate the correctness of the algorithm as a theorem, stating that under: (a) a globally unambiguous data-generating distribution, so that the root label is uniquely determined with high probability by the sentence; (b) access to the exact population moments used by the algorithm; and (c) the asymptotic regime $v\to\infty$ with $m_2=f_2 v$ and $m_3=f_3 v^2$, for finite $f_2,f_3$, the iterative moment-matching procedure recovers the grammar layer by layer, thus solving the root-classification problem.
>
> We will formulate the sample complexity of moment-estimation as a second theorem, then conjecture that the larger of the moment-estimation thresholds controls the sample complexity of neural networks trained by gradient descent on the same task, and position the following empirical sections as tests of this conjecture. We think this logic better separates what is established mathematically from what is hypothesised and then validated experimentally.
>
> # 4. Empirical evaluation of the moment-matching algorithm
>
> In the revision, we will add experiments verifying that the key steps of the moment-matching procedure succeed at the sample complexity predicted by the theory, as already shown in this [figure](https://figshare.com/s/3166b5f0967cb10c236c) for the full set of bottom-layer rules.
>
> # 5. Parsing
>
> We will revise the framing claims so that “parsing” is tied to latent-structure recovery required for root prediction, via the CYK-like procedure of our algorithm, rather than to the supervised label itself. Whether NNs parse is typically studied by probing trained networks, and evaluating constituency F1 score; evaluating the classification task, equivalent to the score on the class label, is strong evidence that the trained NN has learned to parse, as recovering this label accurately requires, by definition, the full tree of latents.
>
> # 6. NNs learn from local statistics
> The essence of our core sample complexity result is that learning is only feasible (not cursed by dimensionality) if the structure of the grammar is acquired from local statistics. Thus, testing our non-trivial prediction across various architectures is already very strong evidence for NNs learning from local statistics, and hence justifies the title.

---

> > ### Author Rebuttal · Reviewer_Wubc · 2026-04-04
> >
> > The author mostly agree with the suggestions.
> >
> > The remaining issue:
> >
> > As we don't have a clear technical presentation of the result, we are still not sure whether parsing ability is implied.
> >
> > In particular, the author suggests
> > > evaluating the classification task, equivalent to the score on the class label, is strong evidence that the trained NN has learned to parse, as recovering this label accurately requires, by definition, the full tree of latents.
> >
> > This is not true in general. Sentiment analysis does not require parsing. In fact, parsing (I worked on parsing in the past) has never been shown to be crucial for any downstream tasks (well, maybe some toy tasks).  Even in vision, models learn to rely on local statistical information for prediction without understanding the global shape.
> >
> > It will really depends on the theoretical result.

---

> > > ### Author Response · Authors · 2026-04-06
> > >
> > > We thank the reviewer for the rapid reply, leading us to clarify further our manuscript along the following lines. We agree that in general, classification does not imply parsing. What we argue is that classification of the top node in *this* class of models requires parsing. Direct evidence includes the following:
> > >
> > > (i)           Most importantly (and justifying the title), our theoretical prediction for sample complexity is based on an algorithm that parses. The fact that deep NNs exhibit precisely the predicted sample complexity strongly suggests that they adopt a mechanism similar to the one we propose—otherwise, the agreement between theory and experiments on NNs would be a very surprising coincidence.
> > >
> > > (ii)         In the simpler Random Hierarchy Model (with a single branching ratio—i.e., a fixed rule length—and hence a fixed tree topology), shallow networks cannot learn to classify the top node with polynomial sample complexity; instead, they exhibit exponential sample complexity, $P^{*}\sim v^d$, as illustrated in this [Figure](https://figshare.com/s/250b2f1760f4d1a158bf) (from Cagnetta et al., PRX 2024). Deep networks are required, as they can represent the tree of latent variables. Higher-level latents are encoded in deeper layers, as confirmed empirically in the above paper.
> > >
> > > The results in [Figure](https://figshare.com/s/250b2f1760f4d1a158bf) are obtained for a simplified version of our model. Our setting allows for varying tree topology and includes, as a subset, topologies with a fixed branching ratio (see, e.g., the first topology in Fig. 12). Therefore, the evidence in the figure above directly implies that, in our classification task with varying tree topology, shallow networks cannot achieve polynomial sample complexity. This, in turn, implies that reconstructing the intermediate latent variables—i.e., parsing—is necessary to solve the task with polynomial sample complexity. Parsing is therefore implied by the task.
> > >
> > > Overall, we thank the referee for his very careful and helpful reviews. We believe that all their concerns have been addressed, leading to a strong paper. We hope that the referee can raise their mark substantially, as we view this work as important and very worthy of publication.

---

### Official Review · Reviewer_gN25 · 2026-03-12

**Soundness:** 3
**Presentation:** 3
**Significance:** 3
**Originality:** 3
**Overall Recommendation:** 5
**Confidence:** 4

**Summary:**

Prior work has established that deep neural networks learn the syntax of language. How they are able to learn hierarchical syntactic structures of language is not fully understood, and this work attempts to provide initial results towards answering that question by exploring a related synthetic setting. Specifically, this work makes use of probabilistic context free grammars (PCFGs) to generate synthetic “languages” which are carefully controlled in terms of degree of ambiguity and the correlation structures. This work then provides an inference algorithm that explains how the DNNs can learn syntactic information based on multi-scale correlations. Finally, this theory of how learning functions is verified through empirical experiments across CNN and transformer based models.

**Compliance With Llm Reviewing Policy:**

Affirmed.

**Final Justification:**

The rebuttal addressed my main concerns and I have changed my evaluation.

This is a strong paper that provides insights into an important problem and provides the basis for significant additional work in this direction.

**Key Questions For Authors:**

PCFGs capture only a small subset of the complexity of natural language. I understand that this is a first step, but what would be suggestive that natural language learning might happen in the same way?

The mechanisms presented do not take into account semantics. Is it possible that the semantics of individual words, in addition to syntax, provide a stronger cue to learning syntactic hierarchies and how would that affect your proposed mechanism?

The fact that DNNs can learn varying-tree random hierarchies from raw text seems to raise the broader question of the role of symbolic scaffolding in language learning. Given this, how should we think about PCFGs here? Do you use them as a useful abstraction for isolating and testing the learning algorithm rather than a commitment to an underlying theory of learning? Do you think that a discussion of this alongside alternative perspectives, including usage-based learning, might be helpful to a reader?

**Limitations:**

Yes

**Strengths And Weaknesses:**

Soundness:
This work, very carefully, and explicitly makes a distinction between how gradient descent works and the learning mechanism proposed herein. The learning method proposed provides insights into how syntactic hierarchies can be learned, rather than a mathematical description of gradient descent.

The algorithm itself is intuitive. It makes sense that parsing is built incrementally, and not globally, and that local ambiguity can be disambiguated using multiscale correlates and finally this mechanism of identifying local constituents which can then be used to recurse upward.

The methods of controlling ambiguity are carefully selected and provide a mechanism for explicitly testing how local ambiguity can be resolved.

There is a gap between synthetic PCFGs and natural language (See questions).

While the use of symbolic generative approaches is well motivated for the current task, it would be helpful to the reader to situate the generative approach alongside alternative views of language learning, including usage-based approaches. Regardless, I believe that the use of PCFGs to test a specific learning algorithm presented herein remains valid.

Presentation:
The paper is clearly presented. Some descriptions could be slightly clearer. For example, Section 2 could benefit from some additional explanation, although I understand that there are limitations of space. Specifically, a clearer high-level roadmap before the content gets dense.

Significance:
The method presented provides a systematic explanation for how local correlates can be used to build hierarchical syntactic structures, which is an important question in language learning.

Originality
The use of a tunable class of PCFGs to systematically explore the mechanism of learning syntactic information.

---

> ### Author Rebuttal · Authors · 2026-03-31
>
> # 1. Limitations of PCFG/ Relevance to natural language learning
>
> We agree PCFGs do not capture some important phenomena of language, such as long-range dependencies and discourse. Our aim is to provide a minimal setting, where core structural challenges of language (notably ambiguity and latent compositional structure) are present, yet retaining full control over the data-generating process. Having a controlled setting allows one to precisely ask the question of whether deep nets can recover latent hierarchical structure from local statistics alone, which can be thought of as a prerequisite for language learning in more realistic settings.
>
> There is very recent evidence (https://arxiv.org/abs/2602.07488) that the mechanism we put forward for parsing is indeed at play when LLMs are trained on natural languages. This work derives and validates neural scaling laws based on the assumption that LLMs become performant on the context scale where token-token correlations become statistically detectable  (that scale thus grows with the training set size). However, this work does not provide a microscopic justification of this assumption. Our work does provide such a justification: indeed, as discussed in our conclusion, from our proposed mechanism for learning PCFGs, it is precisely on that context scale that we expect next token prediction to become performant. **We will add a discussion on that key point and refer to this recent work in the revised manuscript.**
>
> # 2. Role of semantics
>
> This is an interesting point. Semantics can indeed play a role, as proposed for example by semantic bootstrapping (Pinker, 84). Generally, PCFGs can be used to model semantics, and it would be an interesting next step to model semantic cues with synthetic languages. In the bootstrapping theory, children recognize that words label conceptual categories, such as objects or actions, and then use these as a cue to the syntactic categories, such as nouns and verbs. Thinking of the terminals of our language as words, and the latent nonterminals as syntactic categories, one could add additional cues to our synthetic language so that certain nonterminals are made easier to identify due to additional structure. **We will indicate this interesting avenue for future work in the revised conclusions section**.
>
> # 3. Usage-based learning
>
> We thank the reviewer for raising these important points. Our intention is not to take PCFGs as a realistic model of language acquisition, but rather as a controlled setting in which hierarchical structure is explicit and tunable. In this sense, PCFGs serve as a useful abstraction for isolating and testing the learning capabilities of neural networks.
>
> From this perspective, our results suggest a bridge between symbolic and usage-based viewpoints. On the one hand, the data are generated by a symbolic system with well-defined compositional rules. On the other hand, the neural network is trained purely from raw sequences, without access to these rules, and must rely on statistical regularities in the data. The fact that it succeeds indicates that the correlations induced by the underlying symbolic structure are sufficient for a neural system to internalize nontrivial hierarchical organization.
>
> We agree that clarifying this point in the manuscript, and briefly situating our work with respect to alternative perspectives such as usage-based learning, would improve readability, and **we will add the above discussion in the conclusions of the revised manuscript**.
>
> We hope that the referee will be satisfied by these answers and will raise their score. Otherwise, we would be happy to answer any remaining questions they may have.

---

> > ### Author Rebuttal · Reviewer_gN25 · 2026-04-03
> >
> > Thank you very much for the detailed and thoughtful response. I am indeed convinced and will increase my score.
> >
> > **I also agree that the systematic analysis of structured data that you present is NOT a weakness.** It provides a mechanism to perform detailed and systematic analysis that is otherwise not possible.
> >
> > I have increased my score accordingly.

---

### Decision · Program_Chairs · 2026-04-30

**Decision:**

Accept (regular)

**Comment:**

The paper introduces an interesting class of PCFGs to control and later measure ambiguity and correlation recognition, together with a learning mechanism to link learnability and sample complexity with language statistics.  Empirical validation of theoretical findings using synthetically generated data is also carried out.  The authors seem to have responded to all pressing concerns by the reviewers.